# A PROGRAM TO BUILD
# E($n$)-EQUIVARIANT STEERABLE CNNS

**Gabriele Cesa**
Qualcomm AI Research[*]
University of Amsterdam
gcesa@qti.qualcomm.com

**Leon Lang**
University of Amsterdam
l.lang@uva.nl

**Maurice Weiler**
University of Amsterdam
m.weiler.ml@gmail.com

## ABSTRACT

Equivariance is becoming an increasingly popular design choice to build data efficient neural networks by exploiting prior knowledge about the symmetries of the problem at hand. Euclidean steerable CNNs are one of the most common classes of equivariant networks. While the constraints these architectures need to satisfy are understood, existing approaches are tailored to specific (classes of) groups. No generally applicable method that is *practical* for implementation has been described so far. In this work, we generalize the Wigner-Eckart theorem proposed in Lang & Weiler (2020), which characterizes general $G$-steerable kernel spaces for compact groups $G$ over their homogeneous spaces, to arbitrary $G$-spaces. This enables us to directly parameterize filters in terms of a band-limited basis on the whole space rather than on $G$'s orbits, but also to easily implement steerable CNNs equivariant to a large number of groups. To demonstrate its generality, we instantiate our method on a variety of isometry groups acting on the Euclidean space $\mathbb{R}^3$. Our framework allows us to build E(3) and SE(3)-steerable CNNs like previous works, but also CNNs with arbitrary $G \leq$ O(3)-steerable kernels. For example, we build 3D CNNs equivariant to the symmetries of platonic solids or choose $G =$ SO(2) when working with 3D data having only azimuthal symmetries. We compare these models on 3D shapes and molecular datasets, observing improved performance by matching the model's symmetries to the ones of the data.

## 1 INTRODUCTION

In machine learning, it is common for learning tasks to present a number of *symmetries*. A symmetry in the data occurs, for example, when some property (e.g., the label) does not change if a set of transformations is applied to the data itself, e.g. translations or rotations of images. Symmetries are algebraically described by *groups*. If prior knowledge about the symmetries of a task is available, it is usually beneficial to encode them in the models used (Shawe-Taylor, 1989; Cohen & Welling, 2016a). The property of such models is referred to as *equivariance* and is obtained by introducing some *equivariance constraints* in the architecture (see Eq. 2). A classical example are convolutional neural networks (CNNs), which achieve translation equivariance by constraining linear layers to be convolution operators. A wider class of equivariant models are Euclidean steerable CNNs (Cohen & Welling, 2016b; Weiler et al., 2018a; Weiler & Cesa, 2019; Jenner & Weiler, 2022), which guarantee equivariance to isometries $\mathbb{R}^n \rtimes G$ of a Euclidean space $\mathbb{R}^n$, i.e., to translations and a group $G$ of origin-preserving transformations, such as rotations and reflections. As proven in Weiler et al. (2018a; 2021); Jenner & Weiler (2022), this requires convolutions with $G$-*steerable* (equivariant) kernels.

Our goal is developing a program to parameterize with minimal requirements arbitrary $G$-steerable kernel spaces, with compact $G$, which are required to implement $\mathbb{R}^n \rtimes G$ equivariant CNNs. Lang & Weiler (2020) provides a first step in this direction by generalizing the *Wigner-Eckart theorem* from quantum mechanics to obtain a general technique to parametrize $G$-steerable kernel spaces over *orbits* of a compact $G$. The theorem reduces the task of building steerable kernel bases to that of finding some pure representation theoretic ingredients. Since the equivariance constraint only relates points $g.x \in \mathbb{R}^n$ in the same *orbit* $G.x \subset \mathbb{R}^n$, a kernel can take independent values on different orbits. Fig. 1 shows

---

[*]Qualcomm AI Research is an initiative of Qualcomm Technologies, Inc.

examples of orbits in $\mathbb{R}^2$ under the action of continuous planar rotations $G = \mathrm{SO}(2)$ (orbits are rings of different radii) and of discrete rotations by $\frac{\pi}{2}$, $G = \mathrm{C}_4$ (orbits are sets of four points). Hence, Lang & Weiler (2020) suggest *independently* parameterizing orbits of $G$ in $\mathbb{R}^n$ (colored regions in Fig.1).

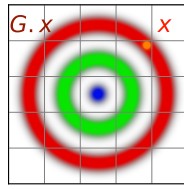

While this guarantees a complete parameterization of $G$-steerable functions, the discretization of the filters requires them to be sufficiently *band-limited*[1]. Indeed, particular care should be taken to avoid *aliasing* effects while discretizing filters on a grid (e.g., to process voxelized data). Lang & Weiler (2020) naturally support band-limiting along each orbit of $G$ but do not restrict the kernel across different orbits.

(a) Orbits of $G = \mathrm{SO}(2)$

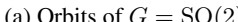
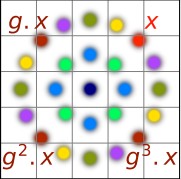

Intuitively, this means that a filter can be decomposed into a finite number of orbits but these should then be patched together with some *smoothness* considerations. This introduces an important requirement and a sensitive design choice. First, building new $G$-steerable CNNs *requires* one to identify the orbits of $G$ in $\mathbb{R}^n$. Second, the choice of orbits to consider affects a filter's band-width, but this relation is hard to explicitly quantify in a general setting. This issue is more severe for smaller groups (particularly, finite ones), as $\mathbb{R}^n$ decomposes into a larger number of orbits. For example, if $G = \mathrm{SO}(2)$ in Fig. 1a, band-limiting along rings is controlled by the Wigner-Eckart theorem but is not enforced along the radial component. If $G = \mathrm{C}_4$ in Fig. 1b, each ring is partitioned into discrete orbits: a larger number of orbits increases the maximum angular frequency, so aliasing is sensitive to this design choice.

(b) Orbits of $G = \mathrm{C}_4$

Figure 1: Orbits in $\mathbb{R}^2$. Each color represents points in the same orbit under $G$'s action.

**Contributions and Outline** In this work, we solve this issue and propose a general solution strategy to build arbitrary CNNs with $G$-steerable kernels, for any compact group $G$. Theorem 2.1 generalizes the result of Lang & Weiler (2020) from $G$-homogeneous spaces to more general spaces $X$ carrying a $G$-action. Essentially, our theorem replaces the harmonic basis on the orbits of $G$ in $X$ with a $G$-steerable basis $\mathcal{B}$ for unconstrained scalar filters (Freeman & Adelson, 1991) over the whole space $X$. Aliasing effects are thereby controlled by the choice of this initial basis. In practice, in this work we focus on Euclidean spaces $X = \mathbb{R}^n$, although the theory developed *holds for any semi-direct product group $X \rtimes G$*, and can be extended with minor changes to *general homogeneous spaces $X$* with compact stabilizers $G$ (as argued in Remark D.15 in Lang & Weiler (2020)). Since a $G'$-steerable basis is $G$-steerable for any subgroup $G \leq G'$, we also propose to use an initial $G' = \mathrm{O}(n)$-steerable basis to support any compact group $G$ with minimal requirements. In summary, the benefits of our method are two-fold: *i)* it allows direct *control on band-width and aliasing* via the initial basis $\mathcal{B}$ and *ii)* it completely disentangles the discretization issues from the choice of $G$, *minimizing the requirements to implement equivariance to new groups*. For example, this enables the parameterization of $\mathrm{C}_4$ filters without discretizing rings into a finite number of points as in Fig. 1b. Algorithm 1 summarizes our method and its requirements.

Since the planar case $n = 2$ was extensively studied in Weiler & Cesa (2019) and since $\mathbb{R}^3$ has a large variety of isometry (sub)groups, in our experiments, we put particular emphasis on the 3D setting. To illustrate the generality of our method, we instantiate it on many different subgroups of $\mathrm{O}(3)$, the group of 3D rotations and reflections, and compare them experimentally. This results in many new 3D convolution networks, equivariant, for example, to icosahedral, axial, cylindrical or conical symmetries. Axial symmetries are particularly relevant since natural scenes generally have vertical orientation while discrete symmetries occur in crystallography or solid state physics; see Sec. G. In Sec.5.1, we discuss many design choices for 3D equivariant networks, and compare them in the experiments in Sec. 5.3. In particular, we find that designs based on different discretizations of the group and pointwise non-linearities to be beneficial when working with volumetric data. Additionally, Sec.5.2 studies the benefit of our basis in terms of equivariance after discretization and accuracy of trained models.

In the Appendix, we derive some results for working with representation theoretic objects to reduce the work needed to implement new equivariant CNNs. This includes real-valued representations of compact groups, harmonic bases for induced representations, numerical irreps decomposition and representation theory of direct product groups. Finally, we implement the program described in this work as a general purpose library based on PyTorch at `github.com/QUVA-Lab/escnn`.

---

[1]A band-limited signal's spectrum has bounded support. Nyquist-Shannon sampling theorem states that only sufficiently band-limited signals are faithfully represented by discrete samples. Otherwise, *aliasing* effects occur.

## 2 THE THEORY OF STEERABLE CNNS AND $G$-STEERABLE KERNEL SPACES

In Sec. 2.1, we discuss preliminaries on steerable CNNs and the equivariance constraints on their filters. Sec. 2.2 describes our generalization of Lang & Weiler (2020) used to parametrize kernels.

### 2.1 EUCLIDEAN STEERABLE CNNS

Equivariant neural networks are characterized by their property to commute with the action of a given symmetry group on their input, intermediate feature spaces, and output. We focus on convolutional networks on Euclidean spaces $X = \mathbb{R}^n$, acted on by the group $(\mathbb{R}^n, +) \rtimes G$, i.e. the *semidirect product* of translations in $(\mathbb{R}^n, +)$ and origin-preserving transformations $G \leq \mathrm{O}(n)$. A general framework of equivariant convolutional models is that of *steerable CNNs*, which we briefly recapitulate here. For more details, see Cohen & Welling (2016b); Weiler et al. (2018a; 2021); Weiler & Cesa (2019).

The fundamental design choice underlying steerable CNNs is that they operate on *feature vector fields*, which are similar to feature maps but differ in that they are associated with a well defined action of $(\mathbb{R}^n, +) \rtimes G$. The geometric *type* of a feature field is prescribed by an orthogonal *group representation* $\rho : G \to \mathbb{R}^{d_\rho \times d_\rho}$ which determines the transformation law of $d_\rho$-dimensional feature vectors under the action of $G$. Specifically, a feature field of type $\rho$ is a map $f : \mathbb{R}^n \to \mathbb{R}^{d_\rho}$ which transforms under $(\mathbb{R}^n, +) \rtimes G$ according to the *induced representation* $\mathrm{Ind}_G^{(\mathbb{R}^n,+) \rtimes G} \rho$:

$$\left( \left[ \mathrm{Ind}_G^{(\mathbb{R}^n,+) \rtimes G} \rho \right] (tg) \, f \right)(x) := \rho(g) f\left( g^{-1}(x - t) \right) \qquad \forall g \in G, t \in (\mathbb{R}^n, +) \,. \tag{1}$$

Intuitively, the induced representation acts by moving feature vectors spatially from $g^{-1}(x - t)$ to $x$ and by transforming each feature $f(x)$ according to $\rho(g)$ (see Fig. 1 in Weiler & Cesa (2019)). Notable examples are scalar or tangent-vector fields, which transform according to the trivial representation $\rho(g) = 1$ or the standard representation $\rho(g) = g$, respectively. Group-convolution networks (Cohen & Welling, 2016a; Kondor & Trivedi, 2018) are special cases with $\rho$ being the *regular representation* of $G$. It is common to define the full feature fields of a network as a direct sum (concatenation) $f := \bigoplus_i f_i$ of multiple individual feature fields $f_i$. The full feature space transforms according to the *direct sum representation* $\rho := \bigoplus_i \rho_i$, which ensures fields transform independently from each other.

Weiler et al. (2018a); Jenner & Weiler (2022) proved that, under mild assumptions, the most general linear equivariant maps between spaces of feature fields are *convolutions with $G$-steerable kernels*. If the input and output fields have types $\rho_{\mathrm{in}} : G \to \mathbb{R}^{d_{\mathrm{in}} \times d_{\mathrm{in}}}$ and $\rho_{\mathrm{out}} : G \to \mathbb{R}^{d_{\mathrm{out}} \times d_{\mathrm{out}}}$, $G$-steerable kernels are convolution kernels $K : \mathbb{R}^n \to \mathbb{R}^{d_{\mathrm{out}} \times d_{\mathrm{in}}}$ satisfying the linear $G$-steerability constraint

$$K(g.x) = \rho_{\mathrm{out}}(g) K(x) \rho_{\mathrm{in}}(g)^{-1} \qquad \forall \, g \in G, \ x \in \mathbb{R}^n \,. \tag{2}$$

Thus, designing steerable CNNs only requires finding a basis of the vector space of $G$-steerable kernels, which is then used to parameterize conventional Euclidean convolutions. Note that Eq. 2 relates the kernel values on all points $g.x$ in the orbit $G.x$, but leaves values on different orbits unrelated.

### 2.2 A WIGNER-ECKART THEOREM FOR SUBGROUP EQUIVARIANT CONVOLUTION KERNELS

First, we need to introduce a few concepts and some notation. To keep the presentation general, in this section we will consider a general space $X$ rather than $\mathbb{R}^n$. Since $G$ is compact, we can assume all representations to be *orthogonal*, i.e., $\rho(g^{-1}) = \rho(g)^{-1} = \rho(g)^T$. To parametrize a kernel $K : X \to \mathbb{R}^{d_{\mathrm{out}} \times d_{\mathrm{in}}}$ satisfying the steerability constraint in Eq. 2, it is convenient to use its *vectorized* form $\kappa(\cdot) = \mathrm{vec}\,(K(\cdot)) : X \to \mathbb{R}^{d_{\mathrm{out}} \cdot d_{\mathrm{in}}}$. The constraint[2] becomes:

$$\kappa(g.x) = \left[ (\rho_{\mathrm{in}} \otimes \rho_{\mathrm{out}})(g) \right] \kappa(x) \quad \forall g \in G, x \in X \,. \tag{3}$$

The matrix $(\rho_{\mathrm{in}} \otimes \rho_{\mathrm{out}})(g)$ is the *Kronecker product* of the two matrices $\rho_{\mathrm{in}}(g)$ and $\rho_{\mathrm{out}}(g)$.

**Irrep Decomposition** A useful property is that there exists a set[3] $\widehat{G}$ of special representations of $G$, called irreducible representations or *irreps*, such that any representation $\rho$ of $G$ can be *decomposed* as a *direct sum* of them: $\rho(g) = Q^T \left( \bigoplus_{i \in I} \rho_i(g) \right) Q$, where $\rho_i \in \widehat{G}$, $I$ is an index set ranging over the elements of $\widehat{G}$ (possibly, with repetition) and $Q$ is an orthogonal matrix. The direct sum $\rho_1(g) \oplus \rho_2(g)$

---

[2] Note that $\mathrm{vec}\,(ABC) = (C^T \otimes A)\,\mathrm{vec}\,(B)$, where $\mathrm{vec}\,(\cdot)$ stacks matrix columns into a vector.

[3] More precisely, irreps come in equivalence classes and the set $\widehat{G}$ contains a representative for each class.

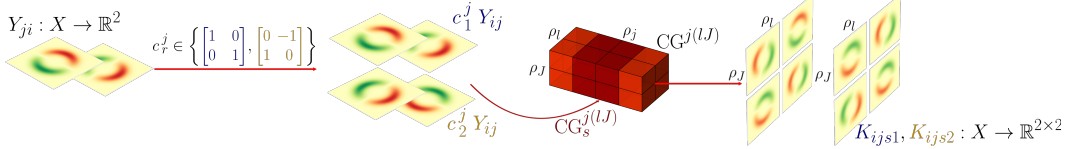

Figure 2: Two (unvectorized) basis kernels ($j=1$, $i=1$, $r \in \{1, 2\}$) from Thm. 2.1 for $G = \mathrm{SO}(2)$, $l = 1$, $J = 2$.

of two matrices is a block diagonal matrix containing the two matrices in its blocks. W.l.o.g., we can assume $\rho_{\mathrm{in}}$ and $\rho_{\mathrm{out}}$ in the kernel constraint from Eq. 2 to be irreps. The solutions for arbitrary $\rho_{\mathrm{in}}$ and $\rho_{\mathrm{out}}$ can be recovered from their irreps decomposition, as explained in Weiler & Cesa (2019). Hence, from now, we will assume irreps $\rho_{\mathrm{in}} = \rho_l$ and $\rho_{\mathrm{out}} = \rho_J$, where $l$ and $J$ are indexes over $\widehat{G}$.

**Tensor Product** The matrix $\rho_{\mathrm{in}} \otimes \rho_{\mathrm{out}}$ built with the Kronecker product in Eq. 3 is itself a representation of $G$, namely a *tensor product* representation. As such, we can decompose it in terms of $G$ irreps: $\rho_l \otimes \rho_J = [\mathrm{CG}^{lJ}]^T \big( \bigoplus_j \bigoplus_s^{[j(lJ)]} \rho_j \big) \mathrm{CG}^{lJ}$, where $[j(lJ)]$ indicates the *multiplicity* of the irrep $\rho_j$, while $\mathrm{CG}^{lJ}$ is the change of basis matrix.[4] The block diagonal structure of the direct sum allows one to distinguish blocks of rows in $\mathrm{CG}^{lJ}$, acted on by individual irreps in the direct sum. The block associated with the $s$-th occurrence of $\rho_j$ is denoted as $\mathrm{CG}_s^{j(lJ)} \in \mathbb{R}^{d_j \times d_l d_J}$.

**Endomorphisms** We are interested in the space of equivariant linear maps between two spaces transforming according to irreps $\rho_j, \rho_{\tilde{\jmath}} \in \hat{G}$. If $j \neq \tilde{\jmath}$, this space is empty, otherwise, this is the *endomorphism space* of $\rho_j$. This is a vector space; assume $\{c_r^j \in \mathbb{R}^{d_j \times d_j}\}_r$ is a basis[5] for it.

**Steerable Basis** To parametrize our kernels, we first need a basis for square-integrable functions on $X$, i.e. for $L^2(X)$. A *G-steerable basis* for $L^2(X)$ is a collection of orthogonal functions $\{Y_{ji}^m : X \to \mathbb{R}\}_{j \in \widehat{G}, m \leq d_j, i \leq m_j}$, with $m_j$ a positive (possibly infinite) integer. Denoting by $Y_{ji} : X \to \mathbb{R}^{d_j}$ the stack $\{Y_{ji}^m\}_{m=1}^{d_j}$, $Y_{ji}$ has the defining property that $\forall g \in G, x \in X$, $Y_{ji}(g.x) = \rho_j(g) Y_{ji}(x)$.

The vectorized constraint in Eq. 3 is equivalent to Eq. 2 with a scalar input field and a $\rho_{\mathrm{in}} \otimes \rho_{\mathrm{out}}$ output one. A (vectorized) kernel $\kappa$ is a $G$-equivariant linear map from $L^2(X)$ to $\mathbb{R}^{d_l \cdot d_J}$, which will be applied convolutionally over a scalar field. Hence, a basis for this kernel space is given by considering the irreps decomposition of $G$'s action on $L^2(X)$ and $\mathbb{R}^{d_l \cdot d_J}$, and, then, equivariant maps between pairs of irreps. The basis $\mathcal{B}$ defines an irrep decomposition of $L^2(X)$, $\mathrm{CG}^{lJ}$ decomposes $\rho_{\mathrm{in}} \otimes \rho_{\mathrm{out}}$ and equivariant maps are parameterized by each irrep's endomorphism basis. This leads to:

**Theorem 2.1** (Basis for $G$-Steerable Kernels). *Let $G$ be a compact group acting on a space $X$. Let $\mathcal{B} = \{Y_{ji}\}_{ji}$ be a $G$-steerable basis for $L^2(X)$. Assume $\rho_{\mathrm{in}} = \rho_l$ and $\rho_{\mathrm{out}} = \rho_J$ in $\widehat{G}$. Under minor conditions, a basis for (vectorized) $G$-steerable kernels over $X$ is given by $\mathcal{K} = \{\kappa_{jisr}\}_{jisr}$, with:*

$$\kappa_{jisr}(x) = [\mathrm{CG}_s^{j(lJ)}]^T \cdot c_r^j \cdot Y_{ji}(x) . \tag{4}$$

The proof is in Appendix B (including the case of complex valued representations); see also Fig. 2.

Designing a new $G$-steerable basis $\mathcal{B}$ can be laborious; however, if $G$ is a subgroup of $G'$ (i.e., $G \leq G'$), a $G'$-steerable basis can be turned into a $G$-steerable one via *group restriction*:

**Group Restriction** Given two groups $G \leq G'$, one can turn an irrep $\rho'$ of $G'$ into a representation $\mathrm{Res}_G^{G'} \rho'$ of $G$ by restricting its domain to $G$. This representation is not irreducible and decomposes as: $\mathrm{Res}_G^{G'} \rho_{j'} = [\mathrm{ID}^{j'}]^T \big( \bigoplus_j \bigoplus_t^{[jj']} \rho_j \big) \mathrm{ID}^{j'}$ where $[jj']$ is the *multiplicity* of $\rho_j$ and $\mathrm{ID}^{j'}$ is the change of basis. As earlier, the block-diagonal structure of the direct sum distinguishes blocks of rows of the matrix $\mathrm{ID}^{j'}$. We denote the block associated with the $t$-th occurrence of $\rho_j$ as $\mathrm{ID}_t^{jj'} \in \mathbb{R}^{d_j \times d_{j'}}$.

**Restricted Steerable Basis** If $\mathcal{B}' = \{Y_{j'i'}\}_{j'i'}$ is a $G'$-steerable basis for $L^2(X)$, the following set is a *G-steerable basis*, with index $i = (j'i't)$ (see Appendix B.3):

$$\mathcal{B} = \{Y_{j(i'j't)} \mid Y_{j(i'j't)}(x) = \mathrm{ID}_t^{jj'} \cdot Y_{j'i'}(x)\}_{j \in \widehat{G}, Y_{j'i'} \in \mathcal{B}', 1 \leq t \leq [jj']} . \tag{5}$$

---

[4]CG stands for *Clebsh-Gordan* since this matrix contains the so-called Clebsch-Gordan coefficients.

[5]In the complex case, such matrices are multiples of the identity, but this space can be larger in the real case.

We present two examples on the circle $X = \mathcal{S}^1$; see Appendix B.4 for more detailed examples.

**Example 1: SO(2)-Steerable Kernels** In Fig. 3, we show a simple example of the basis for a $G = \mathrm{SO}(2)$-steerable kernel space. Since $X = \mathcal{S}^1$ is an orbit of $G$, our parameterization is equivalent to Lang & Weiler (2020). An SO(2)-steerable basis on $X = \mathcal{S}^1$ is given by the circular harmonics: $Y^0_{ji}(\theta) = \cos(j\theta)$ and $Y^1_{ji}(\theta) = \sin(j\theta)$, for $j \in \mathbb{N}$ and $i = 1$. We consider steerable kernels with input irrep $\rho_{\mathrm{in}} = \rho_{l=0}$ and output irrep $\rho_{\mathrm{out}}$ one of $\rho_{J=0}$, $\rho_{J=1}$ or $\rho_{J=2}$.[6] Hence, the kernels have form, respectively, $K_0 : \mathcal{S}^1 \to \mathbb{R}^{1\times 1}$, $K_1 : \mathcal{S}^1 \to \mathbb{R}^{2\times 1}$ and $K_2 : \mathcal{S}^1 \to \mathbb{R}^{2\times 1}$. $\rho_0 \otimes \rho_J = \rho_J$, so we always have $j = J$ and $s = 1$. Additionally, any 2-dimensional irrep $\rho_j$ $(j > 0)$ of SO(2) has a 2-dimensional endomorphism space spanned[7] by $c^j_{r=0} = \begin{bmatrix} 1 & 0 \\ 0 & 1 \end{bmatrix}$ and $c^j_{r=1} = \begin{bmatrix} 0 & 1 \\ -1 & 0 \end{bmatrix}$. This can be verified visually: in Fig. 3b and 3c, the second column is obtained by swapping the rows of the first and by changing the sign of the second row.

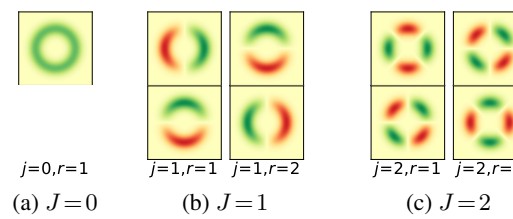

$j{=}0,r{=}1 \qquad j{=}1,r{=}1 \quad j{=}1,r{=}2 \qquad j{=}2,r{=}1 \quad j{=}2,r{=}2$

(a) $J = 0$    (b) $J = 1$    (c) $J = 2$

Figure 3: Basis for $G = \mathrm{SO}(2)$-steerable kernels from Eq. 4. $\rho_{\mathrm{in}} = \rho_{l=0}$ is 1-dimensional. $\rho_{\mathrm{out}} = \rho_J$ for $J \in \{0, 1, 2\}$, which are 1, 2 and 2 dimensional. Each column is an element $K_{jisr}$ of the basis, while each row is an output channel. $i = 1$ always and, since $\rho_0 \otimes \rho_J = \rho_J$, $j = J$.

**Example 2: $\mathrm{C}_4$-Steerable Kernels** In Fig. 4, we show the basis for $G = \mathrm{C}_4$ steerable kernels described by Theorem 2.1, using the $\mathrm{C}_4$-steerable basis generated by Eq. 5 from the circular harmonics basis $\mathcal{B}'$ used for $G' = \mathrm{SO}(2)$ in the previous example. $\rho_j$ occurs in a restricted representation $\mathrm{Res}^{\mathrm{SO}(2)}_{\mathrm{C}_4} \rho_{j'}$ whenever $j' = |j + 4k|$, $k \in \mathbb{Z}$. Indeed, the basis prescribed in Eq. 5 has infinitely many elements, but each element is associated with a circular harmonic of a specific frequency $j'$. It is natural to use only a *band-limited* finite subset of this basis: Fig. 4 shows only frequencies up to $j' = 4$. If $j = 0$ or $j = 2$ and $j' = |j + 4k| > 0$, $\rho_j$ is 1-dimensional but $\rho_{j'}$ is 2-dimensional, so $\mathrm{Res}^{G'}_{G} \rho_{j'}$ contains two copies of $\rho_j$ and $t \in \{1, 2\}$ (see $j' = 2$ or 4 in Fig. 4a). We use the trivial irrep $l = 0$ in the input and $J \in \{0, 1, 2\}$ in the output. $\rho_{j=1}$ has a 2-dimensional space of endomorphisms, spanned by the same basis described in the SO(2) example above, so $r \in \{1, 2\}$ in this case. Note that the parametrization in Fig. 4 is continuous along the ring and does not suffer from the discretization issue in Fig. 1b.

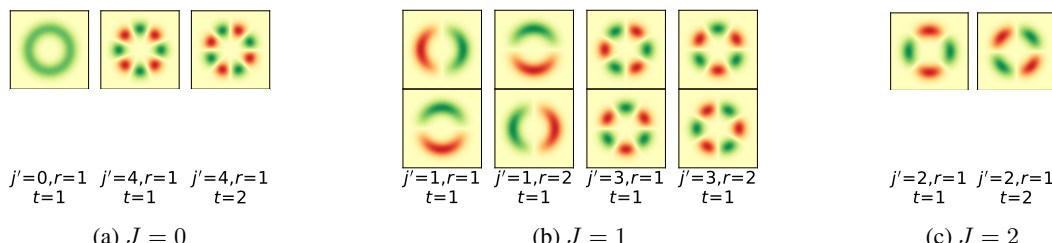

$\begin{array}{cccc} j'{=}0,r{=}1 & j'{=}4,r{=}1 & j'{=}4,r{=}1 \\ t{=}1 & t{=}1 & t{=}2 \end{array}$    $\begin{array}{cccc} j'{=}1,r{=}1 & j'{=}1,r{=}2 & j'{=}3,r{=}1 & j'{=}3,r{=}2 \\ t{=}1 & t{=}1 & t{=}1 & t{=}1 \end{array}$    $\begin{array}{cc} j'{=}2,r{=}1 & j'{=}2,r{=}1 \\ t{=}1 & t{=}2 \end{array}$

(a) $J = 0$      (b) $J = 1$      (c) $J = 2$

Figure 4: Basis for $G = \mathrm{C}_4$-steerable kernels as in Eq. 4 using a steerable basis $\mathcal{B}$ built from circular harmonics using Eq. 5. $\rho_{\mathrm{in}} = \rho_{l=0}$, while $\rho_{\mathrm{out}} = \rho_J$ for $J \in \{0, 1, 2\}$, which are 1, 2, and 2 dimensional. Each column is an element $K_{j(j'i't')sr}$ of the basis, while each row is a different output channel. Since $\rho_0 \otimes \rho_J = \rho_J$, $j = J$; for circular harmonics, $i' = 1$. We require $G = \mathrm{C}_4$ equivariance, so a frequency $j' = 4$ filter is invariant to $G$; analogously, the output of frequency $j' = |j + 4k|$ filters ($k \in \mathbb{Z}$) transforms like the output of a frequency $j$ one.

## 3 IMPLEMENTATION DETAILS

Given the theoretical results described in the previous sections, we now describe the steps and inputs required to build new $G$-steerable kernels. See Algorithm 1 for a summary. Theorem 2.1 relies on: *a)* the irreps $\widehat{G} = \{\rho_j\}_j$, *b)* a $G$-steerable basis $\mathcal{B} = \{Y_{ji}\}_{ji}$ for $L^2(X)$, *c)* the decomposition $\rho_l \otimes \rho_J = [\mathrm{CG}^{lJ}]^T \left( \bigoplus_{j \in \widehat{G}} \bigoplus_{s=1}^{[j(lJ)]} \rho_j \right) \mathrm{CG}^{lJ}$, and *d)* a basis $\{c^j_r\}_r$ for the endomorphism space of $\rho_j$. *Optionally*, $\mathcal{B}$ can be generated via Eq. 5 using *e)* a larger group $G'$ and its irreps $\widehat{G'} = \{\rho_{j'}\}_{j'}$, *f)* a $G'$-steerable basis $\mathcal{B}' = \{Y_{j'i'}\}_{j'i'}$ and *g)* the decomposition $\mathrm{Res}^{G'}_{G} \rho_{j'} = [\mathrm{ID}^{j'}]^T \left( \bigoplus_{j \in \widehat{G}} \bigoplus_t \rho_j \right) \mathrm{ID}^{j'}$.

The irreps of $G$ in *a)* are the main requirement: they allow us to build arbitrary representations (via

---

[6] For $\theta \in [0, 2\pi)$, $\rho_0 : \theta \mapsto 1$ is the *trivial* irrep; if $k > 0$, $\rho_k$ maps $\theta$ to the $2 \times 2$ rotation matrix of $k \cdot \theta$ radians.
[7] One can verify that any linear combination of these two matrices commutes with any $2 \times 2$ rotation matrix.

---

**Algorithm 1** Generate $G$-Steerable basis on space $X$

---

**Require:** $\rho_{\text{in}} = \rho_l$ and $\rho_{\text{out}} = \rho_J$, $G'$-steerable basis $\mathcal{B}' = \{Y_{j'i'}\}_{j'i'}$, $\widehat{G} = \{\rho_j\}_j$ and $\widehat{G'} = \{\rho_{j'}\}_{j'}$

 1: $\{c_r^j\}_r \leftarrow$ basis for endomorphism space of $\rho_j$, for all $\rho_j$           ▷ Appendix C
 2: $\text{CG}^{lJ}, \{[j(lJ)]\}_j \leftarrow \text{decompose}(\rho_l \otimes \rho_J)$           ▷ Appendix E
 3: **for all** $Y_{j'i'} \in \mathcal{B}'$ **do**
 4:      $\text{ID}^{j'}, \{[jj']\}_j \leftarrow \text{decompose}(\text{Res}_G^{G'} \rho_{j'})$           ▷ Appendix E
 5:      **for all** $j \in \hat{G} : [jj'] > 0, s \le [j(lJ)], t \le [jj'], c_r^j \in \{c_r^j\}_r$ **do**
 6:          $Y_{j(j'i't)}(x) \leftarrow \text{ID}_t^{jj'} \cdot Y_{j'i'}(x)$           ▷ Equation 5
 7:          $K_{j(j'i't)sr}(x) \leftarrow \text{unvec}\left([\text{CG}_s^{j(lJ)}]^T \cdot c_r^j \cdot Y_{j(j'i't)}(x)\right)$           ▷ Theorem 2.1
 8:          **yield** $K_{j(j'i't)sr}$

---

direct sum) and reduce the kernel constraint to the form in Sec. 2.2 by decomposing the input and output representations. If $\mathcal{B}$ in $b$) is unknown, one can rely on Eq. 5. A convenient choice for $G'$ in $e$) is a group whose irreps $\widehat{G'}$, together with a steerable basis $\mathcal{B}'$ as in $f$), are known. $G' = \text{O}(n)$ is always possible if $X = \mathbb{R}^n$, and $\mathcal{B}'$ is built via polar decomposition of $\mathbb{R}^n$ and by combining hyper-spherical harmonics with a radial basis. Like Worrall et al. (2017); Weiler et al. (2018b;a), we use Gaussian radial profiles[8]. In Sec. 5, this choice of $G'$ allows easily experimenting with multiple subgroups $G \le \text{O}(3)$, without the overhead of identifying each $G$'s orbits or designing ad-hoc $G$-steerable bases. Note also that *band-limiting* is achieved by modifying $\mathcal{B}'$ and is *independent* from $G$. Other $G'$'s orbits can also be used, combining harmonic bases (see Appendix D) with a Gaussian kernel.

The irreps decomposition of $\rho_l \otimes \rho_J$ in $c$) provides the multiplicity $[j(lJ)]$ of each irrep $\rho_j$ and the matrices $\text{CG}_s^{j(lJ)}$. Similarly, the irreps decomposition of $\text{Res}_G^{G'} \rho_{j'}$ in $g$) provides the multiplicity of each $G$-irrep $\rho_j$ in each $G'$-irrep $\rho_{j'}$ and the projection matrices $\text{ID}_t^{jj'}$. Knowing these decompositions a-priori for any $l, J$ and $j'$ is generally difficult, but they can be easily computed numerically; see Appendix E. Finally, $d$) requires a basis $\{c_r^j\}_r$ for the endomorphism space of each $\rho_j \in \widehat{G}$. This can be computed numerically, but is often unnecessary; see Appendix C. In summary, if $X = \mathbb{R}^n$ and $G' = \text{O}(n)$, implementing new $G$-steerable CNNs only requires knowing the irreps $\widehat{G}$: by knowing the action of $G$ on $\mathbb{R}^n$, one implicitly knows its embedding into $\text{O}(n)$ and, therefore, can apply Eq. 5.

**Practical Example and Experiments** In Sec. 5.3, we implement a variety of subgroups of the isometries of $\mathbb{R}^3$ (see Appendix G). To do so, we choose $G' = \text{O}(3)$ and $\mathcal{B}'$ is built by combining spherical harmonics with a Gaussian radial profile. Since irreps are used to define the types of the feature fields, we need the irreps of $\text{SO}(3)$, $\text{O}(2)$, $\text{SO}(2)$, $\text{C}_N$, I and O. Direct product groups such as $\text{Inv} \times \text{SO}(2)$ are built as in Appendix F. Since there are generally multiple subgroups $G < \text{O}(3)$ isomorphic to the same abstract group (e.g. see $\text{O}(2)$ in Tab. 1), for each $G$, we explicitly define its isomorphism with an abstract group; this enables the automatic restriction of $G'$'s irreps and the numerical computation of the $\text{ID}_t^{jj'}$ matrices. The matrices $\text{CG}_s^{j(lJ)}$ are also computed numerically.

**Limitations** Our choice of $\mathcal{B}'$ implies we only parameterize filters supported on a compact ball, which is slightly more restrictive than the usual parameterization of filters with support on a cube. For a fixed kernel size, an ad-hoc implementation of $G$-steerable filters can exploit larger initial basis, potentially leading to some performance gain (e.g. $\text{C}_4$ allows filters supported on a square rather than a disk). However, if an ad-hoc implementation of $G' < \text{O}(n)$ steerable kernels is available, our method can be used to construct $G \le G'$ steerable spaces by using the $G'$-steerable basis rather than the $\text{O}(n)$-steerable one. Comparing the effect of different $\mathcal{B}'$ on the performance of a $G$-steerable CNN is beyond the scope of this paper, which instead focuses on building a method suitable for any $G$.

## 4 RELATED WORKS

Similar in spirit to our work is the e2cnn library (Weiler & Cesa, 2019), although limited to $X = \mathbb{R}^2$ and $G \le \text{O}(2)$. $G' = \text{O}(2)$ recovers their solutions. Geiger et al. (2020) present a library implementing general 3D steerable CNNs, but limited to the choices $G = \text{SO}(3)$ or $\text{O}(3)$. In comparison, we currently support both 2D and 3D convolution and any compact group $G \le \text{O}(3)$ (including discrete

---

[8]Other radial profiles are suitable as well without any substantial difference in our theory. The Gaussian profile is chosen mostly for presentation and implementation convenience.

and planar subgroups acting on $\mathbb{R}^3$), and other spaces $X$ can be potentially integrated. Finzi et al. (2021) propose a numerical method to parameterize (finite dimensional) MLPs that are equivariant to arbitrary *matrix groups*. Their numerical method to compute a basis of equivariant linear maps resembles our irreps decomposition method; see Appendix E. Bekkers (2020) and Finzi et al. (2020) implement group convolution on Lie groups using a finite number of samples from the continuous group. This is similar to our SO(3) or O(3) architectures with pointwise non-linearities; see Sec. 5.1. However, since our features and filters are explicitly parametrized on a band-limited space, we can adapt the sampling set to control the equivariance error caused by this approximation.

Aronsson (2021) previously discussed bandlimited convolution operators in equivariant CNNs. Jenner & Weiler (2022) generalize Euclidean steerable CNNs to partial differential operators. (Cohen et al., 2019b; Kondor & Trivedi, 2018; Bekkers, 2020) define steerable CNNs on homogeneous spaces. $G$-steerable kernels are necessary to implement gauge equivariant CNNs on general manifolds (Weiler et al., 2021; Cohen et al., 2019a; Kicanaoglu et al., 2019; Haan et al., 2021). Li et al. (2021) use transformed filters to parameterize steerable kernels. Previously, Mallat (2012); Oyallon & Mallat (2015); Sifre & Mallat (2013) described similar architectures based on scattering. Brandstetter et al. (2021) use a non-linear parameterization of steerable convolution in a geometric graph.

## 5 INSTANTIATION OF THE METHOD AND EXPERIMENTS

We demonstrate the generality of our method by instantiating steerable CNNs for different groups $G$. We will mostly focus on $\mathbb{R}^3$ and its isometries; Appendix G briefly summarizes the subgroups used. Sec. 5.1 discusses existing and new design choices for 3D steerable CNNs. We then compare these designs in Sec. 5.3. Sec. 5.2 numerically compares our steerable basis with alternative bases.

### 5.1 DESIGN CHOICES OF 3D STEERABLE CNNS

A $G$-steerable CNN design involves a choice of feature field *types*, i.e. $G$ representations acting on the channels of the features, see Sec. 2.1. The chosen types constrain the non-linear operations permitted.

Many works in the literature focus on designs which achieve perfect SO(3) equivariance (Weiler et al., 2018b; Thomas et al., 2018; Anderson et al., 2019) . To ensure this, a common choice is the use of *irreps* feature types and *gated non-linearities*. This operation combines a feature field $f_\rho(x) \in \mathbb{R}^{d_\rho}$ of type $\rho$ with a *gate* $f_0(x) \in \mathbb{R}$ as $f_\rho(x) \mapsto \sigma(f_0(x))f_\rho(x)$, where $\sigma$ is a sigmoid function and $f_0$ is feature field of trivial type. This non-linearity is equivariant to any $G$ but Weiler & Cesa (2019) found them to perform worse than point-wise non-linearities like those used in GCNNs. Another popular choice is using the *tensor product* of two feature fields as a non-linear operation (Kondor et al., 2018), i.e. $\left(f_{\rho_i}(x), f_{\rho_j}(x)\right) \mapsto f_{\rho_i}(x) \otimes f_{\rho_j}(x)$. However, this non-linearity is quadratic and turns the network into a polynomial function of the input, which can lead to training difficulties (Anderson et al., 2019). Additionally, computing the tensor product $\otimes$ has computational and memory cost $O(d_{\rho_i} d_{\rho_j})$; it is common to project the output features to a smaller feature field through a learnable linear map. A different approach is used in Worrall & Brostow (2018); Winkels & Cohen (2018), which use the octahedral group O on voxelized data. The finiteness of the group allows for a simple GCNN (Cohen & Welling, 2016a) architecture, corresponding to the choice of the *regular representation* $\rho_{\text{reg}}$ as the feature type and standard pointwise non-linearities (e.g. ReLU). The representation $\rho_{\text{reg}}$ of a group $G$ acts on feature vectors in $\mathbb{R}^{|G|}$ by permutations and a vector in this space can be interpreted as a scalar function over $G$; see Weiler & Cesa (2019) for an intuitive description.

Because SO(3) (and O(3)) has only a limited number of discrete subgroups, we also consider discretizations without group structure. We interpret a feature vector $f(x) \in \mathbb{R}^B$ as the coefficients parameterizing a band-limited function over $G$ in Fourier space; see Appendix D. To apply a pointwise non-linearity $\sigma : \mathbb{R} \to \mathbb{R}$ (e.g. ELU) on $f(x)$, we *i)* sample the parameterized function on a finite set $\mathcal{G} \subset G$, *ii)* apply $\sigma$ and, then, *iii)* recover the coefficients $f'(x)$ of the output via a Fourier transform. If $\sigma$ is smooth, the output signal is still approximatively band-limited but can contain higher frequencies than the input $f(x)$. This implies that a larger set $\mathcal{G}$ is required to reconstruct $f'(x)$ than for $f(x)$. If a fixed budget of channels (i.e. number of samples $|\mathcal{G}|$) is available for a non-linear module, the input field $f(x)$ must be strongly bandlimited. While this can reduce the expressiveness of the model's features, it also implies that linear layers are computationally cheaper as they operate on smaller number of channels. A similar result was leveraged in Cheng et al. (2019) for $G = \text{SO}(2)$. This *regular non-linearity* was also used in Kicanaoglu et al. (2019) for $G = \text{SO}(2)$. The same approach can be generalized to *quotient fields*, i.e. feature fields which parameterize functions over

Table 1: Rotated ModelNet10 (O(3) symmetry). ∗ indicates wider models to fix the computational cost.

| $G$ | Description | Accuracy |
|---|---|---|
| $\{e\}$ | Conventional CNN | $82.5 \pm 1.4$ |
| $\mathrm{SO}(2)$ | Axial Symmetry | $86.9 \pm 1.9$ |
| $\mathrm{SO}(2) \rtimes \mathrm{F} \cong \mathrm{O}(2)$ | Dihedral Symmetry | $87.5 \pm 0.7$ |
| $\mathrm{SO}(2) \rtimes \mathrm{M} \cong \mathrm{O}(2)$ | Conical Symmetry | $88.5 \pm 0.8$ |
| $\mathrm{Inv} \times \mathrm{SO}(2)$ | Cylindrical Symmetry | $86.8 \pm 0.7$ |
| $\mathrm{Inv} \times \mathrm{SO}(2) \rtimes \mathrm{F}$ | Full Cylindrical Symmetry | $87.0 \pm 1.0$ |
| O | Octahedral Symmetry (Winkels & Cohen, 2018) | $89.7 \pm 0.6$ |
| I | Icosahedral Symmetry | $90.0 \pm 0.6$ |
| I | Icosahedral Symmetry (finite orbits basis) | $88.2 \pm 1.0$ |
| $\mathrm{SO}(3)$ | Chiral (Tensor product) (Anderson et al., 2019) | $86.3 \pm 1.0$ |
| $\mathrm{SO}(3)$ | Chiral (Gated non-linearity)(Weiler et al., 2018b) | $88.8 \pm 1.2$ |
| $\mathrm{SO}(3)$ | Chiral (Regular, $|\mathcal{G}| = 96$) | $89.1 \pm 1.2$ |
| $\mathrm{SO}(3)$ | Chiral (Regular, $|\mathcal{G}| = 192$)* | $89.4 \pm 1.4$ |
| $\mathrm{SO}(3)$ | Chiral (Quotient $\mathcal{S}^2 = \mathrm{SO}(3)/\mathrm{SO}(2)$, $|\mathcal{X}| = 30$) | $89.5 \pm 1.0$ |
| $\mathrm{O}(3)$ | Achiral (Regular, $|\mathcal{G}| = 120$) | $89.2 \pm 0.6$ |
| $\mathrm{O}(3)$ | Achiral (Regular, $|\mathcal{G}| = 144$)* | $89.4 \pm 0.7$ |
| $\mathrm{O}(3)$ | Achiral (Quotient $\mathrm{Inv} \times \mathcal{S}^2 = \mathrm{O}(3)/\mathrm{SO}(2)$, $|\mathcal{X}| = 60$) | $88.6 \pm 0.9$ |

a quotient space $X = G/H$ (e.g. $\mathcal{S}^2 = \mathrm{SO}(3)/\mathrm{SO}(2)$). See Appendix H.2 for more details on point-wise non-linearities and their computational benefit.

## 5.2 NUMERICAL COMPARISON OF STEERABLE BASES

In this section, we compare our basis with one built according to Lang & Weiler (2020). We consider the example of Icosahedral I steerable 3D convolution. Our basis is composed by a band-limited set of spherical harmonics per spherical shell, combined as in Theorem 2.1. The two baselines use respectively the center of the 20 faces of the dodecahedron and the 12 faces of the icosahedron embedded in each shell as orbits and parameterize the kernel along each orbit directly using (Lang & Weiler, 2020). Fig. 5 shows the histograms of the equivariance errors of the three bases with respect to the Icosahedral group; each point represents an element of the basis and the errors are averaged all transformations; see Appendix H.1. Our basis shows significantly higher stability. Finally, we study the effect of these bases on a model's performance. In Tab. 1 we compare two I-equivariant models: one using our harmonic basis and one using the 20 faces of the Icosahedron as orbit. Our anti-aliased basis leads to a significant improvement in accuracy.

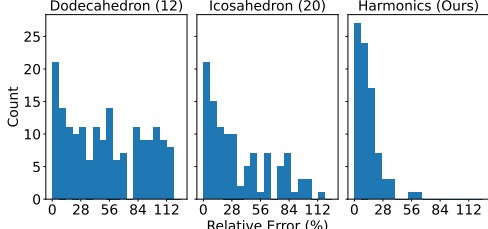

Figure 5: Histograms of relative equivariance errors of three different Icosahedral I-steerable bases. The first two parameterize two different orbits of I while the third uses our basis based on spherical harmonics.

## 5.3 EXPERIMENTS

To emphasize our method is not limited to $\mathbb{R}^3$, we include a simple experiment with 2D images. In the rest of the section, we compare different model designs on two *volumetric* datasets: ModelNet10 (Wu et al., 2015) (and a rotated version of it) and LBA (Townshend et al., 2020). For each model, we run a simple search over hyperparameters and minor variants of the designs described in Sec. 5.1 using validation accuracy, to ensure a fair comparison. Unless specified, within a task, all models approximatively share the same width. In particular, in all tasks, the models which more closely match the symmetries of the data perform the best. See Appendix H for more details on the experiments. The equivariance groups used (column $G$ in the tables) are described in Appendix G.

**Rotated MNIST** As a simple 2D experiment, we train a conventional CNN and a $G = \mathrm{C}_8$ equivariant CNN on rotated MNIST. The steerable kernel bases are similar to those in Weiler & Cesa (2019) and we use *regular* field types with pointwise ELU. The conventional model achieves $96.3 \pm 0.1$ test accuracy while the $\mathrm{C}_8$ equivariant one $96.7 \pm 0.4$.

**Rotated ModelNet10** We generate a voxelized version of ModelNet10 with resolution 33px by sampling 3 random $\mathrm{SO}(3)$ rotations for each object. During training, we use rotations from O for

augmentation. In Tab. 1, we compare different equivariance groups and different equivariant designs. It includes our implementation of some relevant related works; see Sec. 5.1. We first highlight the large margin between all equivariant networks and the conventional CNN. SO(3) and O(3) models achieve better performance than the other continuous subgroups. The best results are achieved by using the discrete groups I and O or the SO(3) model with spherical *quotient* feature types. Recall also that the models with quotient or regular non-linearities are computationally cheaper; see Sec.5.1.

**ModelNet10** To experiment with data with only axial and mirror symmetries, we generate a version of ModelNet10 by sampling the random rotations only around the $Z$ axis. $\frac{\pi}{2}$ rotations and reflection augmentation in the $XY$ plane is used for training but no rotation averaging is used for testing. The final accuracies are in Tab. 2. The models equivariant to 3D rotations are outperformed by the SO(2) and O(2) models, which better match the data symmetries.

Table 2: ModelNet10 (O(2) symmetry)

| $G$ | Description | Accuracy |
|---|---|---|
| $\{e\}$ | Conventional CNN | $91.2 \pm 0.5$ |
| SO(2) | Azimuthal Symmetry | $91.9 \pm 0.8$ |
| SO(3) | Chiral (Regular, $\lvert\mathcal{G}\rvert = 72$) | $89.8 \pm 0.6$ |
| O(2) | Full Azimuthal Symmetry | $92.3 \pm 0.4$ |
| O(3) | Achiral (Regular, $\lvert\mathcal{G}\rvert = 120$) | $89.9 \pm 1.0$ |
| $C_2 \rtimes F$ | Klein Group (dihedral symmetry) | $91.0 \pm 0.6$ |
| | VOXNet (Maturana & Scherer, 2015) | 92.0 |
| $C_2 \rtimes F$ | Klein Group (Worrall & Brostow, 2018) | 94.2 |

**Ligand Binding Affinity (LBA) regression** We also evaluate our models on a regression task using the LBA dataset (Townshend et al., 2020), containing protein-ligand complexes from the PDBBind database (Wang et al., 2004; Liu et al., 2014). In Townshend et al. (2020), an SO(3)-equivariant model with *tensor product* activations performed the best, despite its cost prevented training on the full dataset. We propose a volumetric SO(3) network with a computational cost equivalent to a conventional CNN by leveraging the benefit of *regular non-linearities*. Tab. 3 reports our results.

**Discussion** We again emphasize that the purpose of our experiments is mostly demonstrating the generality and flexibility of our program, but we do not necessarily envision applications of all subgroups $G < O(3)$ considered. Still, some subgroups are practically relevant; indeed, models with azimuthal symmetry perform the best in Table 2 where the data lacks full rotational symmetry. Here, enforcing SO(3) equivariance results in over-constrained models which are not sufficiently expressive and are outperformed even by the conventional CNN. Overall, we observe models whose symmetry matches the data tend to perform the best. Some discrete subgroups, e.g. the platonic ones, constitute a special exception: they approximate the full rotational symmetry but are less restrictive than a fully rotation equivariant model, sometimes better matching the approximate symmetry of voxelized data. Finally, we only found minor differences among the various SO(3) models, although regular and quotient non-linearities stood out for their reduced computational cost (see also Appendix H.7).

## 6 CONCLUSIONS

In this work, we gave a general characterization of $G$-steerable kernel spaces over an arbitrary space $X$ and proposed a general strategy to automatically parameterize them based only on $G$'s irreps. This enabled us to implement steerable CNNs equivariant to a variety of new groups $G \leq O(3)$, which we compare in an exploratory study in our experiments. Finally, we believe that Theorem 2.1 suggests a new theoretical perspective on the construction of equivariant CNNs, reducing the problem to that of devising a suitable starting basis $\mathcal{B}$, regardless of the choice of feature types in the model.

Table 3: Ligand Binding Affinity dataset. $*$ indicates wider models to fix the computational cost.

| $G$ | Description | RMSD | Pearson | Spearman |
|---|---|---|---|---|
| $\{e\}$ | Conventional CNN | $1.419 \pm 0.047$ | $0.575 \pm 0.022$ | $0.569 \pm 0.021$ |
| O | Octahedral Symmetry | $1.417 \pm 0.032$ | $0.589 \pm 0.010$ | $0.581 \pm 0.011$ |
| I | Icosahedral Symmetry | $1.432 \pm 0.020$ | $0.569 \pm 0.023$ | $0.559 \pm 0.023$ |
| SO(3) | Chiral (Regular, $\lvert\mathcal{G}\rvert = 72$) | $1.397 \pm 0.039$ | $0.580 \pm 0.021$ | $0.573 \pm 0.022$ |
| SO(3) | Chiral (Regular, $\lvert\mathcal{G}\rvert = 60$)* | $1.380 \pm 0.033$ | $0.588 \pm 0.015$ | $0.578 \pm 0.018$ |
| SO(3) | Chiral (Quotient $\mathcal{S}^2$, $\lvert\mathcal{X}\rvert = 24$) | $1.405 \pm 0.028$ | $0.582 \pm 0.018$ | $0.576 \pm 0.016$ |
| SO(3) | Chiral (Quotient $\mathcal{S}^2$, $\lvert\mathcal{X}\rvert = 30$)* | $1.385 \pm 0.032$ | $0.587 \pm 0.019$ | $0.576 \pm 0.019$ |
| $\{e\}$ | 3DCNN (Townshend et al., 2020) | 1.520 | 0.558 | 0.556 |
| $\{e\}$ | Graph-NN (GNN) (Townshend et al., 2020) | 1.936 | 0.581 | 0.647 |
| SO(3) | E-GNN (Townshend et al., 2020) | 1.429 | 0.541 | 0.532 |

ACKNOWLEDGMENTS

Gabriele would like to thank Arash Behboodi for fruitful discussions on the approximation of point-wise non-linearities with finite samples.

## REPRODUCIBILITY STATEMENT

To ensure reproducibility, we include a dedicated section in Appendix H to describe our experiments in more details. Moreover, we implemented all the methods described in this work in a Python library which can be found at `github.com/QUVA-Lab/escnn`. We also plan to publish the code used to run our experiments.

## POSSIBLE NEGATIVE SOCIETAL IMPACTS

Since this work is very foundational, many different negative societal impacts are conceivable. In particular, any misuse which is eased by improved computer vision systems can profit from this and similar work. Of course, we cannot discuss them all.

Nevertheless, we want to highlight one specific misuse which seems likely to occur for the field of equivariant deep learning in the future – namely, the adoption in lethal autonomous weapons systems (LAWS). We can easily think of drones using spherical vision which would profit from equivariant CNNs. In particular, since they have a fixed vertical orientation, but arbitrary horizontal orientation, they could profit from spherical CNNs using azymuthal symmetries.

It is important to realize that the mitigation of such misuses cannot happen at the level of individual research projects. Instead, what is needed is a global effort to mitigate the misuse of AI systems for LAWS in general. Fortunately, the broader AI community agrees that AI should not be used in LAWS, and that international regulations are necessary to prevent their use in the future aut (2015). We hope that efforts in this direction are continued.

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

# APPENDIX

In the Appendix, we provide a more detailed description and the proof of the theoretical results mentioned in the main paper. We first give a brief overview of the *representation theory* of compact groups and introduce the notation we will use throughout the next chapters in Section A. In Section B, we state more precisely and prove the main result from Sec. 2.2, which we used to parametrize the steerable filters. In particular, we state both a complex and a real version of Theorem 2.1. In section B.4, we discuss two examples of kernel bases. Sections C, D, E and F provide additional theoretical results and computational methods which can ease the practical implementation of steerable CNNs, but are not necessary for understanding our main results.

Because the implementation of steerable CNNs is necessarily using real numbers, its design should rely on the *real* representation theory of the group considered. In Section C, we review the theory of *real* irreducible representations and derive some useful results to deal with these representations in the implementation. Notably, we show that, for a compact group $G$, any real irrep of $G$ can always be expressed in a particular basis which allows one to recover a harmonic basis for square-integrable functions over $G$ from the matrix coefficients of the irreps. The results shown in Section C will become useful in Section D, E and F.

Regular, quotient and induced features types encode scalar or vector functions over a homogeneous space of $G$ in terms of a harmonic basis. Moreover, harmonic bases provide a simple way to generate the $G$-steerable bases required to define the kernel basis in Sec. 2.2: by leveraging the fact that orbits of $G$ in $X$ are isomorphic to homogeneous spaces, the harmonic bases for the orbits of $G$ in $X$ can be combined with a Gaussian kernel to generate $G$-steerable bases over $X$. In Section D, we derive a method to compute the (either real or complex) harmonic basis for any homogeneous space of $G$ from the harmonic basis of functions over $G$. The method described in Section D turns out to be more general and also supports vectors fields over homogeneous spaces, i.e. general induced representations of $G$.

Since the kernel parameterization in Sec. 2.2 relies on the irreps decomposition of the input and output representations, of the tensor product of irreps (to compute the Clebsh-Gordan coefficients CG) and of the restricted irreps of $G'$ (to compute the ID coefficients), the ability to decompose an arbitrary representation is essential to apply our method in a fully general setting. Therefore, in Section E, we describe a numerical method to perform this decomposition for *real* representations.

As explained in Sec. 3, the main (and, often, only) requirement to apply our method to a new group $G$ is the knowledge of its irreps $\widehat{G}$. A simple operation which can be used to generate new groups is the *direct product* $\times$ of two groups. Examples of direct product groups include many subgroups of $\mathrm{O}(3)$ that we used in Sec. 5. In Section F, we describe a method to derive the *real* irrep of a direct product group $G = A \times B$ from the *real* irreps of the two subgroups $A$ and $B$.

Section H includes additional details on the experimental results reported in Sec. 5. In particular, we describe in more details the different equivariant designs, the architectures used, the datasets and the training and testing configurations.

## A    Preliminaries on the Representation Theory of Compact Groups

In this section, we collect many notions and concepts which we need throughout the appendix for the development of the theory. This section assumes some familiarity with the representation theory of compact groups and is meant as a reminder. For readers with no background in representation theory, we can recommend the appendix of Lang & Weiler (2020).

We will use $\mathbb{K}$ to denote any of the two fields $\mathbb{R}$ or $\mathbb{C}$, i.e., we develop the theory in such a way that it covers both real and complex representations. Whenever we deviate from this assumption, we will say so explicitly.

Throughout, let furthermore $G$ be any compact group.

### A.1    Basics of (Unitary and Irreducible) Representations

*Linear representations* are (in a certain sense continuous) group homomorphisms $\rho : G \to \mathrm{GL}(V)$ for some topological $\mathbb{K}$-vector space $V$. In our applications, $V$ always carries a scalar product $\langle \cdot | \cdot \rangle$ which induces its topology, and which we always assume to be conjugate linear in the first component,

and linear in the second component. The pair $(V, \langle \cdot | \cdot \rangle)$ is assumed to be a Hilbert space, meaning that the scalar product makes $V$ a complete metric space. Note that, as is typical in a physics context, we freely make use of the Bra-Ket convention, and thus for example view $\langle v |$ as a linear functional which maps $|w\rangle := w$ to $\langle v \mid w \rangle$. By the Riesz representation theorem, there are no other continuous linear functionals.

*Unitary representations* are linear representations $\rho : G \to \mathrm{U}(V) \subseteq \mathrm{GL}(V)$, where $\mathrm{U}(V)$ is the *unitary group* of $V$. This is the group of *unitary transformations* on $V$, i.e., linear functions on $V$ which preserve the scalar product, and thus distances. Whenever we explicitly assume $\mathbb{K} = \mathbb{R}$, we will write $\mathrm{O}(V)$ instead of $\mathrm{U}(V)$, which is the group of *orthogonal transformations* on $V$, and the representations are then called *orthogonal* instead of *unitary*.[9]

We will denote the set of isomorphism classes of all *irreducible* unitary representations (irreps) of $G$ as $\widehat{G}$. Thereby, irreducible representations are representations which cannot be decomposed nontrivially into subrepresentations. We usually write indices $j, l, J \in \widehat{G}$ to indicate such isomorphism classes, and then, e.g., $\rho_j : G \to \mathrm{U}(V_j)$ to denote a representative of such an isomorphism class. We set $d_j := \dim V_j$ as the dimension of the representation spaceDof $V_j$. Whenever we consider only one irrep, we may also, by abuse of notation, write $\rho \in \widehat{G}$ to indicate (a representative of an isomorphism class) of an irrep of $G$.

We now collect definitions which we use throughout.

**Definition A.1** (Homomorphisms/Intertwiners). Let $\rho_V : G \to \mathrm{GL}(V)$ and $\rho_W : G \to \mathrm{GL}(W)$ be two linear representations. The space of *homomorphisms*, also called *intertwiners*, $\mathrm{Hom}_{G,\mathbb{K}}(V, W)$, consists of all linear, continuous functions $f : V \to W$ which commute with the two representations, i.e., $f \circ \rho_V(g) = \rho_W(g) \circ f$ for all $g \in G$. We write for this space also $\mathrm{Hom}_{G,\mathbb{K}}(\rho_V, \rho_W)$, depending on whether we want to emphasize the representation spaces or the representations.

**Definition A.2** (Equivalences and Isomorphisms). Let $\rho_V : G \to \mathrm{GL}(V)$ and $\rho_W : G \to \mathrm{GL}(W)$ be two linear representations. An intertwiner $f : V \to W$ is called an *equivalence* if there is an intertwiner $g : W \to V$ such that $g \circ f = \mathrm{id}_V$ and $f \circ g = \mathrm{id}_W$. $\rho_V$ and $\rho_W$, or also $V$ and $W$, are then said to be *equivalent*.

If $\rho_V$ and $\rho_W$ are even unitary representations, and $f$ and $g$ additionally are unitary transformations, i.e., preserve the scalar product and thus distances, then $V$ and $W$ or $\rho_V$ and $\rho_W$ are called *isomorphic*, and $f$ and $g$ *isomorphisms*. We write this as $\rho_V \cong \rho_W$ or $V \cong W$, depending on whether we want to emphasize the representations or the representation spaces.

**Definition A.3** (Endomorphisms, Endomorphism Basis). Let $\rho : G \to \mathrm{GL}(V)$ be a linear representation. The space of *endomorphisms* $\mathrm{End}_{G,\mathbb{K}}(V)$ consists of all linear, continuous maps $c : V \to V$ that commute with $\rho$, i.e., $c \circ \rho(g) = \rho(g) \circ c$ for all $g \in G$.

For $\rho = \rho_j$ being an irrep, this space is finite-dimensional and Das a basis (the *endomorphism basis*) $(c_r^j)_{r=1}^{E_j}$ with $E_j = \dim(\mathrm{End}_{G,\mathbb{K}}(V_j))$.

Note that $\mathrm{End}_{G,\mathbb{K}}(V_j) = \mathrm{Hom}_{G,\mathbb{K}}(V_j, V_j)$.

By Schur's Lemma, the endomorphism spaces of complex irreps are always 1-dimensional, generated by the identity. For real representations, they can be 1, 2, or 4-dimensional, and the irrep is then correspondingly called of *real type, complex type, or quaternionic type*. See Supplementary C for more details on real representations.

### A.2 Homogeneous Spaces, Square-Integrable Functions, and the Peter-Weyl Theorem

Recall that a homogeneous space of the compact group $G$ is a topological (Hausdorff) space $X$ together with a transitive continuous action $. : G \times X \to X, (g, x) \mapsto g.x = gx$.

As is commonly known, $G$ carries an inversion-invariant and left-right invariant so-called *Haar-measure* $\mu$, which can furthermore be assumed to be normalized: $\mu(G) = 1$. This Haar measure also

---

[9]But note that in the general case of unspecified field $\mathbb{K}$, we always speak of unitary representations, by which we just mean "orthogonal" in the case $\mathbb{K} = \mathbb{R}$

induces a corresponding $G$-action invariant measure $\nu$ on $X$, which we also call *Haar-measure* and assume to be normalized: $\nu(X) = 1$.

The normalization makes both Haar-measures the unique measures which are defined on the whole Borel Sigma algebras of these spaces and satisfying the mentioned invariance properties. To reduce clutter, we will mostly not mention the used measure in integral-formulas, but we view it as understood that all integrals are with respect to these unique Haar measures.

Fix from now on a homogeneous space $X$ of $G$.

**Definition A.4** ((The Space of) Square-Integrable Functions). A *square-integrable function* $f : X \to \mathbb{K}$ is a measurable function with the property that the square of its absolute value is integrable: $\int_X |f(x)|^2 dx < \infty$.

We denote the space of all such square-integrable functions by $L^2_{\mathbb{K}}(X)$. It carries a scalar product given by $\langle f \mid g \rangle = \int_X \overline{f(x)} g(x) dx$, making it a Hilbert space. It also carries a unitary representation of $G$ given by $\lambda : G \to \mathrm{U}(L^2_{\mathbb{K}}(X))$, $(\lambda(g)f)(x) := f(g^{-1}x)$.

**Definition A.5** (Regular Representation). In the case that $X = G$ is the group itself, we call $L^2_{\mathbb{K}}(G)$, together with the action $[\lambda(g)f](g') := f(g^{-1}g')$, the *regular representation* of $G$.

**Theorem A.6** (Peter-Weyl Theorem). *There is a decomposition*

$$L^2_{\mathbb{K}}(X) = \widehat{\bigoplus_{j \in \widehat{G}}} \bigoplus_{i=1}^{m_j} V_{ji}$$

*into irreducible subrepresentations. Thereby, $V_{ji}$ is isomorphic to the representative irreducible representation $V_j$ and generated by orthonormal harmonic basis functions $Y_{ji}^m : X \to \mathbb{K}$. We have $m_j \leq d_j := \dim V_j$. If $\mathrm{D} = \mathbb{C}$ and $X = G$, then $m_j = d_j$.*

Note that the hat $\widehat{(\cdot)}$ denotes a *topological closure*. It is a technicality which the reader can mostly ignore without problems.

**Proposition A.7** (Peter-Weyl Theorem: An Explicit Basis for Complex-Valued Square-Integrable Functions on a Compact Group). *In the complex field $\mathbb{K} = \mathbb{C}$, the orthonormal harmonic basis for $L^2_{\mathbb{C}}(G)$ is given explicitly by:*

$$Y_{ji}^m(g) = \sqrt{d_j} \langle \rho_j(g)\boldsymbol{e}_i \mid \boldsymbol{e}_m \rangle = \sqrt{d_j} \, \overline{[\rho_j(g)]_{mi}} = \sqrt{d_j} \, [\rho_j(g^{-1})]_{im}$$

*In other words, the matrix coefficients of the complex irreps of $G$ form an orthogonal basis for $L^2_{\mathbb{C}}(G)$.*

**Definition A.8** (Harmonic Projections). We denote the orthogonal projection from $L^2_{\mathbb{K}}(X)$ to the $i$'th copy of $V_j$ by $p_{ji} : L^2_{\mathbb{K}}(X) \to V_j$. On the harmonic basis, it is given by

$$p_{ji}\big(Y_{\widetilde{j}\widetilde{i}}^{\widetilde{m}}\big) = \delta_{j\widetilde{j}} \delta_{i\widetilde{i}} \cdot |j\widetilde{m}\rangle,$$

which just follows from the fact that the harmonic basis functions form an orthonormal basis of $L^2_{\mathbb{K}}(X)$, which is then mapped to the corresponding orthonormal basis of $V_j$, the representation space of the representative irrep $\rho_j : G \to \mathrm{U}(V_j)$.

One commonly known corollary of the Peter-Weyl theorem is the following. We make use of it in the definitions of the Clebsch-Gordan coefficients and irrep-decomposition coefficients which we define in the next two subsections:

**Corollary 1.** *Any unitary representation of $G$, even infinite dimensional, can be decomposed into a (topological closure of a) direct sum of irreps of $G$ with a unitary isomorphism, i.e.: for all unitary representations $\pi : G \to \mathrm{U}(V)$ of $G$, there exists a unitary transformation $M : V \to \widehat{\bigoplus}_{i \in I} V_i$, where $\rho_i : G \to \mathrm{U}(V_i)$ are unitary irreducible representations such that for all $g \in G$ the following identity holds:*

$$\pi(g) = M^{-1} \left( \widehat{\bigoplus_{i \in I}} \rho_i(g) \right) M \ .$$

*$I$ is a set indexing irreps in $\widehat{G}$, and the same irrep may occur multiple times. If such an $M$ exists, we also write $V \cong \widehat{\bigoplus}_{i \in I} V_i$.*

That $M$ is a unitary transformation means, as explained earlier, that it preserves the scalar product and thus distances. Consequently, one has $M^{-1} = M^{\dagger}$, i.e., the inverse is equal to the adjoint[10]. In the finite-dimensional case, and working with matrices, one has that $M^{\dagger}$ is given by the conjugate transpose of $M$, and if we additionally consider real-valued representations, then $M^{\dagger} = M^T$ is just the transpose.

## A.3 CONSTRUCTING UNITARY REPRESENTATIONS FROM OTHERS AND THEIR RELATIONS

Now we come to tensor-products and Clebsch-Gordan coefficients. Recall that given two finite-dimensional unitary representations $\rho_V : G \to \mathrm{U}(V)$ and $\rho_W : G \to \mathrm{U}(W)$, one can define a unitary representation $\rho_V \otimes \rho_W : G \to \mathrm{U}(V \otimes W)$ called *tensor product* given by $(\rho_V \otimes \rho_W)(g) = \rho_V(g) \otimes \rho_W(g) : v \otimes w \mapsto [\rho_V(g)](v) \otimes [\rho_W(g)](w)$. Thereby, $V \otimes W$ is the tensor product of the vector spaces $V$ and $W$. If $V$ and $W$ have bases $\{v_1, \ldots, v_n\}$ and $\{w_1, \ldots, w_m\}$, respectively, then the tensor product has a basis $\{v_i \otimes w_j \mid i = 1, \ldots, n, j = 1, \ldots, m\}$. Correspondingly, represented with respect to given bases, the matrix corresponding to $\rho_V(g) \otimes \rho_W(g)$ is the Kronecker product of the matrices corresponding to $\rho_V(g)$ and $\rho_W(g)$. The scalar product on the tensor product is given by $\langle v \otimes w \mid v' \otimes w' \rangle = \langle v \mid v' \rangle \cdot \langle w \mid w' \rangle$. The tensor product $V \otimes W$ has a *universal property* which relates bilinear functions on $V \times W$ to linear functions on $V \otimes W$.

Recall that for an irrep $\rho_j : G \to \mathrm{U}(V_j)$, $d_j$ denotes the dimension of $V_j$.

**Definition A.9** (Clebsch-Gordan Coefficients). Let $\rho_l : G \to \mathrm{U}(V_l)$ and $\rho_J : G \to \mathrm{U}(V_J)$ be irreducible unitary representations of $G$. There is a decomposition

$$V_l \otimes V_J \cong \bigoplus_{j \in \widehat{G}} \bigoplus_{s=1}^{[j(lJ)]} V_j$$

of the tensor product in irreps, where $[j(lJ)]$ is the multiplicity of $V_j$ in $V_l \otimes V_J$. This multiplicity is zero for all but finitely many $j \in \widehat{G}$. We denote the coupling coefficients between basis elements coming from this decomposition by $\langle s, jm|ln; JM \rangle$, where $m \leq d_j$, $n \leq d_l$, $M \leq d_J$ and $s \leq [j(lJ)]$. They are called the *Clebsch-Gordan coefficients*.

*Remark* A.10. The main application of the tensor product will be in the Wigner-Eckart theorem which we state in Section B, where $\rho_j$ will be the irrep corresponding to harmonic basis functions on a homogeneous space, $\rho_l$ will be the input representation, and $\rho_J$ will be the output representation. Compared to the work Lang & Weiler (2020), this means that we will decompose $V_l \otimes V_J$ instead of $V_j \otimes V_l$, which leads to a simpler implementation for real representations, the case we are concerned with, at the cost of the necessity to invoke dual representations in the case that one works with complex representations.

The definition of these coefficients can be made more explicit by invoking the corresponding projections and embeddings, which we will use in the proof of our adapted Wigner-Eckart Theorem. We thereby define $|ln\rangle$, $n \leq d_l$, as the $n$'th basis vector in $V_l$, for a fixed basis, and similarly for $|jm\rangle$ and $|JM\rangle$.

**Definition A.11** (Clebsch-Gordan Projections and Embeddings). We denote the projection from $V_l \otimes V_J$ to the $s$'th copy of $V_j$ by $\mathrm{CG}_s^{j(lJ)} : V_l \otimes V_J \to V_j$. It has as matrix elements the Clebsch-Gordan coefficients for fixed $s, j, l, J$:

$$\langle s, jm \mid ln; JM \rangle = \left\langle jm \middle| \mathrm{CG}_s^{j(lJ)} \left( |ln\rangle \otimes |JM\rangle \right) \right\rangle. \tag{6}$$

Dually, we denote the embedding of the $s$'th copy of $V_j$ into $V_l \otimes V_J$ by $i_s^{j(lJ)} : V_j \to V_l \otimes V_J$. It relates to the Clebsch-Gordan coefficients as follows:

$$\langle s, jm|ln; JM \rangle = \left\langle i_s^{j(lJ)}(|jm\rangle) \middle| ln; JM \right\rangle. \tag{7}$$

Note that this definition of Glebsh-Gordan coefficient is compatible with the one in Sec. 2.2, for real representations. Indeed, in Sec. 2.2, $\mathrm{CG}_s^{j(lJ)}$ denotes the matrix of Clebsch-Gordan coefficients $\langle s, j\widetilde{m}|lm; JM \rangle$ of shape $d_j \times (d_l \cdot d_J)$.

---

[10]The adjoint of a continuous linear operator $f : V \to W$ between Hilbert spaces is defined as the unique continuous linear operator $f^{\dagger} : W \to V$ satisfying $\langle f(v) \mid w \rangle = \langle v \mid f^{\dagger}(w) \rangle$ for all $v \in V$ and $w \in W$.

For a Hilbert space $V$, we set $V^\dagger$ as the space of continuous linear functionals on $V$. As mentioned above, by the Riesz representation theorem these linear functionals are given by all $v^\dagger := \langle v|$ for $v \in V$. $V^\dagger$ is then also a Hilbert space with the scalar product given by $\langle v^\dagger | w^\dagger \rangle = \langle w | v \rangle$.[11] This definition of a dual can be extended to a dual of unitary representations:

**Definition A.12** (Dual Representation). Let $\rho : G \to \mathrm{U}(V)$ be a finite-dimensional unitary representation. Then we can define the dual representation $\rho^\dagger : G \to \mathrm{U}(V^\dagger)$ as follows:

$$[\rho^\dagger(g)](v^\dagger) := v^\dagger \circ \rho(g^{-1}) : V \to \mathbb{K}. \tag{8}$$

If $\rho_l : G \to \mathrm{U}(V_l)$ is a finite-dimensional irreducible representation, then we write $V_{l^\dagger} := (V_l)^\dagger$ and $(\rho_l)^\dagger := \rho_{l^\dagger}$ for a new auxiliary label $l^\dagger$. If $|ln\rangle$, $n = 1, \dots, d_l$, are chosen basis elements of $V_l$, then we choose as the basis of $V_{l^\dagger}$ the vectors $|l^\dagger n\rangle := \langle ln|$, which are the functionals defined by the property $\langle l\tilde{n}|ln\rangle = \delta_{\tilde{n}n}$.

Similarly to how we defined a tensor product representation above, one can also define Hom-representations: Let $\rho_V : G \to \mathrm{U}(V)$ and $\rho_W : G \to \mathrm{U}(W)$ be finite-dimensional unitary representations. Let $\mathrm{Hom}_{\mathbb{K}}(V, W)$ be defined as the vector space of linear, not necessarily equivariant, functions from $V$ to $W$. Choosing bases of $V$ and $W$, this space is isomorphic to $\mathbb{K}^{\dim V \cdot \dim W}$ and thus carries a Euclidean scalar product which makes it a Hilbert space. The Hom-representation is then given by $\rho_{\mathrm{Hom}} : G \to \mathrm{U}(\mathrm{Hom}_{\mathbb{K}}(V, W))$ with $[\rho_{\mathrm{Hom}}(g)](f) := \rho_W(g) \circ f \circ \rho_V(g)^{-1}$. Tensor product representations and Hom-representations are related as follows:

**Proposition A.13.** *Let $\rho_V : G \to \mathrm{U}(V)$ and $\rho_W : G \to \mathrm{U}(W)$ be finite-dimensional unitary representations. The function $\Psi_{\mathrm{HT}} : V^\dagger \otimes W \to \mathrm{Hom}_{\mathbb{K}}(V, W)$ given by*

$$\Psi_{\mathrm{HT}} : v^\dagger \otimes w \mapsto \left(v^\dagger\right)_w, \quad \left(v^\dagger\right)_w (v') := v^\dagger(v') \cdot w \tag{9}$$

*is an isomorphism of unitary representations. The indices HT stand for "Hom" and "Tensor".*

Note that when working with real representations instead of complex representations, the map $v \mapsto v^\dagger$ is linear instead of only conjugate linear, and consequently, dual representations are isomorphic to the original representation. In that case, one thus obtains $V \otimes W \cong \mathrm{Hom}_{\mathbb{R}}(V, W)$.

## A.4 Compact Subgroups and Restricted Representations

Let in this subsection $G \subseteq G'$ be a compact subgroup of the compact group $G'$. By restriction, one can view any unitary representation of $G'$ as one of $G$, which will become important both in our treatment of the adapted Wigner-Eckart Theorem in Section B and in considerations on the induced representation of a given irrep in Section D:

**Definition A.14** (Restricted Representation). Let $\rho' : G' \to \mathrm{U}(V')$ be a unitary representation of $G'$. We define $\mathrm{Res}_G^{G'} V' := V'$ and $\mathrm{Res}_G^{G'} \rho'$ by

$$\mathrm{Res}_G^{G'} \rho' : G \to \mathrm{U}(\mathrm{Res}_G^{G'} V') = \mathrm{U}(V'), \quad g \mapsto \rho'(g).$$

It is called the *restricted representation* of $\rho$ on $G$.

Crucially, restrictions of irreps of $G'$ need not be irreps of $G$. In analogy to the Clebsch-Gordan coefficients, which emerge when decomposing tensor products, one can define irrep-decomposition coefficients, which emerge when decomposing restrictions of irreps of the group $G'$ into irreps of $G$:

**Definition A.15** (Irrep-Decomposition Coefficients). Let $\rho_{j'} : G' \to \mathrm{U}(V_{j'})$ be an irrep of $G'$. Then there is a decomposition

$$\mathrm{Res}_G^{G'} V_{j'} \cong \bigoplus_{j \in \widehat{G}} \bigoplus_{t=1}^{[jj']} V_j,$$

where $[jj']$ is the multiplicity of irrep $V_j$ of $G$ in $\mathrm{Res}_G^{G'} V_{j'}$. This multiplicity is zero for all but finitely many $j \in \widehat{G}$. We denote the coupling coefficients between basis elements coming from this decomposition by $\langle t, jm|j'm'\rangle$, where $m \leq d_j$, $m' \leq d_{j'}$, $t \leq [jj']$. We call them the *irrep-decomposition coefficients*.

---

[11]Notice the change of order.

**Definition A.16** (Irrep-Decomposition Projections). We denote the projection from $\mathrm{Res}_G^{G'} V_{j'}$ to the $t$'th copy of $V_j$ by $\mathrm{ID}_t^{jj'} : \mathrm{Res}_G^{G'} V_{j'} \to V_j$. Is has as matrix coefficients the irrep-decomposition coefficients for fixed $t, j, j'$:

$$\langle t, jm \mid j'm' \rangle = \langle jm \vert \, \mathrm{ID}_t^{jj'} \, \vert j'm' \rangle. \tag{10}$$

Here, ID stands for "irrep" and "decomposition".

# B  A Generalized Wigner-Eckart Theorem for Steerable Kernels

The kernel space solution from this work (Theorem 2.1) will be based on the Wigner-Eckart theorem for steerable kernels from Lang & Weiler (2020). We thereby adapt that theorem in two practically relevant ways, and thus have to reprove the theorem: First, rather than working on a homogeneous space $X$ of $G$, we work with an arbitrary space $X$ equipped with an action of $G$. The special case where $X$ is a homogeneous space of $G$ then recovers the result in Lang & Weiler (2020). Second, by decomposing a certain Hom-representation instead of a tensor product representation, we make the computational process easier which finds the harmonic basis functions that build the equivariant kernels.

Finally, in Section B.3, we extend the previous result by reusing a $G'$-steerable basis to parameterize a $G$-steerable kernel space.

In this whole section, we freely make use of the concepts and notation defined in Section A.

## B.1  Basis-Independent Formulation of the Group-Restricted Wigner-Eckart Theorem

**Assumptions**  We explain the kernel space solution for the following general setting. We assume $G$ is a compact group. Also, $\mathbb{K}$ is one of the two fields $\mathbb{R}$ or $\mathbb{C}$ (with our applications focused on the case $\mathbb{R}$). Let $X$ be a *topological Hausdorff space* equipped with the Borel $\sigma$-algebra, making it a measurable space. We assume $X$ is equipped with a measure $\mu$ and a continuous action of $G$ on $X$. Since we want $L_{\mathbb{K}}^2(X)$ to be a unitary representation over $G$, we assume the action of $G$ on $X$ to preserve the measure $\mu$. All these assumptions are, for example, satisfied if $X$ is any homogeneous space of $G$, or if $G$ is a subgroup of $\mathrm{O}(n)$ and $X \subseteq \mathbb{R}^n$ any subset of the Euclidean space that is preserved under the action of $G$.

Finally, $\rho_l : G \to \mathrm{U}(V_l)$ and $\rho_J : G \to \mathrm{U}(V_J)$ are irreducible unitary input and output representations of $G$. Their dimension is denoted $d_l$ and $d_J$, respectively.

In this setting, our goal is to determine a basis for the space of $G$-steerable kernels $K : X \to \mathrm{Hom}_{\mathbb{K}}(V_l, V_J)$, which we define as any *square-integrable* function such that the steerability constraint is satisfied, i.e.: for all $g \in G$ and $x \in X$, the following holds:

$$K(gx) = \rho_{\mathrm{out}}(g) \circ K(x) \circ \rho_{\mathrm{in}}(g)^{-1}.$$

Note that, compared to Lang & Weiler (2020), the space $X$ is in general *not* a homogeneous space of the group $G$. We denote the space of all these $G$-steerable kernels by $\mathrm{Hom}_G(X, \mathrm{Hom}_{\mathbb{K}}(V_l, V_J))$. More details and intuitions on this "abstract" definition of steerable kernels (however, only for the special case where $X$ is a homogeneous space of $G$) can be found in Lang & Weiler (2020), Appendix C.1.

First, note that the space $L_{\mathbb{K}}^2(X)$ of square-integrable functions on $X$ carries an action of $G$ as well. Similarly as in Lang & Weiler (2020), the main ingredient of the Wigner-Eckart Theorem is a correspondence between $G$-steerable kernels on $X$ and *kernel operators*, i.e., intertwiners, on $L_{\mathbb{K}}^2(X)$. For this, recall that both $\mathrm{Hom}_{\mathbb{K}}(V_l, V_J)$ and $L_{\mathbb{K}}^2(X)$ carry a $G$-representation.

**Definition B.1** (The Kernel-Operator Correspondence). For any intertwiner $\mathcal{K} : L_{\mathbb{K}}^2(X) \to \mathrm{Hom}_{\mathbb{K}}(V_l, V_J)$, we define its restriction to $X$ by

$$\mathcal{K}|_X(x) := \mathcal{K}(\delta_x),$$

where $\delta_x$ is the Dirac delta function at $x \in X$.[12] $\mathcal{K}|_X$ is then a $G$-steerable kernel.

---

[12]Since the Dirac delta function is not actually square-integrable, formally, this is defined by performing a limit over square-integrable functions that more and more resemble the Dirac delta, see Lang & Weiler (2020).

In the other direction, for any $G$-steerable kernel $K : X \to \mathrm{Hom}_{\mathbb{K}}(V_l, V_J)$, we define the extension to $L^2_{\mathbb{K}}(X)$ by

$$\widehat{K}(f) := \int_X f(x)K(x)d\mu x.$$

This is then an intertwiner.

**Theorem B.2.** *Those two operations are inverse to each other, i.e., $\widehat{K}|_X = K$ and $\widehat{\mathcal{K}}|_X = \mathcal{K}$.*

*Proof.* This follows from the well-known fact that $L^2_{\mathbb{K}}(X)$ is isomorphic to its own *dual space* .

First, note that an intertwiner $\mathcal{K} : L^2_{\mathbb{K}}(X) \to \mathrm{Hom}_{\mathbb{K}}(V_l, V_J)$ belongs to $\bigoplus^{d_l \cdot d_J} L^2_{\mathbb{K}}(X)^*$, i.e. to $d_l \cdot d_J$ copies of the *dual space* of $L^2_{\mathbb{K}}(X)$. Conversely, a kernel $K : X \to \mathrm{Hom}_{\mathbb{K}}(V_l, V_J)$ belongs to $\bigoplus^{d_l \cdot d_J} L^2_{\mathbb{K}}(X)$, i.e. to $d_l \cdot d_J$ copies of $L^2_{\mathbb{K}}(X)$.

For $1 < p < \infty$ and $q = \frac{p}{p-1}$, there is a natural isomorphism between the spaces $L^p(X)^*$ and $L^q(X)$ given by the map:

$$\Phi : L^q(X) \to L^p(X)^*, \quad f \mapsto \hat{f}, \hat{f}(g) = \int_X f(x)g(x)d\mu(x)$$

$$\Phi^{-1} : L^p(X)^* \to L^q(X), \quad \hat{f} \mapsto f, f(x) = \hat{f}(\delta_x)$$

By choosing $p = q = 2$ we obtain the isomorphism between $L^2_{\mathbb{K}}(X)$ and $L^2_{\mathbb{K}}(X)^*$. Then, note that the operations $\widehat{(\cdot)}$ and $(\cdot)|_X$ are respectively equivalent to $\Phi$ and $\Phi^{-1}$ applied independently to each of the $d_l \cdot d_J$ subspaces isomorphic to $L^2_{\mathbb{K}}(X)$ or $L^2_{\mathbb{K}}(X)^*$. And finally, note that this isomorphism is still well-defined when restricted to the *equivariant* kernels and kernel operators, respectively. $\square$

Note now that, since $G$ is acting as an isometry on $X$, its action on $L^2_{\mathbb{K}}(X)$ is unitary. By Corollary 1, it follows that this actions decomposes into a *direct sum* of irreps of $G$. It follows that $L^2_{\mathbb{K}}(X)$ also decomposes into a direct sum of invariant subspaces $L^2_{\mathbb{K}}(X) \cong \widehat{\bigoplus}_{j \in \hat{G}} \widehat{\bigoplus}_i^{m_j} V_j^i$, where $m_j$ is the multiplicity of the irrep $\rho_j \in \hat{G}$ and each $V_j^i \cong V_j$ is acted on by $G$ through $\rho_j \in \hat{G}$. We denote a basis for $V_j^i$ with $\{Y_{ji}^m : X \to \mathbb{K}\}_m^{d_j}$. Note that $m_j$ may be infinite, even uncountably large, since we do not assume $X$ to be a homogeneous space of $G$.

**Definition B.3** (Steerable Basis). The union of $\{Y_{ji}^m : X \to \mathbb{K} \mid j \in \hat{G}, i \le m_j, m \le d_j\}$ forms a basis for $L^2_{\mathbb{K}}(X)$. We call this basis *steerable* since the action of $G$ on a function $L^2_{\mathbb{K}}(X)$ is realized just by recombination of the linear coefficients used to expand this basis (via its action on each invariant subspace through its irreps).

Note that this definition is compatible with the one in Freeman & Adelson (1991).

Similar to the harmonic projection in Def. A.8, we define an orthogonal projection on the invariant subspaces of $L^2_{\mathbb{K}}(X)$:

**Definition B.4** (Steerable Basis Projections). We denote the orthogonal projection from $L^2_{\mathbb{K}}(X)$ to the $i$'th copy of $V_j$ by $p_{ji} : L^2_{\mathbb{K}}(X) \to V_j$. On the steerable basis, it is given by

$$p_{ji}\big(Y_{\widetilde{ji}}^{\widetilde{m}}\big) = \delta_{j\widetilde{j}}\delta_{i\widetilde{i}} \cdot |j\widetilde{m}\rangle,$$

which just follows from the fact that the steerable basis forms an orthonormal basis of $L^2_{\mathbb{K}}(X)$, which is then mapped to the corresponding orthonormal basis of $V_j$, the representation space of $\rho_j \in \hat{G}$.

With all this in mind, we can formulate our basis-independent Wigner-Eckart Theorem for $G$-steerable kernels:

**Theorem B.5** (Wigner-Eckart Theorem for Steerable Kernels). *The basis-independent version of the Wigner-Eckart theorem consists of two parts:*

*1. There is an isomorphism of vector spaces*

$$\mathrm{GKer} : \overset{m_j}{\overbrace{\bigoplus_{j \in \hat{G}} \bigoplus_{i=1}}} \overset{[j(l^\dagger J)]}{\bigoplus_{s=1}} \mathrm{End}_{G,\mathbb{K}}(V_j) \to \mathrm{Hom}_G(X, \mathrm{Hom}_{\mathbb{K}}(V_l, V_J)),$$

where the hat $\widehat{(\cdot)}$ denotes a topological closure and can be ignored by the reader.

2. *Explicitly,* GKer *is given by*

$$\mathrm{GKer}((c_{jis})_{jis}) = \sum_{j \in \hat{G}} \sum_{i=1}^{m_j} \sum_{s=1}^{[j(l^\dagger J)]} \Psi_{\mathrm{HT}} \circ i_s^{j(l^\dagger J)} \circ c_{jis} \circ p_{ji}|_X.$$

*Proof.* For the first statement, we observe:

$$\mathrm{Hom}_G(X, \mathrm{Hom}_{\mathbb{K}}(V_l, V_J)) \overset{(1)}{\cong} \mathrm{Hom}_{G,\mathbb{K}}\left(L^2_{\mathbb{K}}(X), \mathrm{Hom}_{\mathbb{K}}(V_l, V_J)\right)$$

$$\overset{(2)}{\cong} \widehat{\bigoplus_{j \in \hat{G}}} \bigoplus_{i=1}^{m_j} \mathrm{Hom}_{G,\mathbb{K}}(V_j, V_{l^\dagger} \otimes V_J)$$

$$\overset{(3)}{\cong} \widehat{\bigoplus_{j \in \hat{G}}} \bigoplus_{i=1}^{m_j} \bigoplus_{s=1}^{[j(l^\dagger J)]} \mathrm{End}_{G,\mathbb{K}}(V_j)$$

In $(1)$, we perform the linear extension to the space of square integrable functions from Definition B.1, using Theorem B.2. In $(2)$, we use the Peter-Weyl Theorem A.6, and the isomorphism $\Psi_{\mathrm{HT}}$ which we defined in Proposition A.13. Note that the decomposition from the Peter-Weyl Theorem is $G$-equivariant, so this step is valid. In $(3)$, we use the Clebsch-Gordan decomposition from Definition A.9, and Schur's Lemma which shows that $V_j$ can only have a nontrivial homomorphism to $V_j$, and not to any other irrep. From right to left, we call the isomorphism GKer. This proves the first statement.

For the second statement, we go through the sequence of isomorphisms from bottom to top and trace back where an arbitrary endomorphism tuple "comes from":

$$\mathrm{GKer} : (c_{jis})_{jis} \overset{(3)}{\mapsto} \left( \sum_{s=1}^{[j(l^\dagger J)]} i_s^{j(l^\dagger J)} \circ c_{jis} \right)_{ji}$$

$$\overset{(2)}{\mapsto} \sum_{j \in \hat{G}} \sum_{i=1}^{m_j} \sum_{s=1}^{[j(l^\dagger J)]} \Psi_{\mathrm{HT}} \circ i_s^{j(l^\dagger J)} \circ c_{jis} \circ p_{ji}$$

$$\overset{(1)}{\mapsto} \sum_{j \in \hat{G}} \sum_{i=1}^{m_j} \sum_{s=1}^{[j(l^\dagger J)]} \Psi_{\mathrm{HT}} \circ i_s^{j(l^\dagger J)} \circ c_{jis} \circ p_{ji}|_X.$$

This finishes the proof of the second statement, and thus of the theorem. $\qquad\square$

## B.2   STEERABLE KERNEL BASES

After the basis-independent version of the theorem, we now state how a basis for the space of steerable kernels can be constructed.

The basis elements of the representation space $V_l$, $V_J$ and $V_j$ are denoted $|ln\rangle$, $|JM\rangle$ and $|jm\rangle$, respectively, where the indices range in $0 \le n \le d_l$, $0 \le M \le d_J$, and $0 \le m \le d_j$. Sometimes, we write also $Y_j^M$ instead of $|JM\rangle$, etc., in order to remind to the connection with harmonic basis functions.

**Theorem B.6** (Steerable Basis Kernels). *A basis of the space of $G$-steerable kernels* $\mathrm{Hom}_G(X, \mathrm{Hom}_{\mathbb{K}}(V_l, V_J))$ *is given by basis kernels* $\left\{ K_{jisr} | j \in \hat{G}, i \le m_j, s \le [j(l^\dagger J)], r \le E_j \right\}$, *which are given as follows:* $K_{jisr} = \mathrm{GKer}(\boldsymbol{c}^{jisr})$, *where* $\boldsymbol{c}^{jisr}$ *is zero at every entry except at position* $jis$, *where it takes on value* $c_r^j$.[13]

---

[13]Remember that $c_r^j$ is the $r$'th basis endomorphism of $\rho_j$.

*Furthermore, the matrix-elements of such a basis kernel with respect to orthonormal bases $|ln\rangle$ of $V_l$ and $|JM\rangle$ of $V_J$, where $M \leq d_J$ and $n \leq d_l$, are given by:*

$$\langle JM|K_{jisr}(x)|ln\rangle$$
$$= \sum_{\widetilde{m}=1}^{d_j} \overline{\langle s, j\widetilde{m}|l^\dagger n; JM\rangle} \sum_{m=1}^{d_j} \langle j\widetilde{m}|c_r^j|jm\rangle \langle i, jm|x\rangle \tag{11}$$

*where the overline denotes complex conjugation, and with $\langle i, jm|x\rangle := \overline{Y_{ji}^m(x)}$ for steerable basis functions $Y_{ji}^m : X \to \mathbb{K}$.*

*Proof.* That these kernels form a basis follows from the fact that the $c^{jisr}$ obviously form a basis of endomorphism tuples, and that GKer is an isomorphism by Theorem B.5. Concretely, this theorem then shows that

$$K_{jisr} = \Psi_{\mathrm{HT}} \circ i_s^{j(l^\dagger J)} \circ c_r^j \circ p_{ji}|_X.$$

We write $\widehat{K_{jisr}}$ for the extension from Definition B.1, which is just given by $\widehat{K_{jisr}} = \Psi_{\mathrm{HT}} \circ i_s^{j(l^\dagger J)} \circ c_r^j \circ p_{ji}$. Furthermore, we expand the Dirac delta function $\delta_x$ for $x \in X$ in the steerable basis of $L_{\mathbb{K}}^2(X)$ as

$$\delta_x = \sum_{\widetilde{j} \in \hat{G}} \sum_{\widetilde{i}=1}^{m_{\widetilde{j}}} \sum_{\widetilde{m}=1}^{d_{\widetilde{j}}} \left\langle Y_{\widetilde{j}\widetilde{i}}^{\widetilde{m}} \middle| \delta_x \right\rangle \cdot Y_{\widetilde{j}\widetilde{i}}^{\widetilde{m}}.$$

This expansion can be justified with a similar approximation procedure as in the proof of Lang & Weiler (2020), Theorem D.13. Thereby, we will in the following write the coefficients as $\left\langle \widetilde{i}, \widetilde{j}\widetilde{m} \middle| x \right\rangle := \left\langle Y_{\widetilde{j}\widetilde{i}}^{\widetilde{m}} \middle| \delta_x \right\rangle$, which is also equal to $\overline{Y_{\widetilde{j}\widetilde{i}}^{\widetilde{m}}(x)}$. With these tricks, we obtain:

$$\langle JM|K_{jisr}(x)|ln\rangle$$
$$\overset{(1)}{=} \left\langle JM \middle| \widehat{K_{jisr}}(\delta_x) \middle| ln \right\rangle$$
$$\overset{(2)}{=} \sum_{\widetilde{j} \in \hat{G}} \sum_{\widetilde{i}=1}^{m_{\widetilde{j}}} \sum_{\widetilde{m}=1}^{d_{\widetilde{j}}} \left\langle \widetilde{i}, \widetilde{j}\widetilde{m} \middle| x \right\rangle \left\langle JM \middle| \left( \Psi_{\mathrm{HT}} \circ i_s^{j(l^\dagger J)} \circ c_r^j \circ p_{ji} \right) (Y_{\widetilde{j}\widetilde{i}}^{\widetilde{m}}) \middle| ln \right\rangle$$
$$\overset{(3)}{=} \sum_{m=1}^{d_j} \langle i, jm|x\rangle \left\langle JM \middle| \left( \Psi_{\mathrm{HT}} \circ i_s^{j(l^\dagger J)} \circ c_r^j \right) (Y_j^m) \middle| ln \right\rangle$$
$$\overset{(4)}{=} \sum_{m=1}^{d_j} \langle i, jm|x\rangle \sum_{\widetilde{m}=1}^{d_j} \langle j\widetilde{m}|c_r^j|jm\rangle \left\langle JM \middle| \left( \Psi_{\mathrm{HT}} \circ i_s^{j(l^\dagger J)} \right) (Y_j^{\widetilde{m}}) \middle| ln \right\rangle$$
$$\overset{(5)}{=} \sum_{\widetilde{m}=1}^{d_j} \left\langle JM \middle| \left( \Psi_{\mathrm{HT}} \circ i_s^{j(l^\dagger J)} \right) (Y_j^{\widetilde{m}}) \middle| ln \right\rangle \sum_{m=1}^{d_j} \langle j\widetilde{m}|c_r^j|jm\rangle \langle i, jm|x\rangle$$

We first explain the steps so far and will then deal with the last term. Step (1) uses the extension of $K$ from $X$ to $L_{\mathbb{K}}^2(X)$. Step (2) uses the expansion of $\delta_x$ explained before. Step (3) uses the property of the projection discussed in Definition B.4 and a renaming of $\widetilde{m}$ to $m$. Step (4) expands $c_r^j(Y_j^m)$ in terms of the basis vectors $Y_j^{\widetilde{m}}$, with the expansion coefficients being the matrix elements of $c_r^j$. Step (5) is then just a reordering of the terms.

Now we deal with the first factor in the last term:

$$\left\langle JM \middle| \left(\Psi_{\mathrm{HT}} \circ i_s^{j(l^\dagger J)}\right)(Y_j^{\widetilde{m}}) \middle| ln \right\rangle$$

$$\overset{(a)}{=} \sum_{\widetilde{M}} \sum_{\widetilde{n}} \left\langle l^\dagger \widetilde{n}; J\widetilde{M} \middle| i_s^{j(l^\dagger J)} \middle| j\widetilde{m} \right\rangle \left\langle JM \middle| \Psi_{\mathrm{HT}} \left(\middle| l^\dagger \widetilde{n}; J\widetilde{M} \right\rangle\right) \middle| ln \right\rangle$$

$$\overset{(b)}{=} \sum_{\widetilde{M}} \sum_{\widetilde{n}} \overline{\left\langle s, j\widetilde{m} \middle| l^\dagger \widetilde{n}; J\widetilde{M} \right\rangle} \cdot \left\langle JM \middle| \left(\langle l\widetilde{n}|\right)_{|J\widetilde{M}\rangle} \middle| ln \right\rangle$$

$$\overset{(c)}{=} \sum_{\widetilde{M}} \sum_{\widetilde{n}} \overline{\left\langle s, j\widetilde{m} \middle| l^\dagger \widetilde{n}; J\widetilde{M} \right\rangle} \left\langle JM \middle| \langle l\widetilde{n}|ln\rangle \middle| J\widetilde{M} \right\rangle$$

$$\overset{(d)}{=} \sum_{\widetilde{M}} \sum_{\widetilde{n}} \overline{\left\langle s, j\widetilde{m} \middle| l^\dagger \widetilde{n}; J\widetilde{M} \right\rangle} \delta_{n\widetilde{n}} \delta_{M\widetilde{M}}$$

$$\overset{(e)}{=} \overline{\langle s, j\widetilde{m}|l^\dagger n; JM\rangle}.$$

In step (a), we expand $i_s^{j(l^\dagger J)}(Y_j^{\widetilde{m}})$ in the basis of $V_{l^\dagger} \otimes V_J$. In step (b), we use the definition of the Clebsch-Gordan coefficients using the embedding $i_s^{j(l^\dagger J)}$ and of $\Psi_{\mathrm{HT}}$ given in Proposition A.13. Also, the definition of the dual basis given after Definition A.12 is used as the equality $|l^\dagger \widetilde{n}\rangle = \langle l\widetilde{n}|$. In step (c), we use eq. 9. Step (d) and $(e)$ are clear. Plugging this intermediate result into our earlier computation finishes the proof. $\qquad\square$

We now formulate a matrix-version of this result that is more suitable to implementation. We thereby identify $K_{jisr}(x)$ as a matrix in $\mathbb{K}^{d_J \times d_l}$, for chosen bases $|JM\rangle$ in $V_J$ and $|ln\rangle$ in $V_l$. I.e., $K_{jisr}(x)$ is viewed as the matrix with coefficients $\langle JM|K_{jisr}(x)|ln\rangle$. Furthermore, we identify $c_r^j$ with the matrix in $\mathbb{K}^{d_j \times d_j}$ with coefficients $\langle j\widetilde{m}|c_r^j|jm\rangle$. Additionally, with $Y_{ji}(x)$ we mean the column vector in $\mathbb{K}^{d_j}$ with entries $\langle i, jm|x\rangle$. Finally, with $\mathrm{CG}_s^{j(l^\dagger J)}$ we mean the matrix of Clebsch-Gordan coefficients $\left\langle s, j\widetilde{m} \middle| l^\dagger m; JM \right\rangle$ of shape $d_j \times (d_l \times d_J)$. Its conjugate transpose is denoted as $\left[\mathrm{CG}_s^{j(l^\dagger J)}\right]^\dagger$, which is of shape $(d_J \times d_l) \times d_j$. When using it in matrix multiplications, it is interpreted as having $d_J \cdot d_l$ "rows" and $d_j$ columns. This is compatible with the vectorization of steerable kernels which was discussed in the main paper.

**Theorem B.7** (Matrix-Version of Steerable Basis Kernels). *The basis kernel evaluated at $x$, $K_{jisr}(x)$, is given by the matrix*

$$K_{jisr}(x) = \left[\mathrm{CG}_s^{j(l^\dagger J)}\right]^\dagger \cdot c_r^j \cdot Y_{ji}(x) \tag{12}$$

*where each dot means conventional matrix multiplication.*

*Proof.* We do the sanity check, i.e., we test whether the resulting matrix is well-defined and of shape $d_J \times d_l$. This can be verified by observing that the shape of the right-hand-side of eq. 12 is:

$$[(d_J \times d_l) \times d_j] \cdot [d_j \times d_j] \cdot [d_j] = [d_J \times d_l].$$

The full proof of eq. 12 follows directly from eq. 11. $\qquad\square$

*Remark* B.8. Note that if $X$ is a homogeneous space for $G$, then the steerable basis $\{Y_{ji}^m\}_{jim}$ is the harmonic basis in Theorem A.6. Additionally, if all endomorphism spaces are 1-dimensional, which is for example always the case for complex representations, then $c_r^j$ can be chosen to be the identity matrix and thus completely omitted from the formula.

Additionally, if one is concerned with real representations, then $l^\dagger = l$ and the complex conjugation in the Clebsch-Gordan coefficients can be omitted, which leads to the following formula which we implement in this work:

$$K_{jisr}(x) = \left[\mathrm{CG}_s^{j(lJ)}\right]^T \cdot c_r^j \cdot Y_{ji}(x). \tag{13}$$

For the special case that $G = \mathrm{SO}(3)$, $X = S^2$ and $\mathbb{K} = \mathbb{R}$, one obtains the basis kernels

$$K_j(x) = \left[\mathrm{CG}^{j(lJ)}\right]^T \cdot Y_j(x),$$

where for $|l - J| \leq j \leq l + J$, the $Y_j : S^2 \to \mathbb{R}^{2j+1}$ are the spherical harmonics. This result was derived in Weiler et al. (2018a). Many other examples for $X$ homogeneous space of $G$, with a tensor-product decomposition instead of a Hom-space decomposition, can be found in Lang & Weiler (2020).

## B.3 A Wigner-Eckart Theorem for Group-Restricted Steerable Kernels

In this section, we consider a special case of the previous theorem where $X$ carries an action of a compact group $G'$, with $G \leq G'$. We assume this action to extend the action of $G$ on $X$. This result is useful to reuse $G'$-steerable kernels to parameterize new $G$-steerable spaces without designing a new basis for $L^2_{\mathbb{K}}(\mathbb{R}^n)$.

First of all, we denote by $\hat{G}'$ a set of representatives of the irreps of $G'$ and by $\{Y^{m'}_{j'i'} \,|\, j' \in \hat{G}', i' \leq m_{j'}, m' \leq d_{j'}\}$ a $G'$-steerable basis for $X$. Recall also the definition of group restriction and irrep decomposition coefficients from Definition A.15. In particular, $[jj']$ is the multiplicity of $\rho_j \in \hat{G}$ inside $\mathrm{Res}^G_{G'} \rho_{j'}$, with $j' \in \hat{G}'$, and $\mathrm{ID}^{jj'}_t$ is the projection from $\mathrm{Res}^{G'}_G V_{j'}$ to the $t$'th copy of $V_j$ (Definition A.16).

**Proposition B.9.** *The $G'$-steerable basis for $X$ given by $\{Y^{m'}_{j'i'} \,|\, j' \in \hat{G}', i' \leq m_{j'}, m' \leq d_{j'}\}$ can be turned into a $G$-steerable basis defined as:*

$$\left\{ Y^m_{j(i'j't)} = \left[ \mathrm{ID}^{[jj']}_t Y_{j'i'} \right]^m \,\Big|\, j' \in \hat{G}', i' \leq m_{j'}, j \in \hat{G}, t \leq [jj'], m \leq d_j \right\}$$

*Proof.* Because the space $X$ carries an action of $G'$, we can find the action of $G$ on $L^2_{\mathbb{K}}(X)$ via group restriction:

$$\mathrm{Res}^G_{G'} L^2_{\mathbb{K}}(X) = \mathrm{Res}^G_{G'} \widehat{\bigoplus}_{j' \in \hat{G}'} \widehat{\bigoplus}^{m_{j'}}_{i'=1} V_{j'}$$

$$= \widehat{\bigoplus}_{j' \in \hat{G}'} \widehat{\bigoplus}^{m_{j'}}_{i'=1} \mathrm{Res}^G_{G'} V_{j'}$$

$$= \widehat{\bigoplus}_{j' \in \hat{G}'} \widehat{\bigoplus}^{m_{j'}}_{i'=1} \widehat{\bigoplus}_{j \in \hat{G}} \widehat{\bigoplus}^{[jj']}_{t=1} V_j$$

$\square$

We can plug this basis in Theorem B.7 to obtain the following result:

**Theorem B.10** (Matrix-Version of Group-Restricted Steerable Basis Kernels)**.** *The basis kernel evaluated at $x$, $K_{j'i'jtsr}(x')$, is given by the matrix*

$$K_{j'i'jtsr}(x) = \left[ \mathrm{CG}^{j(l^\dagger J)}_s \right]^\dagger \cdot c^j_r \cdot \mathrm{ID}^{jj'}_t \cdot Y_{j'i'}(x) \tag{14}$$

*where each dot means conventional matrix multiplication.*

## B.4 Extended Examples of Steerable Kernel Bases

In this section, we provide an extended version of the two examples in Section 2.2.

In the following examples we will consider the space $X = \mathbb{R}^2$ and the two groups $\mathrm{SO}(2)$ and $\mathrm{C}_4$, i.e. the group of all planar rotations and the group of rotations by multiples of $\pi/2$ radians.

$\mathrm{SO}(2)$**-Steerable Kernels** First of all, recall the (real) irreducible representations of $G = \mathrm{SO}(2)$. For $r_\theta \in \mathrm{SO}(2)$:

$$\rho_0(r_\theta) = 1$$

$$\rho_j(r_\theta) = \begin{bmatrix} \cos j \cdot \theta & -\sin j \cdot \theta \\ \sin j \cdot \theta & \cos j \cdot \theta \end{bmatrix} \quad j \in \mathbb{N}^+$$

where $j$ is a non-negative integer which can be interpreted as the rotational frequency. All irreps are 2 dimensional but for the frequency $j = 0$, which is 1 dimensional. We consider the basis $\mathcal{B} = \{Y_{ji}\}_{ji}$

generated by circular harmonics combined with a Gaussian radial profile, as in Worrall et al. (2017); Weiler et al. (2018b); Weiler & Cesa (2019), defined as:

$$Y_{ji}(r, \phi) = \omega_{R_i}(r) \cdot \begin{bmatrix} \cos(j \cdot \phi) \\ \sin(j \cdot \phi) \end{bmatrix} \in \mathbb{R}^2 \tag{15}$$

where $\omega_{R_i} : \mathbb{R} \to \mathbb{R}$ is a Gaussian radial kernel centered around $R_i \in \mathbb{R}$, defining a ring of radius $R_i$. Hence, in this basis, $i$ indexes the circular shells, while $j$ indexes the angular frequencies along each ring. For simplicity, in the following examples we will only consider a single ring, indexed by $i = 1$. This setting is similar to the $X = \mathcal{S}^1$ examples discussed in Section 2.2, since each ring is isomorphic to a circle $\mathcal{S}^1$. One can verify that the basis elements $Y_{ji}$ in $\mathcal{B}$ are $G = \mathrm{SO}(2)$-steerable using the irrep $\rho_j$, i.e.

$$Y_{ji}(r, \phi + \theta) = \rho_j(r_\theta) Y_{ji}(r, \phi) .$$

To apply Theorem 2.1, we need a basis for the endomorphism space of each irrep and the irreps decomposition between their tensor-products. If $j > 0$, $\rho_j$ is a real irrep of *complex type* so its endomorphism space is 2-dimensional and is spanned by[14]:

$$\left\{ c_1^j = \begin{bmatrix} 1 & 0 \\ 0 & 1 \end{bmatrix}, c_2^j = \begin{bmatrix} 0 & 1 \\ -1 & 0 \end{bmatrix} \right\} .$$

If $J = 0$, $\rho_0 \otimes \rho_l = \rho_l$ and if $l = 0$, $\rho_J \otimes \rho_0 = \rho_J$. For $J, l > 0$, $\rho_J \otimes \rho_l \cong \rho_{|J-l|} \oplus \rho_{J+l}$ if $J \neq l$ or $\rho_0 \oplus \rho_0 \oplus \rho_{J+l}$ otherwise. In particular, in the first case, using some trigonometric identities, one can verify that

$$\mathrm{CG}^{lJ} = \frac{1}{\sqrt{2}} \begin{bmatrix} 1 & 0 & 0 & 1 \\ 0 & 1 & -1 & 0 \\ 1 & 0 & 0 & -1 \\ 0 & 1 & 1 & 0 \end{bmatrix}$$

satisfies $\rho_J \otimes \rho_l = [\mathrm{CG}^{lJ}]^T \left( \rho_{|J-l|} \oplus \rho_{J+l} \right) \mathrm{CG}^{lJ}$. In this case, $\mathrm{CG}_1^{j(lJ)} = \frac{1}{\sqrt{2}} \begin{bmatrix} 1 & 0 & 0 & 1 \\ 0 & 1 & -1 & 0 \end{bmatrix}$ if $j = |J - l|$ and $\mathrm{CG}_1^{j(lJ)} = \frac{1}{\sqrt{2}} \begin{bmatrix} 1 & 0 & 0 & -1 \\ 0 & 1 & 1 & 0 \end{bmatrix}$ if $j = J + l$. Thus, for $l \neq J > 0$, Theorem 2.1 prescribes a basis containing the following elements (for a single ring at radius $R_i$):

$$\kappa_{|J-l|,i,1,1}(r, \phi) = \omega_{R_i}(r) \cdot \begin{bmatrix} \cos(|J-l|\phi) \\ \sin(|J-l|\phi) \\ -\sin(|J-l|\phi) \\ \cos(|J-l|\phi) \end{bmatrix} ,$$

$$\kappa_{|J-l|,i,1,2}(r, \phi) = \omega_{R_i}(r) \cdot \begin{bmatrix} \sin(|J-l|\phi) \\ -\cos(|J-l|\phi) \\ \cos(|J-l|\phi) \\ \sin(|J-l|\phi) \end{bmatrix} ,$$

$$\kappa_{J+l,i,1,1}(r, \phi) = \omega_{R_i}(r) \cdot \begin{bmatrix} \cos((J+l)\phi) \\ \sin((J+l)\phi) \\ \sin((J+l)\phi) \\ -\cos((J+l)\phi) \end{bmatrix} ,$$

$$\kappa_{J+l,i,1,2}(r, \phi) = \omega_{R_i}(r) \cdot \begin{bmatrix} \sin((J+l)\phi) \\ -\cos((J+l)\phi) \\ -\cos((J+l)\phi) \\ -\sin((J+l)\phi) \end{bmatrix} .$$

Observe that these 4 basis kernels *are identical to* the ones in Table 8 of Weiler & Cesa (2019), up to vectorization and a minus sign for $\kappa_{|J-l|,i,1,2}$ and $\kappa_{J+l,i,1,2}$. Fig. 2 shows the two kernels among these four obtained with $j = |J - l| = 1$, in the case $l = 1$ and $J = 2$. For $l = 0$, $j = J$ and $\mathrm{CG}^{0J}$

---

[14]Observe that we expressed $\rho_j$ in a basis such that is has the form described in Lemma C.13, which implies the endomorphism space is spanned by the basis described in Section C.3.

is the identity matrix; then, for $J > 0$, Theorem 2.1 prescribes the following basis elements:

$$\kappa_{J,i,1,1}(r, \phi) = \omega_{R_i}(r) \cdot \begin{bmatrix} \cos(J\phi) \\ \sin(J\phi) \end{bmatrix},$$

$$\kappa_{J,i,1,2}(r, \phi) = \omega_{R_i}(r) \cdot \begin{bmatrix} \sin(J\phi) \\ -\cos(J\phi) \end{bmatrix}.$$

which are also shown in Fig. 3.

$C_4$-**Steerable Kernels**   We construct a $G = C_4$-steerable basis through Eq. 5 by considering the $G' = \mathrm{SO}(2)$-steerable basis described before in Eq. 15. First, recall the representation theory of $G = C_4$. For $p \in \{0, 1, 2, 3\}$, $r_{p\frac{\pi}{2}} = (r_{\frac{\pi}{2}})^p \in C_4$; $C_4$ has the following three irreps:

$$\rho_0(r_{p\frac{\pi}{2}}) = 1$$

$$\rho_1(r_{p\frac{\pi}{2}}) = \begin{bmatrix} \cos p\frac{\pi}{2} & -\sin p\frac{\pi}{2} \\ \sin p\frac{\pi}{2} & \cos p\frac{\pi}{2} \end{bmatrix}$$

$$\rho_2(r_{p\frac{\pi}{2}}) = (-1)^p$$

which are, respectively, 1, 2 and 1 dimensional. Again, we index the irreps by $j \in \{0, 1, 2\}$, which can be interpreted as the rotational frequency. To apply Eq. 5, we need to know the irreps decomposition of $\mathrm{Res}_{C_4}^{\mathrm{SO}(2)} \rho_{j'}$, for each irrep $\rho_{j'}$ of $\mathrm{SO}(2)$. We have, for $t \in \mathbb{N}$:

$$\mathrm{Res}_{C_4}^{\mathrm{SO}(2)} \rho_0 = \rho_0$$

$$\mathrm{Res}_{C_4}^{\mathrm{SO}(2)} \rho_{4t} = \rho_0 \oplus \rho_0$$

$$\mathrm{Res}_{C_4}^{\mathrm{SO}(2)} \rho_{4t+1} = \rho_1$$

$$\mathrm{Res}_{C_4}^{\mathrm{SO}(2)} \rho_{4t+2} = \rho_2 \oplus \rho_2$$

$$\mathrm{Res}_{C_4}^{\mathrm{SO}(2)} \rho_{4t+3} = \begin{bmatrix} 1 & 0 \\ 0 & -1 \end{bmatrix} \rho_1 \begin{bmatrix} 1 & 0 \\ 0 & -1 \end{bmatrix}$$

so, $\mathrm{ID}^{j'} = \begin{bmatrix} 1 & 0 \\ 0 & -1 \end{bmatrix}$ if $j' = 4t + 3$ and $\mathrm{ID}^{j'}$ is the identity otherwise. Thus, if $\mathcal{B}' = \{Y_{j'i'}\}_{j'i'}$ is the $\mathrm{SO}(2)$-steerable basis described before in Eq. 15, according to Eq. 5, a $C_4$-steerable basis $\mathcal{B}$ contains the following elements for each $t \in \mathbb{N}$:

$$Y_{0,(i',0,1)}(r, \phi) = Y_{0,i'}(r, \phi) = \omega_{R_{i'}}(r),$$

$$Y_{0,(i',4t,1)}(r, \phi) = Y_{4t,i'}^1(r, \phi) = \omega_{R_{i'}}(r) \cos(4t\phi)$$

$$Y_{0,(i',4t,2)}(r, \phi) = Y_{4t,i'}^2(r, \phi) = \omega_{R_{i'}}(r) \sin(4t\phi),$$

$$Y_{1,(i',4t+1,1)}(r, \phi) = Y_{4t+1,i'}(r, \phi) = \omega_{R_{i'}}(r) \begin{bmatrix} \cos((4t+1)\phi) \\ \sin((4t+1)\phi) \end{bmatrix},$$

$$Y_{2,(i',4t+2,1)}(r, \phi) = Y_{4t+2,i'}^1(r, \phi) = \omega_{R_{i'}}(r) \cos((4t+2)\phi),$$

$$Y_{2,(i',4t+2,2)}(r, \phi) = Y_{4t+2,i'}^2(r, \phi) = \omega_{R_{i'}}(r) \sin((4t+2)\phi),$$

$$Y_{1,(i',4t+3,1)}(r, \phi) = \begin{bmatrix} 1 & 0 \\ 0 & -1 \end{bmatrix} Y_{4t+3,i'}(r, \phi) = \omega_{R_{i'}}(r) \begin{bmatrix} \cos((4t+3)\phi) \\ -\sin((4t+3)\phi) \end{bmatrix}$$

The endomorphism space of $\rho_0$ and $\rho_2$ is one dimensional and is spanned by the identity, while the endomorphism space of $\rho_1$ is spanned by

$$\left\{ c_1^1 = \begin{bmatrix} 1 & 0 \\ 0 & 1 \end{bmatrix}, c_2^1 = \begin{bmatrix} 0 & 1 \\ -1 & 0 \end{bmatrix} \right\}.$$

The tensor products decompose as follows:

$$\rho_0 \otimes \rho_j = \rho_j \otimes \rho_0 = \rho_j$$

$$\rho_2 \otimes \rho_2 = \rho_0$$

$$\rho_1 \otimes \rho_2 = \rho_2 \otimes \rho_1 = \begin{bmatrix} 1 & 0 \\ 0 & -1 \end{bmatrix} \rho_1 \begin{bmatrix} 1 & 0 \\ 0 & -1 \end{bmatrix}$$

$$\rho_1 \otimes \rho_1 = [\mathrm{CG}^{11}]^T (\rho_0 \oplus \rho_0 \oplus \rho_2 \oplus \rho_2) \, \mathrm{CG}^{11}$$

where

$$CG^{11} = \frac{1}{\sqrt{2}} \begin{bmatrix} 1 & 0 & 0 & 1 \\ 0 & 1 & -1 & 0 \\ 1 & 0 & 0 & -1 \\ 0 & 1 & 1 & 0 \end{bmatrix}$$

whose rows are respectively denoted $CG_1^{0(11)}, CG_2^{0(11)}, CG_1^{2(11)}$ and $CG_2^{2(11)}$. Finally, Theorem 2.1 prescribes the following basis elements for each $t \in \mathbb{N}$ and each $i'$:

- $l, J \in \{0, 2\}$:

$$\kappa_{j,(i',4t+j,1),1,1}(r, \phi) = \omega_{R_{i'}}(r) \cos((4t+j)\phi)$$
$$\kappa_{j,(i',4t+j,2),1,1}(r, \phi) = \omega_{R_{i'}}(r) \sin((4t+j)\phi)$$

  where $j = |l - J| \in \{0, 2\}$.

- $l = 0$ and $J = 1$ (or $l = 1$ and $J = 0$):

$$\kappa_{1,(i',4t+1,1),1,1}(r, \phi) = \omega_{R_{i'}}(r) \begin{bmatrix} \cos((4t+1)\phi) \\ \sin((4t+1)\phi) \end{bmatrix},$$

$$\kappa_{1,(i',4t+1,1),1,2}(r, \phi) = \omega_{R_{i'}}(r) \begin{bmatrix} \sin((4t+1)\phi) \\ -\cos((4t+1)\phi) \end{bmatrix},$$

$$\kappa_{1,(i',4t+3,1),1,1}(r, \phi) = \omega_{R_{i'}}(r) \begin{bmatrix} \cos((4t+3)\phi) \\ -\sin((4t+3)\phi) \end{bmatrix},$$

$$\kappa_{1,(i',4t+3,1),1,2}(r, \phi) = \omega_{R_{i'}}(r) \begin{bmatrix} -\sin((4t+3)\phi) \\ -\cos((4t+3)\phi) \end{bmatrix}$$

- $l = 2$ and $J = 1$ (or $l = 1$ and $J = 2$):

$$\kappa_{1,(i',4t+1,1),1,1}(r, \phi) = \omega_{R_{i'}}(r) \begin{bmatrix} \cos((4t+1)\phi) \\ -\sin((4t+1)\phi) \end{bmatrix},$$

$$\kappa_{1,(i',4t+1,1),1,2}(r, \phi) = \omega_{R_{i'}}(r) \begin{bmatrix} \sin((4t+1)\phi) \\ \cos((4t+1)\phi) \end{bmatrix},$$

$$\kappa_{1,(i',4t+3,1),1,1}(r, \phi) = \omega_{R_{i'}}(r) \begin{bmatrix} \cos((4t+3)\phi) \\ \sin((4t+3)\phi) \end{bmatrix},$$

$$\kappa_{1,(i',4t+3,1),1,2}(r, \phi) = \omega_{R_{i'}}(r) \begin{bmatrix} -\sin((4t+3)\phi) \\ \cos((4t+3)\phi) \end{bmatrix}$$

- $l = 1$ and $J = 1$:

$$\kappa_{0,(i',4t,1),1,1}(r, \phi) = \omega_{R_{i'}}(r) \begin{bmatrix} \cos(4t\phi) & 0 & 0 & \cos(4t\phi) \end{bmatrix}^T,$$

$$\kappa_{0,(i',4t,1),2,1}(r, \phi) = \omega_{R_{i'}}(r) \begin{bmatrix} 0 & \cos(4t\phi) & -\cos(4t\phi) & 0 \end{bmatrix}^T,$$

$$\kappa_{0,(i',4t,2),1,1}(r, \phi) = \omega_{R_{i'}}(r) \begin{bmatrix} \sin(4t\phi) & 0 & 0 & \sin(4t\phi) \end{bmatrix}^T,$$

$$\kappa_{0,(i',4t,2),2,1}(r, \phi) = \omega_{R_{i'}}(r) \begin{bmatrix} 0 & \sin(4t\phi) & -\sin(4t\phi) & 0 \end{bmatrix}^T,$$

$$\kappa_{2,(i',4t+2,1),1,1}(r, \phi) = \omega_{R_{i'}}(r) \begin{bmatrix} \cos((4t+2)\phi) & 0 & 0 & -\cos((4t+2)\phi) \end{bmatrix}^T,$$

$$\kappa_{2,(i',4t+2,1),2,1}(r, \phi) = \omega_{R_{i'}}(r) \begin{bmatrix} 0 & \cos((4t+2)\phi) & \cos((4t+2)\phi) & 0 \end{bmatrix}^T,$$

$$\kappa_{2,(i',4t+2,2),1,1}(r, \phi) = \omega_{R_{i'}}(r) \begin{bmatrix} \sin((4t+2)\phi) & 0 & 0 & -\sin((4t+2)\phi) \end{bmatrix}^T,$$

$$\kappa_{2,(i',4t+2,2),2,1}(r, \phi) = \omega_{R_{i'}}(r) \begin{bmatrix} 0 & \sin((4t+2)\phi) & \sin((4t+2)\phi) & 0 \end{bmatrix}^T$$

Fig 4 shows the basis kernels for $l = 0$ and $t = 0$. For simplicity, in Fig 4, we use $k \in \mathbb{Z}$ rather that $t \in \mathbb{N}$ to index the basis; this difference emerges from the fact that $\rho_{j'}$ is isomorphic to $\rho_1$ when restricted from SO(2) to $C_4$ for any $j' = |1 + 4k|$, but $k > 0$ and $k < 0$ require a different change of basis (see $4t + 1$ and $4t + 3$ above). Observe also that these basis kernels span the same space of the ones in Table 11 of Weiler & Cesa (2019), for $N = 4$, up to vectorization.

## C  REAL VALUED REPRESENTATIONS

In this section, we discuss the representation theory of compact groups on the *real* field $\mathbb{R}$.

Usually in representation theory, one assumes *complex*-valued signals in $L^2_{\mathbb{C}}(G)$ and complex representations of $G$. However, in this work, we implement equivariant CNNs using real representations, and therefore must discuss general issues arising in this case.

When using real representations, the theory becomes more difficult: while the matrix coefficients of the irreducible representation span the space of real square-integrable functions $L^2_{\mathbb{R}}(G)$, they are not necessarily orthogonal to each other anymore. More precisely, the matrix coefficients belonging to the same irrep might be linearly dependent, while different irreps still contain orthogonal coefficients. This is a consequence of the fact that the strong version of *Schur's Lemma* stating that the endomorphism space of irreps are one dimensional does not hold for representations over $\mathbb{R}$.

As before, we freely make use of the preliminaries from Section A. In particular, recall that for real representations, we talk about *orthogonal* representations instead of unitary representations, and such representations take values in the *orthogonal group* $\mathrm{O}(V)$ of some real Hilbert-space $V$.

In Section C.1, we introduce some properties of real irreducible representations and characterize their endomorphism space. In Section C.2, we show that any real irrep can always be expressed with respect to a convenient basis such that a set of linearly independent matrix coefficients are contained in a subset of the columns of the irrep. This will be particularly useful to construct real harmonic bases in Section D and, in particular, in Corollary 5. Additionally, we describe a method to find such a basis for an arbitrary real irrep. Finally, in Section C.3, we show an example of such basis, which makes the result more intuitive. Additionally, we use this specific basis to simplify computations in Section F.

### C.1  PRELIMINARIES ON THE STRUCTURE OF $\mathrm{End}_{G,\mathbb{R}}(V_\psi)$

Fix an orthogonal irreducible real representation $\psi : G \to \mathrm{O}(V_\psi)$ of the compact group $G$. Let $\mathrm{End}_{G,\mathbb{R}}(V_\psi)$ be its endomorphism algebra. It is well-known that this endomorphism algebra is isomorphic (in a sense which will be made precise in the following proposition) to one of the division algebras $\mathbb{R}$, $\mathbb{C}$, and $\mathbb{H}$, i.e., the real numbers, complex numbers, or quaternions, see Bröcker & Dieck (2003), Theorem 6.7. Let $\mathbb{K} \cong \mathrm{End}_{G,\mathbb{R}}(V_\psi)$ be this division algebra. Let $E_\psi = \dim \mathrm{End}_{G,\mathbb{R}}(V_\psi)$ its dimension. Let $i_1 = 1$, $i_2 = i$, $i_3 = j$ and $i_4 = k$ with $1, i, j, k \in \mathbb{H}$ the usual basis. It has the characterizing properties $i^2 = j^2 = k^2 = -1$, that $1$ is a multiplicative identity, and that $ij = k = -ji$, $jk = i = -kj$, and $ki = j = -ik$. Then $\mathbb{K}$ has as an orthonormal basis $\{i_l\}_{l=1}^{E_\psi}$. We now make the properties of the isomorphism precise:

**Proposition C.1.** *There is $\mathbb{K} \in \{\mathbb{R}, \mathbb{C}, \mathbb{H}\}$ such that there is a function $\Phi : \mathbb{K} \to \mathrm{End}_{G,\mathbb{R}}(V_\psi)$ with the following properties:*

1.  *$\Phi(x + y) = \Phi(x) + \Phi(y)$ for all $x, y \in \mathbb{K}$.*

2.  *$\Phi(\lambda x) = \lambda \Phi(x)$ for all $\lambda \in \mathbb{R}$ and $x \in \mathbb{K}$.*

3.  *$\Phi(xy) = \Phi(x) \circ \Phi(y)$ for all $x, y \in \mathbb{K}$.*

4.  *$\Phi(1) = \mathrm{id}_{V_\psi}$.*

5.  *$\Phi$ is bijective.*

6.  *$\Phi(i_k) \in \mathrm{O}(V_\psi)$ for all $k \in \{1, \ldots, E_\psi\}$.*

*Proof.* Properties 1 to 5 are the properties of an $\mathbb{R}$-algebra isomorphism. Such an $\mathbb{R}$-algebra isomorphism $\Phi$ exists by Bröcker & Dieck (2003), Theorem 6.7.

We now show that this automatically already implies property 6 as well: $\Phi(i_l) \in \mathrm{End}_{G,\mathbb{R}}(V_\psi)$ is a non-zero endomorphism, and thus, by Schur's Lemma for orthogonal representations (see Lang & Weiler (2020), Lemma B.29), there exists a coefficient $\mu_l \in \mathbb{R}$ such that $\mu_l \Phi(i_l) \in \mathrm{O}(V_\psi)$. We

deduce:

$$\mu_l \Phi(i_l) \circ \mu_l \Phi(i_l) = \mu_l^2 \Phi(i_l \cdot i_l)$$
$$= \pm \mu_l^2 \Phi(1)$$
$$= \pm \mu_l^2 \operatorname{id}_{V_\psi}.$$

Together with $\mu_l \Phi(i_l) \circ \mu_l \Phi(i_l) \in O(V_\psi)$ we necessarily have $\mu_l^2 \in \{\pm 1\}$, and thus $\mu_l = \pm 1$.

Thus, $\Phi(i_l) = \pm \mu_l \Phi(i_l) \in O(V_l)$, which shows 6. $\qquad \square$

**Definition C.2.** Let $\psi$ be a real irrep. Depending on its endomorphism algebra $\operatorname{End}_G(\psi)$, $\psi$ is classified in one of the following three categories:

- *real type*: if $\operatorname{End}_G(V_\psi) \cong \mathbb{R}$

- *complex type*: if $\operatorname{End}_G(V_\psi) \cong \mathbb{C}$

- *quaternionic type*: if $\operatorname{End}_G(V_\psi) \cong \mathbb{H}$

From now on, fix the isomorphism $\Phi : \mathbb{K} \to \operatorname{End}_{G,\mathbb{R}}(V_\psi)$ and denote $I_l := \Phi(i_l)$.

For simplifying the notation, we set $d_\psi := \dim_\psi := \dim V_\psi$. From now on, we assume for simplicity that $V_\psi = \mathbb{R}^{d_\psi}$, which holds up to isomorphism. The scalar product on $V_\psi$ is then just given by $\langle v \mid w \rangle = v^T w$, and $V_\psi$ has the standard basis $\{e_1, \ldots, e_{d_\psi}\}$ with $e_i$ having a 1 at position $i$ and being zero elsewhere.

For any $A \in \operatorname{End}_{G,\mathbb{R}}(V_\psi)$, we define $A^T \in \operatorname{End}_{G,\mathbb{R}}(V_\psi)$ as the transpose, which is also the unique matrix and linear function $A^T : V_\psi \to V_\psi$ such that $\langle Av \mid w \rangle = \langle v \mid A^T w \rangle$ for all $v, w \in V_\psi$. That $A^T$ is also an endomorphism is for the fact that $\psi$ is an *orthogonal* representation.

We define the following inner product on $\operatorname{End}_{G,\mathbb{R}}(V_\psi)$:

$$\langle A \mid B \rangle = \frac{1}{d_\psi} \operatorname{Tr}(AB^T), \tag{16}$$

where $\operatorname{Tr}$ is the trace of a matrix, given by the sum of diagonal elements. The trace is linear and continuous and has the property $\operatorname{Tr}(AB) = \operatorname{Tr}(BA)$, which results in the trace of a matrix being independent of the basis in which it is expressed.

**Lemma C.3.** *We have the following:*

1. *$I_l = -I_l^{-1}$ for $l > 1$.*

2. *$[I_l]_{ii} = 0$ for $l > 1$ and $i \in \{1, \ldots, d_\psi\}$.*

3. *$\langle I_l \mid I_{l'} \rangle = \delta_{ll'}$ for all $l$.*

*Proof.*     1. This follows from $I_l \circ I_l = \Phi(i_l^2) = -\Phi(1) = -\operatorname{id}_{V_\psi}$ for all $l > 1$.

2. From Proposition C.1 we know that $I_l \in O(V_\psi)$ for all $l$. So, $I_l^{-1} = I_l^T$. For $l > 1$, we deduce $I_l = -I_l^T$ from 1, i.e., $I_l + I_l^T = 0$. It follows $[I_l]_{ii} = \frac{1}{2}[I_l + I_l^T]_{ii} = 0$.

3. Note that there is some $l'' \in \{1, \ldots, E_\psi\}$ such that the following holds:

$$\langle I_l \mid I_{l'} \rangle = \frac{1}{d_\psi} \operatorname{Tr}\left(I_l I_{l'}^T\right)$$
$$= \frac{1}{d_\psi} \operatorname{Tr}\left(\Phi(i_l) \cdot \Phi(i_{l'})^{-1}\right)$$
$$= \frac{1}{d_\psi} \operatorname{Tr}\left(\Phi(i_l \cdot i_{l'}^{-1})\right)$$
$$= \frac{1}{d_\psi} \operatorname{Tr}\left(\Phi(\pm i_{l''})\right)$$
$$= \frac{1}{d_\psi} \operatorname{Tr}\left(\pm I_{l''}\right)$$

Thereby, we made use of $I_{l'} \in O(V_\psi)$ in step 2, and used the multiplication properties of the quaternions in the second to last step. Note that $i_{l''} = 1$ if and only if $l = l'$, and that in this case, we have $I_{l''} = \Phi(i_{l''}) = \mathrm{id}_{V_\psi}$, leading to $\langle I_l | I_{l'} \rangle = 1$. If, however, $l'' > 1$ (i.e., $l \neq l'$), then we have $[I_{l''}]_{ii} = 0$ for all $i$ by part 2, and thus we get a zero trace, proving the result. $\qquad\square$

**Corollary 2.** *The function* $\Phi : \mathbb{K} \rightarrow \mathrm{End}_{G,\mathbb{R}}(V_\psi)$ *is a* unitary *isomorphism, meaning: When considering* $\mathbb{K}$ *as an* $\mathbb{R}$-*vector space with standard inner product* $\langle \cdot | \cdot \rangle$ *and considering* $\mathrm{End}_{G,\mathbb{R}}(V_\psi)$ *as an* $\mathbb{R}$-*vector space with inner product given by eq. 16, then we have for all* $x, y \in \mathbb{K}$

$$\langle \Phi(x) | \Phi(y) \rangle = \langle x | y \rangle.$$

*Furthermore,* $\{I_1, \ldots, I_{E_\psi}\}$ *forms an orthonormal basis of* $\mathrm{End}_{G,\mathbb{R}}(V_\psi)$.

*Proof.* First, note that eq. 16 clearly defines a positive-definite inner product on $\mathrm{End}_{G,\mathbb{R}}(V_\psi)$. Write $x = \sum_{l=1}^{E_\psi} x_l i_l$ and $y = \sum_{l'=1}^{E_\psi} y_{l'} i_{l'}$. Using Lemma C.3, part 3, we obtain:

$$\begin{aligned}
\langle \Phi(x) | \Phi(y) \rangle &= \sum_{l=1}^{E_\psi} \sum_{l'=1}^{E_\psi} x_l y_{l'} \langle I_l | I_{l'} \rangle \\
&= \sum_{l=1}^{E_\psi} x_l y_l \\
&= \langle x | y \rangle
\end{aligned}$$

The last statement that the $I_l$ form an orthonormal basis of $\mathrm{End}_{G,\mathbb{R}}(V_\psi)$ follows from Lemma C.3 part 3 as well, together with the fact that $\Phi(i_l) = I_l$ and that $\Phi$ is an isomorphism. $\qquad\square$

For $x = \sum_{l=1}^{E_\psi} x_l i_l \in \mathbb{K}$, the conjugation $\overline{x}$ is defined by

$$\overline{x} = x_1 - \sum_{l=2}^{E_\psi} x_l i_l.$$

Furthermore, we define the norm $\|W\| := \sqrt{\langle W | W \rangle}$ for $W \in \mathrm{End}_{G,\mathbb{R}}(V_\psi)$.

**Proposition C.4.** *We have the following:*

1. *For $x \in \mathbb{K}$, we have $\Phi(\overline{x}) = \Phi(x)^T$.*

2. *For any $W \in \mathrm{End}_{G,\mathbb{R}}(V_\psi)$, we have $WW^T = \|W\|^2 \cdot \mathrm{id}_{V_\psi}$.*

*Proof.*      1. We have

$$\begin{aligned}
\Phi(\overline{x}) &= \Phi\left( x_1 i_1 - \sum_{l=2}^{E_\psi} x_l i_l \right) \\
&= x_1 I_1 - \sum_{l=2}^{E_\psi} x_l I_l \\
&= x_1 I_1^T + \sum_{l=2}^{E_\psi} x_l I_l^T \\
&= \left( \sum_{l=1}^{E_\psi} x_l I_l \right)^T \\
&= \left( \Phi\left( \sum_{l=1}^{E_\psi} x_l i_l \right) \right)^T \\
&= \Phi(x)^T.
\end{aligned}$$

In the third step, we used $I_1 = \mathrm{id}_{V_\psi}$ and $I_l^T = -I_l$, which was shown in Lemma C.3 together with $I_l \in O(V_l)$.

2. This follows from the well-known fact that $x\overline{x} = \|x\|^2$ for $x \in \mathbb{K}$ and part 1.

$\square$

## C.2 ORTHOGONALITY AND ALIGNMENT OF MATRIX COEFFICIENTS

The investigations here on the orthogonality relations of matrix coefficients of irreducible real representations generalize what's presented in Knapp (2002) for complex representations to real representations. The results thereby become a bit more involved.

Remember that $V_\psi = \mathbb{R}^{d_\psi}$ has the standard basis $\{e_1, \ldots, e_{d_\psi}\}$. Define $\boldsymbol{E}_{ij} \coloneqq \boldsymbol{e}_i \boldsymbol{e}_j^T \in \mathbb{R}^{d_\psi \times d_\psi}$. Furthermore, define

$$\boldsymbol{O}_{ij} \coloneqq \int_G \psi(g) \boldsymbol{E}_{ij} \psi(g)^{-1} \mathrm{d}g = \int_G \psi(g) \boldsymbol{E}_{ij} \psi(g)^T \mathrm{d}g.$$

Remember that $\psi : G \to \mathrm{O}(V_\psi) = \mathrm{O}(\mathbb{R}^{d_\psi}) \subseteq \mathbb{R}^{d_\psi \times d_\psi}$ takes values in $d_\psi \times d_\psi$-matrices. For $k, j \in \{1, \ldots, d_\psi\}$, define $\psi_{kj} : G \to \mathbb{R}$ as the function $\psi_{kj} \in L^2_{\mathbb{R}}(G)$ given by $\psi_{kj}(g) \coloneqq \psi(g)_{kj}$. Remember that we have a scalar product on $L^2_{\mathbb{R}}(G)$ given by $\langle f \mid h \rangle \coloneqq \int_G f(g) h(g) \mathrm{d}g$, which means we can evaluate also scalar products of different matrix coefficients.

**Lemma C.5.** *We have the following:*

1. $\boldsymbol{O}_{ij} \in \mathrm{End}_{G,\mathbb{R}}(V_\psi)$.

2. $\boldsymbol{O}_{ij} = \frac{1}{d_\psi} \sum_{l=1}^{E_\psi} [I_l]_{ij} \cdot I_l$ .

3. $\boldsymbol{O}_{ij} = \left( \langle \psi_{k_1 i} \mid \psi_{k_2 j} \rangle \right)_{k_1, k_2 = 1}^{d_\psi}$.

*Proof.* We prove the three statements as follows:

1. From the integration-measure on $G$ being a Haar measure, i.e., left-invariant, it can easily be checked that $\boldsymbol{O}_{ij} \psi(g') = \psi(g') \boldsymbol{O}_{ij}$ for all $g' \in G$.

2. From $\boldsymbol{O}_{ij} \in \mathrm{End}_{G,\mathbb{R}}(V_\psi)$ and the fact that the $I_l$ form an orthonormal basis of the space $\mathrm{End}_{G,\mathbb{R}}(V_\psi)$ by Corollary 2, we obtain $\boldsymbol{O}_{ij} = \sum_{l=1}^{E_\psi} \langle I_l \mid \boldsymbol{O}_{ij} \rangle I_l$. We now determine the coefficients:

$$\begin{aligned}
\langle I_l \mid \boldsymbol{O}_{ij} \rangle &= \langle \boldsymbol{O}_{ij} \mid I_l \rangle \\
&= \frac{1}{d_\psi} \mathrm{Tr} \left( \boldsymbol{O}_{ij} I_l^T \right) \\
&= \frac{1}{d_\psi} \mathrm{Tr} \left( \int_G \psi(g) \boldsymbol{E}_{ij} \psi(g)^{-1} I_l^T \mathrm{d}g \right) \\
&\overset{(1)}{=} \frac{1}{d_\psi} \int_G \mathrm{Tr} \left( \psi(g) \boldsymbol{E}_{ij} I_l^T \psi(g)^{-1} \right) \mathrm{d}g \\
&\overset{(2)}{=} \frac{1}{d_\psi} \int_G \mathrm{Tr} \left( \boldsymbol{E}_{ij} I_l^T \right) \mathrm{d}g \\
&= \frac{1}{d_\psi} \mathrm{Tr} \left( \boldsymbol{E}_{ij} I_l^T \right) \\
&= \frac{1}{d_\psi} [I_l]_{ij}
\end{aligned}$$

In (1), we use that $I_l$ is an endomorphism, i.e., commutes with $\psi(g)$. Additionally, we use that $\mathrm{Tr}$ is linear and continuous, and thus commutes with integrals. Step (2) uses that the trace satisfies $\mathrm{Tr}(AB) = \mathrm{Tr}(BA)$. In the last step, we make use of $\boldsymbol{E}_{ij} = \boldsymbol{e}_i \boldsymbol{e}_j^T$.

3. We have

$$
\begin{aligned}
\boldsymbol{O}_{ij} &= \int_G \psi(g) \boldsymbol{E}_{ij} \psi(g)^{-1} \mathrm{d}g \\
&= \int_G \psi(g) \boldsymbol{e}_i \boldsymbol{e}_j^T \psi(g)^T \mathrm{d}g \\
&= \int_G \begin{pmatrix} \psi_{1i}(g) \\ \vdots \\ \psi_{d_\psi i}(g) \end{pmatrix} \cdot \begin{pmatrix} \psi_{1j}(g) & \cdots & \psi_{d_\psi j}(g) \end{pmatrix} \mathrm{d}g \\
&= \int_G \left( \psi_{k_1 i}(g) \psi_{k_2 j}(g) \right)_{k_1,k_2=1}^{d_\psi} \mathrm{d}g \\
&= \left( \langle \psi_{k_1 i} \mid \psi_{k_2 j} \rangle \right)_{k_1,k_2=1}^{d_\psi}.
\end{aligned}
$$

$\square$

**Corollary 3.** *We have* $\langle \psi_{k_1 j} \mid \psi_{k_2 j} \rangle = \frac{\delta_{k_1 k_2}}{d_\psi}$ *for all* $k_1, k_2, j \in \{1, \ldots, d_\psi\}$. *In other words, the matrix coefficients in the same column are orthogonal to each other, and they are normalized up to a constant factor:* $\|\sqrt{d_\psi} \cdot \psi_{kj}\| = 1$.

*Proof.* Remember that in Lemma C.3 we showed $[I_l]_{jj} = 0$ for $l > 1$ and that $I_1 = \mathrm{id}_{V_\psi}$. Together with Lemma C.5, we obtain:

$$
\begin{aligned}
\langle \psi_{k_1 j} \mid \psi_{k_2 j} \rangle &= [\boldsymbol{O}_{jj}]_{k_1 k_2} \\
&= \left[ \frac{1}{d_\psi} \sum_{l=1}^{E_\psi} [I_l]_{jj} \cdot I_l \right]_{k_1 k_2} \\
&= \frac{1}{d_\psi} [I_1]_{jj} \cdot [I_1]_{k_1 k_2} \\
&= \frac{\delta_{k_1 k_2}}{d_\psi}.
\end{aligned}
$$

The statement about the normalization is clear. $\square$

**Lemma C.6.** *For any* $p, q \in \{1, \ldots, E_\psi\}$ *and for any* $\boldsymbol{v} \in V_\psi$, $\langle I_p \boldsymbol{v} \mid I_q \boldsymbol{v} \rangle = \delta_{pq} \langle \boldsymbol{v} \mid \boldsymbol{v} \rangle$.

*Proof.* Note that there is a $l$ such that $I_p^T I_q = \pm I_l$. If $p = q$, $I_p^T I_p = I_1 = \mathrm{id}_{V_\psi}$ is the identity matrix, hence $\boldsymbol{v}^T I_p^T I_q \boldsymbol{v} = \boldsymbol{v}^T \boldsymbol{v}$. Otherwise, $l > 1$ and $I_l = -I_l^T$ is a skew symmetric matrix, hence $\boldsymbol{v}^T I_l \boldsymbol{v} = 0$. Indeed, note that, for any matrix $A$, it holds that $\boldsymbol{v}^T A \boldsymbol{v} = \boldsymbol{v}^T A^T \boldsymbol{v}$. If $A$ is a skew symmetric matrix ($A^T = -A$), it follows that $\boldsymbol{v}^T A \boldsymbol{v} = 0$. $\square$

**Lemma C.7.** *For each column* $j$, *the set of matrices* $\{\boldsymbol{O}_{ij}\}_i$ *spans the space* $\mathrm{End}_G(V_\psi)$, *which is an* $E_\psi$ *dimensional space.*

*Proof.* To prove this, it is sufficient to look at the set of coefficient vectors of each of these matrices when expressed with respect to the basis $\{I_l\}_l$ of $\mathrm{End}_G(V_\psi)$. If this set of coefficient vectors spans an $E_\psi$ dimensional space, the proof is complete. Because $\boldsymbol{O}_{ij} = \frac{1}{d_\psi} \sum_l [I_l]_{ij} I_l$, let's consider the set of vectors $\{([I_l]_{ij})_l\}_{i=1}^{d_\psi}$ and stack them in the rows of a matrix $M \in \mathbb{R}^{d_\psi \times E_\psi}$. The set of vectors spans an $E_\psi$ dimensional space if and only if $M$ has row rank or, equivalently, column rank $E_\psi$. Because $M$ has $E_\psi$ nonzero columns (namely, the $l$'th column of $M$ is the $j$'th column of $I_l$, which is non-zero since $I_l$ is invertible), then it has column rank equal to $E_\psi$ if and only if all its columns are pairwise orthogonal.

The orthogonality of the columns of $M$ follows from Lemma C.6. Indeed, note that the $l$-th column of $M$ contains the $j$-th column of $I_l$. By choosing $\boldsymbol{v} = \boldsymbol{e}_j$ in Lemma C.6, it follows that the column $j$ of two different basis elements $I_{l_1}$ and $I_{l_2}$ are orthogonal. Thus, the columns of $M$ are orthogonal. $\square$

From Lemma C.7, and since $\mathrm{End}_G(V_\psi)$ contains only the zero-map and isomorphisms, it follows that, for any $j$, the set of vectors $\{\boldsymbol{O}_{ij}\boldsymbol{e}_j\}_{i=1}^{d_\psi}$ spans an $E_\psi$ dimensional vector space. We will show that this means that there is a change of basis $B \in \mathbb{R}^{d_\psi \times d_\psi}$ and a corresponding irrep $\widetilde{\psi} = B^T \psi B$ isomorphic to $\psi$ such that, for any $j$, $\{\widetilde{\boldsymbol{O}_{ij}}\boldsymbol{e}_j\}_{i=1}^{d_\psi}$ contains only $E_\psi$ non-zero vectors, where $\widetilde{\boldsymbol{O}_{ij}} \coloneqq \int_G \widetilde{\psi}(g)\boldsymbol{E}_{ij}\widetilde{\psi}(g)^T \mathrm{d}g$.

We give a constructive proof now. First, we build one such matrix $B$ starting from the basis elements $\{I_l\}_{l=1}^{E_\psi}$. Then, we show that for $\widetilde{\psi} = B^T \psi B$, for each $j$, the matrix $\widetilde{\boldsymbol{O}_{ij}}$ is non-zero precisely for $E_\psi$ different values of $i$.

**Lemma C.8.** *For any vectors $\boldsymbol{v}, \boldsymbol{w} \in \mathbb{R}^{d_\psi}$, if $\langle I_p \boldsymbol{v} | \boldsymbol{w} \rangle = 0$ for all $p$, then $\langle I_p \boldsymbol{v} | I_q \boldsymbol{w} \rangle = 0$ for any $p$ and $q$.*

*Proof.* For any $p$ and $q$, there exists $l$ such that $I_p I_q^T = \pm I_l$. Then $\langle I_p \boldsymbol{v} | I_q \boldsymbol{w} \rangle = \boldsymbol{v}^T I_p^T I_q \boldsymbol{w} = \pm \boldsymbol{v}^T I_l^T \boldsymbol{w} = \pm \langle I_l \boldsymbol{v} | \boldsymbol{w} \rangle$ which is zero by assumption. $\square$

To construct the matrix $B$, we use the following algorithm. Let $\mathcal{B} = \emptyset \subseteq \mathbb{R}^{d_\psi}$ the empty set. Repeat the following process $d_\psi / E_\psi$ times:[15]

Take a normalized vector $\boldsymbol{v} \in \mathbb{R}^{d_\psi}$ which is orthogonal to all vectors in $\mathcal{B}$. Add the vectors $\{I_l \boldsymbol{v}\}_{l=1}^{E_\psi}$ to $\mathcal{B}$. Note that all vectors $\{I_l \boldsymbol{v}\}_{l=1}^{E_\psi}$ are orthogonal to each other because of Lemma C.6. Note that if $\mathcal{B}$ contains the vectors $\{I_l \boldsymbol{w}\}_l$ and we choose a vector $\boldsymbol{v}$ such that it is orthogonal to all $\{I_l \boldsymbol{w}\}_l$, then, due to Lemma C.8, the updated set $\mathcal{B}$ still contains only pairwise orthogonal vectors.

The matrix $B$ is built by stacking horizontally the vectors in $\mathcal{B}$. Note that since all $v$ in the process are normalized, and the $I_l v$ are normalized as well by Lemma C.6, the matrix is orthogonal, i.e. $B^T B = B B^T = \mathrm{id}$.

By construction, for any $l$, the left multiplication of $B$ by $I_l$ only results in a permutation of the columns of $B$ (and, potentially, flipping their sign). In particular, $I_l$ only permutes the vectors in $B$ belonging to the same group $\{I_l \boldsymbol{v}\}_l$. Hence, there exists a permutation matrix $P_l$ (with, potentially, negative entries) such that $I_l B = B P_l$.

Let, as mentioned before, $\widetilde{\psi}(g) \coloneqq B^T \psi(g) B$, i.e., $\widetilde{\psi}$ is isomorphic to $\psi$ via $B$. Also, let $\widetilde{\boldsymbol{O}}_{ij} = \int_G \widetilde{\psi}(g)\boldsymbol{E}_{ij}\widetilde{\psi}(g)^T \mathrm{d}g$.

**Lemma C.9.** *Let $\psi$ be a real irrep and $B$ a matrix computed as above. Fix $j \in \{1, \ldots, d_\psi\}$. Then, the following holds:*

1. *Fix also $i \in \{1, \ldots, d_\psi\}$. Then $\widetilde{\boldsymbol{O}}_{ij} \neq 0$ if and only if there is a unique $l \in \{1, \ldots, E_\psi\}$ such that $[P_l]_{ij} \neq 0$. In that case, $\widetilde{\boldsymbol{O}}_{ij} = \frac{1}{d_\psi}[P_l]_{ij} P_l$.*

2. *The matrix $\widetilde{\boldsymbol{O}}_{ij}$ is non-zero for precisely $E_\psi$ different values of $i$.*

---

[15]The process itself will reveal that $d_\psi / E_\psi$ is a whole number.

*Proof.* 1. Let $B_i$ and $B_j$, respectively, the $i$-th and the $j$-th columns of $B$.

$$
\begin{aligned}
\widetilde{\boldsymbol{O}}_{ij} &= \int_G B^T \psi(g) B \boldsymbol{E}_{ij} B^T \psi(g)^T B \mathrm{d}g \\
&= B^T \left( \int_G \psi(g) B_i B_j^T \psi(g)^T \mathrm{d}g \right) B \\
&= B^T \left( \int_G \psi(g) \left( \sum_{ab} B_{ai} B_{bj} \boldsymbol{E}_{ab} \right) \psi(g)^T \mathrm{d}g \right) B \\
&= \sum_{a,b} B_{ai} B_{bj} B^T \boldsymbol{O}_{ab} B \\
&= \frac{1}{d_\psi} \sum_l \left( \sum_{a,b} B_{ai} B_{bj} [I_l]_{ab} \right) B^T I_l B \\
&= \frac{1}{d_\psi} \sum_l [B^T I_l B]_{ij} B^T I_l B \\
&= \frac{1}{d_\psi} \sum_l [P_l]_{ij} P_l,
\end{aligned}
$$

where in the last step we used that $B$ has the property $B^T I_l B = B^T B P_l = P_l$, where $P_l$ has the structure of a signed permutation matrix. Thus, if $\widetilde{O}_{ij} \neq 0$, then there exists $l$ such that $[P_l]_{ij} \neq 0$.

For the other direction, assume that there is such an $l$. We want to show that there cannot be another $k$ with $[P_k]_{ij} \neq 0$. If we can do that, then it follows $\widetilde{\boldsymbol{O}}_{ij} = \frac{1}{d_\psi} [P_l]_{ij} P_l \neq 0$ and we are done.

To do so, first observe that $\langle P_l \boldsymbol{e}_j | P_k \boldsymbol{e}_j \rangle = \delta_{lk}$. Indeed,

$$
\begin{aligned}
\langle P_l \boldsymbol{e}_j | P_k \boldsymbol{e}_j \rangle &= \boldsymbol{e}_j^T P_l^T P_k \boldsymbol{e}_j \\
&= \boldsymbol{e}_j^T B^T I_l^T B B^T I_k B \boldsymbol{e}_j \\
&= \boldsymbol{e}_j^T B^T I_l^T I_k B \boldsymbol{e}_j.
\end{aligned}
$$

Now, note that there exists a $m$ such that $I_l^T I_k = \pm I_m$. In particular, if $l = k$, $I_l^T I_l = I_1 = \mathrm{id}$ and, therefore, $\boldsymbol{e}_j^T B^T I_l^T I_k B \boldsymbol{e}_j = \boldsymbol{e}_j^T \boldsymbol{e}_j = 1$. Otherwise, $\boldsymbol{e}_j^T B^T I_l^T I_k B \boldsymbol{e}_j = \pm \boldsymbol{e}_j^T B^T I_m B \boldsymbol{e}_j = 0$, thanks to Lemma C.6.

Now, assume we have $[P_l]_{ij} \neq 0 \neq [P_k]_{ij}$. Then, since $P_l$ and $P_k$ are signed permutation matrices, all other entries in column $j$ are zero, meaning that

$$
0 \neq [P_l]_{ij} \cdot [P_k]_{ij} = \sum_i [P_l]_{ij} \cdot [P_k]_{ij} = \langle P_l \boldsymbol{e}_j \mid P_k \boldsymbol{e}_j \rangle = \delta_{kl},
$$

meaning that $k = l$.

2. For $l = 1, \ldots, E_\psi$, let $i_l \in \{1, \ldots, d_\psi\}$ be the unique index such that $[P_l]_{i_l j} \neq 0$. It follows $\widetilde{\boldsymbol{O}}_{i_l j} \neq 0$ from part 1. We also know from part 1 that all $i_l$ are pairwise different, so we found $E_\psi$ such indices. Now, assume $i$ is such that $\widetilde{\boldsymbol{O}}_{ij} \neq 0$. Then, from part 1 again, we get that there exists $l$ such that $[P_l]_{ij} \neq 0$, and thus $i = i_l$. That finishes the proof.

$\square$

As a last classical result, we will use the Cauchy-Schwartz inequality, which is well-known and therefore without a proof:

**Lemma C.10.** *Let $V$ be any Hilbert space with scalar product $\langle \cdot \mid \cdot \rangle$. Let $v, w \in V$. Then one has*

$$
|\langle v \mid w \rangle| \leq \|v\| \cdot \|w\|,
$$

*with equality if and only if $v$ and $w$ are linearly dependent.*

**Theorem C.11.** *Let $\psi$ be a real irrep. Then, there exists a real irrep $\widetilde{\psi}$, isomorphic to $\psi$, such that the following holds. Fix $j \in \{1, \ldots, d_\psi\}$:*

1. *The matrix coefficients in column $j$ of $\widetilde{\psi}$ are orthogonal to all matrix coefficients of $\widetilde{\psi}$ in all but precisely $E_\psi$ columns $i$.*

2. *If $i$ is one of the remaining $E_\psi$ columns, then $\widetilde{\psi}e_i$ is a signed permutation of $\widetilde{\psi}e_j$.*

*Proof.* Let $B$ and $\widetilde{\psi} = B^T \psi B$ as before. We know from Lemma C.9 that there are precisely $E_\psi$ values for $i$ such that $\widetilde{O}_{ij} \neq 0$.

1. Assume $\widetilde{O}_{ij} = 0$. Then, since by Lemma C.5 we have $\widetilde{O}_{ij} = \left( \langle \widetilde{\psi}_{k_1 i} | \widetilde{\psi}_{k_2 j} \rangle \right)_{k_1, k_2 = 1}^{d_\psi}$, it follows that columns $i$ and $j$ of $\widetilde{\psi}$ are fully orthogonal to each other.

2. Assume $\widetilde{O}_{ij} \neq 0$. Then, by Lemma C.9, there is $l$ such that $[P_l]_{ij} \neq 0$ and $\widetilde{O}_{ij} = \frac{1}{d_\psi} \cdot [P_l]_{ij} \cdot P_l$, where $P_l$ is a signed permutation matrix. We have $[P_l]_{ij} = \pm 1$, i.e., we can wlog. assume $[P_l]_{ij} = 1$, i.e., $\widetilde{O}_{ij} = \frac{1}{d_\psi} P_l$.

   Now, let $k_1 \in \{1, \ldots, d_\psi\}$ be arbitrary and look at the matrix coefficient $\widetilde{\psi}_{ik_1}$. Then, since $\frac{1}{d_\psi} P_l = \left( \langle \widetilde{\psi}_{k_1 i} | \widetilde{\psi}_{k_2 j} \rangle \right)_{k_1, k_2 = 1}^{d_\psi}$ by Lemma C.5 again, and due to the structure of $P_l$, there is precisely one $k_2 \in \{1, \ldots, d_\psi\}$ such that $\langle \widetilde{\psi}_{k_1 i} | \widetilde{\psi}_{k_2 j} \rangle = \pm \frac{1}{d_\psi}$. By the Cauchy-Schwartz inequality Lemma C.10 it follows:

   $$\frac{1}{d_\psi} = \left| \langle \widetilde{\psi}_{k_1 i} | \widetilde{\psi}_{k_2 j} \rangle \right| \leq \left\| \widetilde{\psi}_{k_1 i} \right\| \cdot \left\| \widetilde{\psi}_{k_2 j} \right\| = \frac{1}{\sqrt{d_\psi}} \cdot \frac{1}{\sqrt{d_\psi}} = \frac{1}{d_\psi}.$$

   In the second to last step, we thereby made use of Corollary 3. We obtain $\left| \langle \widetilde{\psi}_{k_1 i} | \widetilde{\psi}_{k_2 j} \rangle \right| = \left\| \widetilde{\psi}_{k_1 i} \right\| \cdot \left\| \widetilde{\psi}_{k_2 j} \right\|$ and thus, by the last statement in Lemma C.10, that $\widetilde{\psi}_{k_1 i}$ and $\widetilde{\psi}_{k_2 j}$ are linearly dependent. Since they have the same norm, it follows $\widetilde{\psi}_{k_1 i} = \pm \widetilde{\psi}_{k_2 j}$. Since for different $k_1$, the value of $k_2$ needs to differ as well due to $P_l$ being a signed permutation matrix, the result follows.

   $\square$

This result is convenient to define a real harmonic basis $\{Y_{ij}^m\}$ for $L_\mathbb{R}^2(G)$, which we then use in Corollary 5.

## C.3 A Convenient Choice of Basis for Real Irreps

At the end of the previous section, we described a method that, given a real irrep $\psi$ and a basis for $\mathrm{End}_G(V_\psi)$, generates an isomorphic irrep $\widetilde{\psi}$ which satisfies the convenient properties in Theorem C.11. To provide an intuition of what structure these real irreps have and to simplify some computations in Section F, in this section we describe a choice of basis to express each real irreducible representation, satisfying the properties in Theorem C.11. Note that this particular choice of basis *is not necessary* in most of our results, since we rely only on a generic orthonormal basis for $\mathrm{End}_G(V_\psi)$, but is useful to simplify Section F.

The content of this section is mostly based on Boardman (2007) and Bröcker & Dieck (2003). Recall first the classification of real irreps in *real*, *complex* or *quaternionic* type according to Definition C.2. We will rely on the following useful results:

**Lemma C.12.** *For any* real *type real irrep $\psi$, its complexification is isomorphic to a complex irrep $\sigma$, with $\sigma \cong \overline{\sigma}$. The complexification of $\psi$ is a representation like $\psi$ which acts on $\mathbb{C}^{d_\psi}$ rather than $\mathbb{R}^{d_\psi}$.*

and

**Lemma C.13.** *Any* complex *or* quaternionic *type real irrep $\psi$ is isomorphic to the* realification *of a complex irrep $\sigma$. In other words, for any complex or quaternionic type real irrep $\psi$, there exists a complex irrep $\sigma$ such that:*

$$\psi \cong \begin{bmatrix} \mathrm{Re}(\sigma) & -\mathrm{Im}(\sigma) \\ \mathrm{Im}(\sigma) & \mathrm{Re}(\sigma) \end{bmatrix} \cong \sigma \oplus \overline{\sigma}$$

*Additionally, if it has* quaternionic type*, it also holds that $\sigma \cong \overline{\sigma}$.*

Indeed, using $\mathrm{Re}(x) = \frac{1}{2}(x + \overline{x})$ and $\mathrm{Im}(x) = -\frac{1}{2}i(x - \overline{x})$, one can verify that:

$$\begin{bmatrix} \mathrm{Re}(\sigma) & -\mathrm{Im}(\sigma) \\ \mathrm{Im}(\sigma) & \mathrm{Re}(\sigma) \end{bmatrix} = D \begin{bmatrix} \sigma(g) & 0 \\ 0 & \overline{\sigma}(g) \end{bmatrix} D^{\dagger}$$

where

$$D = \frac{1}{\sqrt{2}} \begin{bmatrix} i\,\mathrm{id}_{d_\sigma} & -i\,\mathrm{id}_{d_\sigma} \\ \mathrm{id}_{d_\sigma} & \mathrm{id}_{d_\sigma} \end{bmatrix}$$

Using these results, we define a convenient choice of basis for $\psi$ for each possible irrep type. For each case, we also define an associated basis for $\mathrm{End}_G(V_\psi)$.

**Real type irrep** Since $\psi \cong \sigma$ (as complex irreps), we do not need further assumptions on the basis of $\psi$. Note also that $\mathrm{End}_G(V_\psi)$ contains only scalar multiples of the identity; hence, regardless of the choice of basis for $\psi$, a basis for $\mathrm{End}_G(V_\psi)$ is given by the identity matrix.

**Complex type irrep** If $\psi$ has complex type, we assume that $\psi$ is expressed in a basis such that $\psi = \begin{bmatrix} \mathrm{Re}(\sigma) & -\mathrm{Im}(\sigma) \\ \mathrm{Im}(\sigma) & \mathrm{Re}(\sigma) \end{bmatrix}$ for some complex irrep $\sigma$. Note that this is a natural choice, for example, for $G = \mathrm{SO}(2)$. It follows that a basis for $\mathrm{End}_G(V_\psi)$ is given by the following homomorphism $\Phi : \mathbb{C} \to \mathrm{End}_G(V_\psi)$:

$$\Phi(1) = \begin{bmatrix} \mathrm{id}_n & 0 \\ 0 & \mathrm{id}_n \end{bmatrix}$$

$$\Phi(i) = \begin{bmatrix} 0 & -\mathrm{id}_n \\ \mathrm{id}_n & 0 \end{bmatrix}$$

where $n = d_\psi/2$ and $\mathrm{id}_n$ is the identity matrix of size $n$.

**Quaternionic type irrep** Similarly, if $\psi$ has quaternionic type, we assume that $\psi$ is expressed in a basis such that $\psi = \begin{bmatrix} \mathrm{Re}(\sigma) & -\mathrm{Im}(\sigma) \\ \mathrm{Im}(\sigma) & \mathrm{Re}(\sigma) \end{bmatrix}$ for some complex irrep $\sigma$. One can show that if $\sigma$ has *quaternionic* type, it can always be expressed in a basis such that it has the following block structure:

$$\sigma(g) = \begin{bmatrix} \sigma_1(g) & -\overline{\sigma_2(g)} \\ \sigma_2(g) & \overline{\sigma_1(g)} \end{bmatrix} \tag{17}$$

and, therefore:

$$\psi(g) = \begin{bmatrix} \mathrm{Re}\left( \begin{bmatrix} \sigma_1(g) & -\overline{\sigma_2(g)} \\ \sigma_2(g) & \overline{\sigma_1(g)} \end{bmatrix} \right) & -\mathrm{Im}\left( \begin{bmatrix} \sigma_1(g) & -\overline{\sigma_2(g)} \\ \sigma_2(g) & \overline{\sigma_1(g)} \end{bmatrix} \right) \\ \mathrm{Im}\left( \begin{bmatrix} \sigma_1(g) & -\overline{\sigma_2(g)} \\ \sigma_2(g) & \overline{\sigma_1(g)} \end{bmatrix} \right) & \mathrm{Re}\left( \begin{bmatrix} \sigma_1(g) & -\overline{\sigma_2(g)} \\ \sigma_2(g) & \overline{\sigma_1(g)} \end{bmatrix} \right) \end{bmatrix}$$

A basis for $\text{End}_G(V_\psi)$ is given by the following homomorphism $\Phi : \mathbb{H} \to \text{End}_G(V_\psi)$:

$$\Phi(1) = \begin{bmatrix} \text{id}_n & 0 & 0 & 0 \\ 0 & \text{id}_n & 0 & 0 \\ 0 & 0 & \text{id}_n & 0 \\ 0 & 0 & 0 & \text{id}_n \end{bmatrix}$$

$$\Phi(i) = \begin{bmatrix} 0 & 0 & -\text{id}_n & 0 \\ 0 & 0 & 0 & -\text{id}_n \\ \text{id}_n & 0 & 0 & 0 \\ 0 & \text{id}_n & 0 & 0 \end{bmatrix}$$

$$\Phi(j) = \begin{bmatrix} 0 & -\text{id}_n & 0 & 0 \\ \text{id}_n & 0 & 0 & 0 \\ 0 & 0 & 0 & \text{id}_n \\ 0 & 0 & -\text{id}_n & 0 \end{bmatrix}$$

$$\Phi(k) = \begin{bmatrix} 0 & 0 & 0 & -\text{id}_n \\ 0 & 0 & \text{id}_n & 0 \\ 0 & -\text{id}_n & 0 & 0 \\ \text{id}_n & 0 & 0 & 0 \end{bmatrix}$$

where $n = d_\psi/4$.

# D   THE COMPUTATION OF THE INDUCED REPRESENTATION OF COMPACT GROUPS

*Quotient* and *induced feature fields*, i.e. features fields where the feature type $\rho$ is an *quotient* or *induced representation* (see Sec 2.1), were studied in Weiler & Cesa (2019) for the planar groups. These feature types encode square-integrable scalar or vector functions over a homogeneous space $X = G/H$ of $G$. If $X$ is not finite, one restricts the consideration to a subspace of *bandlimited* functions by using a finite subset of an harmonic basis for the space of all functions. A band-limited function is then parameterized by a coefficients vector, which carries an action of $G$ as a direc-sum of $G$ irreps. Given one such vector, the function it parametrizes can be sampled at any point of $x \in X$ by evaluating the harmonic basis on $x$. For example, in Sec.5.3, we employed models with quotient features over the sphere $\mathcal{S}^2 \cong \text{SO}(3)/\text{SO}(2)$ or the space $\text{Inv} \times \mathcal{S}^2 \cong \text{O}(3)/\text{SO}(2)$. See also Section H.3 for more details on how this is used in our neural network design.

Recall also that a $G$-orbit is isomorphic to an homogeneous space for $G$ and, therefore, to a quotient space $G/H = \{g.H | g \in G\}$, where $H < G$ is a stabilizer subgroup of the space; e.g. $\text{SO}(3).x \cong \mathcal{S}^2 \cong \text{SO}(3)/\text{SO}(2)$ for $x \in \mathbb{R}^3$ non-zero. The Wigner-Eckart theorem from Lang & Weiler (2020) relies on an *harmonic basis* over an orbit $G.x$ of the equivariance group $G$ inside the base-space $X$. While our Theorem B.7 relies on a $G$-*steerable basis* for $X$, a simple way (although, generally more complicated to band-limit and discretize) to generate such basis is by embedding $G$-orbits inside $X$ and use the harmonic bases of such orbits. This is visualized for $G = \text{C}_4$ in Fig. 1b and used for the $G = \text{I}$ baselines in Sec. 5.2

In all these settings, an harmonic basis for functions on an homogeneous space is required. In this section, we describe how to compute the harmonic basis for scalar and vector fields over a compact homogeneous space[16] $X \cong G/H$, directly from the matrix coefficients of the irreps of $G$. A vector field over $X = G/H$ is a function $f : X \to \mathbb{R}^{d_\psi}$ (or $\mathbb{C}^{d_\psi}$) and transforms under the action of $G$ according to an *induced representation* $\text{Ind}_H^G \psi$, where $\psi$ is an irrep of $H$. We give a precise definition of *induced representation* acting on such vector fields in Def. D.3. Note that scalar fields are considered a special case of vector fields. In particular, scalar fields on $G/H$ transform according to $\text{Ind}_H^G \psi_0$, where $\psi_0 : H \to \{1\}$ is the trivial representation of $H$. We occasionally call this representation acting on scalar fields a *quotient representation*.

Similar investigations are well-documented for complex representations: for the case $X = G$, a *complex* harmonic basis is given by the matrix coefficients of its irreps, as described by the Peter-

---

[16]**Note that in this section we use $X$ with a different meaning with respect to the rest of the paper.**

Weyl theorem A.6. A similar result for *complex-valued* representations and *scalar* functions on homogeneous spaces was also described in Kondor & Trivedi (2018).

**Outline and goal**    After some definitions and assumptions in Supplementary D.1, we provide different equivalent definitions of the induced representation in Supplementary D.2. Then, we construct the harmonic basis of induced representations in Supplementary D.3. Finally, Corollaries 4 and 5 provide the final harmonic bases for the complex and real settings, respectively.

We freely make use of the concepts defined in Supplementary A in this whole section.

### D.1    USEFUL DEFINITIONS, ASSUMPTIONS AND NOTATIONS

In this section, we will always assume a compact group $G$ and a compact subgroup $H \subseteq G$. The neutral elements of these groups are just denoted $e \in H$, which is the same for both groups. As before, for generality, we allow $\mathbb{K}$ to be any of the two fields $\mathbb{R}$ or $\mathbb{C}$ before we will later separately consider complex and real representations. $\widehat{H}$ is the set of all isomorphism classes of unitary irreps of $H$ and $\psi \in \widehat{H}$ is, by abuse of notation, meant to refer to a *representative* of such an isomorphism class. For the rest of the section, we fix such an irrep $\psi : G \to \mathrm{U}(V_\psi)$.

In order to generate a harmonic basis for the functions in the induced representation, we only rely on an harmonic basis for $L^2_{\mathbb{K}}(G)$, a basis for the endomorphism space of each irrep $\rho \in \hat{G}$ and the irrep decomposition of the irreps of $G$ when restricted to $H < G$. Assuming these are known, in the rest of this section we will use them to explicitly build a harmonic basis for functions transforming according to an induced representation.

### D.2    INDUCED REPRESENTATIONS

We can now introduce the *induced representation*. In this section, we will limit to the case where $\psi : H \to \mathrm{U}(V_\psi)$ is an irreducible representation of $H$. This is fixed in this whole section. Induction from non-irreducible representations of $H$ can be built by combining the induced representations of the $H$ irreps composing the reducible $H$-representation.

We now define the induced representation as a space of Mackey Functions, and then alternatively as a space of vector fields over a quotient space. The constructions are well-known and can for example be found in Taylor & Kaniutrh (2013). This book also contains more theoretical justifications, including the definition of the scalar product making the induced representation a Hilbert space and proofs that both versions of the representation are actually unitary.

**Definition D.1** (Mackey Functions). A *Mackey function* is a $\psi$-vector field $f : G \to V_\psi$ which has square-integrable component-functions, i.e., in $L^2_{\mathbb{K}}(G)$, and with the following equivariance-property: for all $h \in H$ and $g \in G$ one has

$$f(gh^{-1}) = \psi(h)f(g). \tag{18}$$

We denote the space of Mackey functions by $\mathrm{Hom}_H(G, V_\psi)$.[17]    This becomes a unitary $G$-representation with the action

$$[g.f](g') := f(g^{-1}g') . \tag{19}$$

*Remark* D.2. One can verify that $g.f$ still belongs to the space of Mackey functions. Indeed, for $g, g' \in G$, $h \in H$ and $f \in \mathrm{Hom}_H(G, V_\psi)$ one has

$$\begin{aligned}
[g.f](g'h^{-1}) &= f(g^{-1}(g'h^{-1})) \\
&= f((g^{-1}g')h^{-1}) \\
&= \psi(h)f(g^{-1}g') \\
&= \psi(h)[g.f](g').
\end{aligned}$$

Before we come to the definition as quotient vector fields, recall that for the pair of compact groups $H \subseteq G$, one can build the space of cosets

$$G/H := \{gH \mid g \in G\} . \tag{20}$$

---

[17]These functions can be viewed as left-$H$-equivariant when using the left action $h \star g := gh^{-1}$ of $H$ on $G$, which explains the notation.

There is a projection $\pi : G \to G/H$ given by $g \mapsto gH$ which induces a topology on $G/H$, making it a compact topological Hausdorff space. Together with the action $g.(g'H) := (gg')H$, $G/H$ becomes a homogeneous space of $G$, and thus carries a Haar measure.

Now, pick an arbitrary measurable *section* $\mathrm{r} : G/H \to G$ associating a representative element to each coset $gH \in G/H$.[18] In other words, the section $r$ satisfies $\pi \circ \mathrm{r} = \mathrm{id}_{G/H}$, with the projection as defined above. This can also be written as

$$\mathrm{r}(gH)H = gH, \quad \text{for all } g \in G. \tag{21}$$

Since $r(gH)H = gH$, we automatically have that

$$\mathrm{h}(g) := \mathrm{r}(gH)^{-1}g \in H, \quad \text{for all } g \in G. \tag{22}$$

Since $g = \mathrm{r}(gH) \cdot \mathrm{h}(g)$, it follows that every $g \in G$ is fully characterized by the pair $(gH, \mathrm{h}(g)) \in (G/H) \times H$. We fix from now on the definitions of the section $\mathrm{r} : G/H \to G$ and the map $\mathrm{h} : G \to H$, with the latter implicitly depending on $\mathrm{r}$.

**Definition D.3** ($\psi$-Vector Fields over the Quotient Space). A $\psi$-*vector field over the quotient space* $G/H$ is defined to be any function of the form

$$f' : G/H \to V_\psi$$

that has square-integrable component-functions with respect to the Haar measure on $G/H$, i.e., the component-functions are in $L^2_{\mathbb{K}}(G/H)$. The space of these $\psi$-vector fields is denoted by $\mathrm{Hom}(G/H, V_\psi)$. This becomes a unitary $G$-representation with the action

$$[g.f'](g'H) := \psi\Big(\mathrm{h}\left(g\,\mathrm{r}(g^{-1}g'H)\right)\Big)f'\left(g^{-1}g'H\right). \tag{23}$$

Note that the action of $G$ on $\mathrm{Hom}(G/H, V_\psi)$ depends on the choice of the (not necessarily continuous) section $\mathrm{r} : G/H \to G$ of the quotient space.

We now show that these two definitions are equivalent:

**Proposition D.4.** *There is an isomorphism of unitary representations*

$$\mathrm{Hom}_H(G, V_\psi) \underset{\overline{(\cdot)}}{\overset{\widetilde{(\cdot)}}{\rightleftarrows}} \mathrm{Hom}(G/H, V_\psi)$$

*between the space of Mackey functions on the left and the space of quotient vector fields on the right, given as follows:*

1. *For $f : G \to V_\psi$ a Mackey function, we define*
$$\widetilde{f}(gH) := f(\mathrm{r}(gH))$$
*using the section defined in eq. 21.*

2. *For a quotient vector field $f' : G/H \to V_\psi$, we define*
$$\overline{f'}(g) := \psi(\mathrm{h}(g)^{-1})f'(gH)$$
*using the function $\mathrm{h} : G \to H$ defined in eq. 22.*

*Proof.* In this proof, we freely make use of eq. 21 and eq. 22. First, we show the equivariance of $\widetilde{(\cdot)}$: Let $f : G \to V_\psi$ satisfy eq. 18. First, note that

$$
\begin{aligned}
\mathrm{h}\left(g\,\mathrm{r}\left(g^{-1}g'H\right)\right) &= \mathrm{r}\left(g\,\mathrm{r}\left(g^{-1}g'H\right)H\right)^{-1}g\,\mathrm{r}\left(g^{-1}g'H\right) \\
&= \mathrm{r}\left(gg^{-1}g'H\right)^{-1}g\,\mathrm{r}\left(g^{-1}g'H\right) \\
&= \mathrm{r}(g'H)^{-1}g\,\mathrm{r}\left(g^{-1}g'H\right) \\
&= \left(\mathrm{r}\left(g^{-1}g'H\right)^{-1}g^{-1}\mathrm{r}(g'H)\right)^{-1} \\
&= \mathrm{h}\left(g^{-1}\mathrm{r}(g'H)\right)^{-1}.
\end{aligned}
\tag{24}
$$

---

[18]This section can usually not be chosen to be continuous, but being measurable is enough for our purposes.

From this, we deduce:

$$\widetilde{g.f}(g'H) = [g.f]\big(\mathrm{r}(g'H)\big)$$
$$= f\big(g^{-1}\,\mathrm{r}(g'H)\big)$$
$$= f\Big(\mathrm{r}\big(g^{-1}\,\mathrm{r}(g'H)H\big)\,\mathrm{h}\big(g^{-1}\,\mathrm{r}(g'H)\big)\Big)$$
$$= f\Big(\mathrm{r}\big(g^{-1}g'H\big)\,\mathrm{h}\big(g^{-1}\,\mathrm{r}(g'H)\big)\Big)$$
$$= \psi\Big(\mathrm{h}\big(g^{-1}\,\mathrm{r}(g'H)\big)^{-1}\Big)\,f\big(\mathrm{r}(g^{-1}g'H)\big)$$
$$= \psi\Big(\mathrm{h}\big(g\,\mathrm{r}(g^{-1}g'H)\big)\Big)\,\widetilde{f}(g^{-1}g'H)$$
$$= [g.\widetilde{f}](g'H),$$

i.e., the desired equivariance $\widetilde{g.f} = g.\widetilde{f}$.

Given $f' : G/H \to V_\psi$, we first verify that $\overline{f'}$ satisfies eq. 18, i.e. that is lives in $\mathrm{Hom}_H(G, V_\psi)$. First, note that

$$\mathrm{h}\big(gh^{-1}\big) = \mathrm{r}\big(gh^{-1}H\big)^{-1}gh^{-1} \tag{25}$$
$$= \mathrm{r}\big(gH\big)^{-1}gh^{-1}$$
$$= \mathrm{h}(g)h^{-1}.$$

From this, we deduce:

$$\overline{f'}\big(gh^{-1}\big) = \psi\big(\mathrm{h}(gh^{-1})^{-1}\big)f'(gH)$$
$$= \psi\big(h\,\mathrm{h}(g)^{-1}\big)f'(gH)$$
$$= \psi(h)\psi(\mathrm{h}(g))f'(gH)$$
$$= \psi(h)\overline{f'}(g).$$

We now show the equivariance of $\overline{(\cdot)}$. We first note, using the result of eq. 24 and eq. 25:

$$\mathrm{h}(g')^{-1}\,\mathrm{h}\Big(g\,\mathrm{r}\big(g^{-1}g'H\big)\Big) = \mathrm{h}(g')^{-1}\,\mathrm{h}\Big(g^{-1}\,\mathrm{r}\big(g'H\big)\Big)^{-1}$$
$$= \Big[\mathrm{h}\Big(g^{-1}\,\mathrm{r}\big(g'H\big)\Big)\,\mathrm{h}(g')\Big]^{-1}$$
$$= \mathrm{h}\Big(g^{-1}\,\mathrm{r}\big(g'H\big)h(g')\Big)^{-1}$$
$$= \mathrm{h}\big(g^{-1}g'\big)^{-1}.$$

We deduce:

$$\overline{g.f'}(g') = \psi\big(\mathrm{h}(g')^{-1}\big)[g.f'](g'H)$$
$$= \psi\big(\mathrm{h}(g')^{-1}\big)\psi\Big(\mathrm{h}\big(g\,\mathrm{r}(g^{-1}g'H)\big)\Big)f'\big(g^{-1}g'H\big)$$
$$= \psi\big(\mathrm{h}(g^{-1}g')^{-1}\big)f'\big(g^{-1}g'H\big)$$
$$= \overline{f'}(g^{-1}g')$$
$$= [g.\overline{f'}](g'),$$

i.e., the desired equivariance $\overline{g.f'} = g.\overline{f'}$.

Finally, we need to show that $\overline{\widetilde{f}} = f$ and $\widetilde{\overline{f'}} = f'$, i.e., that the maps $\widetilde{(\cdot)}$ and $\overline{(\cdot)}$ are inverse to each other. We have

$$\overline{\widetilde{f}}(g) = \psi\big(\mathrm{h}(g)^{-1}\big)\widetilde{f}(gH)$$
$$= \psi\big(\mathrm{h}(g)^{-1}\big)f\big(\mathrm{r}(gH)\big)$$
$$= f\big(\mathrm{r}(gH)\,\mathrm{h}(g)\big)$$
$$= f(g)$$

and

$$\begin{aligned}
\widetilde{\overline{f'}}(gH) &= \overline{f'}\big(\mathrm{r}(gH)\big)\\
&= \psi\big(\mathrm{h}(\mathrm{r}(gH))^{-1}\big)f'\big(\mathrm{r}(gH)H\big)\\
&= f'\big(gH\big),
\end{aligned}$$

where we used $\mathrm{r}(gH)H = gH$, $\mathrm{h}\big(\mathrm{r}(gH)\big) = e$, and $\psi(e) = \mathrm{id}_{V_\psi}$ in the last step. More technical details, especially on the unitarity of the isomorphism, can be found in Taylor & Kaniutrh (2013). This finishes the proof. □

Proposition D.4 justifies the following definition:

**Definition D.5** (Induced Representation). For any irrep $\psi$ of $H \subseteq G$, we define the *induced representation* $\mathrm{Ind}_H^G \psi$ as the unitary $G$-representation acting on $\mathrm{Ind}_H^G V_\psi := \mathrm{Hom}_H(G, V_\psi) \cong \mathrm{Hom}(G/H, V_\psi)$ as in eq. 19 or eq. 23, depending on which representation space is chosen. We will always make the representation space (Mackey functions or quotient vector fields) explicit in the following.

### D.3 HARMONIC ANALYSIS OF INDUCED REPRESENTATIONS

Before we come to finding harmonic bases of induced representations, we must deal with a small technicality: in the induced representation given by Mackey functions, the equivariance property $f(gh^{-1}) = \psi(h)f(g)$ is used, which suggests that we look at the following left-action of $G$ on itself: $g \star g' := g'g^{-1}$, compared to the usual left action $g.g' := gg'$. Similarly to how we solved for a basis of the space of steerable kernels $K : X' \to \mathrm{Hom}_G(V_l, V_J)$ with our Wigner-Eckart theorem, Theorem B.5, by studying $L^2_{\mathbb{K}}(X')$, we want to find a basis of the space of Mackey functions $f : G \to V_\psi$ by studying $L^2_{\mathbb{K}}(G)$ – however, due to the adapted left-action on $G$, we also need to consider a different action on $L^2_{\mathbb{K}}(G)$.

Thus, define $L^2_{\mathbb{K}}(G)_2 := L^2_{\mathbb{K}}(G)$ together with the action $(g \star f)(g') := f(g'g)$, which can be shown to be a well-defined left-action on $L^2_{\mathbb{K}}(G)_2$. This is then a unitary representation. Fortunately, it turns out to be isomorphism to the standard regular representation:

**Lemma D.6.** *There is an isomorphism of unitary representations*

$$L^2_{\mathbb{K}}(G) \underset{\Phi}{\overset{\Phi}{\rightleftarrows}} L^2_{\mathbb{K}}(G)_2$$

*given in both directions by the same function:*

$$\big[\Phi(f)\big](g) := f(g^{-1}).$$

*Proof.* Since in $L^2_{\mathbb{K}}(G)$, we integrate over an inversion-invariant Haar measure, $\Phi$ is a unitary transformation. It is also clear that $\Phi$ is its own inverse. Thus, we are only left with checking the equivariance in one of the two directions:

$$\begin{aligned}
\big[\Phi(g.f)\big](g') &= [g.f]\big(g'^{-1}\big)\\
&= f\big(g^{-1}g'^{-1}\big)\\
&= f\big((g'g)^{-1}\big)\\
&= \big[\Phi(f)\big](g'g)\\
&= \big[g \star \Phi(f)\big](g').
\end{aligned}$$

It follows $\Phi(g.f) = g \star \Phi(f)$, i.e., the desired equivariance. □

We remind the reader of the following notation introduced in Section A, which we use in the harmonic analysis of the induced representation: $\widehat{G}$ is the set of isomorphism classes of unitary irreps of $G$. $m_j$ is the multiplicity of the irrep $\rho_j : G \to \mathrm{U}(V_j)$ in the regular representation $L^2_{\mathbb{K}}(G)$. For $j \le m_j$, $p_{ji} : L^2_{\mathbb{K}}(G) \to V_j$ is the corresponding harmonic projection. $[\psi j]$ is the multiplicity of the irrep

$\psi : H \to U(V_\psi)$ in $\operatorname{Res}_H^G V_j$. $E_\psi = \dim \operatorname{End}_{H,\mathbb{K}}(V_\psi)$ is the dimension of the endomorphism space of $V_\psi$. The $c_r^\psi : V_\psi \to V_\psi$, $r \le E_\psi$, form a basis of endomorphisms of $V_\psi$. $\operatorname{ID}_t^{\psi j} : \operatorname{Res}_H^G V_j \to V_\psi$ is an irrep decomposition projection. The $Y_{ji}^m : G \to \mathbb{K}$ form a harmonic basis of the regular representation $L_\mathbb{K}^2(G)$. $Y_{ji} : G \to \mathbb{K}^{d_j} \cong V_j$ is the column-vector of all the $Y_{ji}^m$. $\overline{Y_{ji}}$ is its complex conjugation.

Finally, note that after choosing bases, by abuse of notation we also refer with $c_r^\psi$ and $\operatorname{ID}_t^{\psi j}$ to the matrices corresponding to the linear maps.

**Theorem D.7.** *A basis of Mackey functions* $\operatorname{Ind}_H^G V_\psi = \operatorname{Hom}_H(G, V_\psi)$ *is given by*

$$\left\{ f_{jitr} : G \to V_\psi \cong \mathbb{K}^{d_\psi} \mid j \in \widehat{G}, i \le m_j, t \le [\psi j], r \le E_\psi \right\}.$$

*Thereby, we have*

$$f_{jitr}(g) = c_r^\psi \cdot \operatorname{ID}_t^{\psi j} \cdot \overline{Y_{ji}}\big(g^{-1}\big)$$

*Consequently, a basis of $\psi$-vector fields over the quotient space,* $\operatorname{Ind}_H^G V_\psi = \operatorname{Hom}(G/H, V_\psi)$*, is given by*

$$\left\{ \widetilde{f_{jitr}} : G/H \to V_\psi \cong \mathbb{K}^{d_\psi} \mid j \in \widehat{G}, i \le m_j, t \le [\psi j], r \le E_\psi \right\}$$

*with* $\widetilde{f_{jitr}}(gH) := f_{jitr}(\mathrm{r}(gH))$*, where* $\mathrm{r} : G/H \to G$ *is a measurable section.*

*Proof.* We have

$$\operatorname{Hom}_H(G, V_\psi) \overset{(1)}{\cong} \operatorname{Hom}_{H,\mathbb{K}}\big( \operatorname{Res}_H^G L_\mathbb{K}^2(G)_2, V_\psi \big)$$

$$\overset{(2)}{\cong} \operatorname{Hom}_{H,\mathbb{K}}\big( \operatorname{Res}_H^G L_\mathbb{K}^2(G), V_\psi \big)$$

$$\overset{(3)}{\cong} \overset{m_j}{\overbrace{\bigoplus_{j \in \widehat{G}} \bigoplus_{i=1}}} \operatorname{Hom}_{H,\mathbb{K}}\big( \operatorname{Res}_H^G V_j, V_\psi \big)$$

$$\overset{(4)}{\cong} \overset{m_j}{\overbrace{\bigoplus_{j \in \widehat{G}} \bigoplus_{i=1}}} \overset{[\psi j]}{\bigoplus_{t=1}} \operatorname{End}_{H,\mathbb{K}}(V_\psi).$$

Step (1) follows from Theorem B.2 with $G$ replaced by $H$, $G'$ replaced by $G$, $X'$ replaced by $G$ together with the left-action $g \star g' := g'g^{-1}$, and $\operatorname{Hom}_\mathbb{K}(V_l, V_J)$ replaced by $V_\psi$.[19] Step (2) follows from Lemma D.6. Step (3) is the Peter-Weyl Theorem A.6. And finally, step (4) uses the irrep decompositions $\operatorname{ID}_t^{\psi j} : \operatorname{Res}_H^G V_j \to V_\psi$.

Explicitly, from right to left the isomorphism is given by:

$$\big(c_{jit}\big)_{jit} \overset{(4)}{\mapsto} \left( \sum_{t=1}^{[\psi j]} c_{jit} \circ \operatorname{ID}_t^{\psi j} \right)_{ji}$$

$$\overset{(3)}{\mapsto} \sum_{j \in \widehat{G}} \sum_{i=1}^{m_j} \sum_{t=1}^{[\psi j]} c_{jit} \circ \operatorname{ID}_t^{\psi j} \circ p_{ji}$$

$$\overset{(2)}{\mapsto} \sum_{j \in \widehat{G}} \sum_{i=1}^{m_j} \sum_{t=1}^{[\psi j]} c_{jit} \circ \operatorname{ID}_t^{\psi j} \circ p_{ji} \circ \Phi$$

$$\overset{(1)}{\mapsto} \sum_{j \in \widehat{G}} \sum_{i=1}^{m_j} \sum_{t=1}^{[\psi j]} c_{jit} \circ \operatorname{ID}_t^{\psi j} \circ p_{ji} \circ \Phi|_G.$$

---

[19]Note that in the original theorem, the structure of the Hom-representation is not actually used, so we can just replace it by any finite-dimensional representation; in this case, $V_\psi$

Thus, a basis is given by all $f_{jitr} := c_r^\psi \circ \text{ID}_t^{\psi j} \circ p_{ji} \circ \Phi|_G$. The result now follows by expanding the dirac delta function $\delta_g$ in the harmonic basis:

$$
\begin{aligned}
p_{ji} \circ \Phi|_G(g) &= p_{ji}\big(\Phi(\delta_g)\big) \\
&= p_{ji}\big(\delta_{g^{-1}}\big) \\
&= \sum_{m=1}^{d_j} \langle Y_{ji}^m \mid \delta_{g^{-1}} \rangle \cdot p_{ji}\big(Y_{ji}^m\big) \\
&= \sum_{m=1}^{d_j} \overline{Y_{ji}^m}\big(g^{-1}\big) \cdot |jm\rangle \\
&= \overline{Y_{ji}}\big(g^{-1}\big).
\end{aligned}
$$

In the last step, we identified $V_j \cong \mathbb{K}^{d_j}$, and thus $|jm\rangle$ with a standard basis vector $e_m$. Note that, to be precise, one would need to approximate the Dirac delta function with square-integrable functions. The details of such arguments can be found in the proof of Theorem D.13 of Lang & Weiler (2020).

The last statement about a basis of $\text{Hom}(G/H, V_\psi)$, finally, follows from Proposition D.4. $\qquad\square$

**Corollary 4.** *An harmonic basis for* complex *Mackey functions is given by:*

$$
\left\{ f_{jit} : G \to V_\psi \cong \mathbb{C}^{d_\psi} \mid j \in \widehat{G}, i \le d_j, t \le [\psi j] \right\}.
$$

*where*

$$
f_{jit}(g) = \text{ID}_t^{\psi j} \cdot \sqrt{d_j} \rho_j(g^{-1}) e_i
$$

*Proof.* Using Proposition A.7, we find that $\overline{Y_{ji}}(g^{-1}) = \sqrt{d_j}\rho_j(g^{-1})e_i$. Because we assume *complex* representations, $\text{End}_{G,\mathbb{C}}(\psi)$ only contains scalar multiples of the identity, due to Schur's Lemma. In other words, $c_r^\psi$ is always the identity matrix. $\qquad\square$

**Corollary 5.** *Assume that all real irreps are defined over a basis as in Theorem C.11. An harmonic basis for* real *Mackey functions is given by:*

$$
\left\{ f_{jitr} : G \to V_\psi \cong \mathbb{R}^{d_\psi} \mid j \in \widehat{G}, i \le m_j = d_j/E_j, t \le [\psi j], r \le E_\psi \right\}.
$$

*where*

$$
f_{jitr}(g) = c_r^\psi \cdot \text{ID}_t^{\psi j} \cdot \sqrt{d_j} \rho_j(g^{-1}) e_{\mathcal{I}_i^j}
$$

$\mathcal{I}^j$ *is an index set containing the indexes of the* $m_j = d_j/E_j$ *columns of* $\rho_j$ *with linearly independent matrix coefficients.*

*Proof.* Using Proposition A.7, If $\rho_j$ is expressed in a basis as in Theorem C.11, the matrix coefficients of $\rho_j$ in a column $i$ are linearly dependent with those in $E_j$ columns of $\rho_j$. Therefore, it is sufficient to consider a subset of $m_j = d_j/E_j$ columns of $\rho_j$ which contain linearly independents matrix coefficients. Additionally, if $\rho_j$ is expressed in the basis described in Supplementary C.3, $\mathcal{I}^j = \{1, 2, \ldots, d_j/E_j\}$ so $\mathcal{I}_i^j = i$. $\qquad\square$

In Section D.5, we show that this basis is indeed harmonic and we explicitly derive a the action of $G$ on it as a direc-sum of irreps.

## D.4 RECONSTRUCTING THE IRREP DECOMPOSITION: THE COMPLEX CASE

The goal of this and the next section is to use write the action of $G$ on the bases found above explicitly as a direct sum of irreps of $G$. We first consider the simpler case of complex valued irreps.

Recall Corollary 4. Consider the set of all basis elements for a fixed value of $j$, i.e. $\{f_{jit}\}_{it}$ where:

$$
f_{jit}(g) = \text{ID}_t^{\psi j} \cdot \sqrt{d_j} \rho_j(g^{-1}) e_i
$$

Let $f$ be a Mackey function in the span of these basis elements, i.e. there exists $\{w_{it} \in \mathbb{C}\}_{it}$ such that:

$$
\begin{aligned}
f(g) &= \sum_{it} w_{it} f_{jit}(g) \\
&= \sum_{it} w_{it} \, \mathrm{ID}_t^{\psi t} \sqrt{d_j} \rho_j(g^{-1}) e_i \\
&= \sum_t \mathrm{ID}_t^{\psi t} \sqrt{d_j} \rho_j(g^{-1}) \sum_i w_{it} e_i \\
&= \sum_t \mathrm{ID}_t^{\psi t} \sqrt{d_j} \rho_j(g^{-1}) w_t
\end{aligned}
$$

where $[\boldsymbol{w}_t]_i = w_{it}$ and $\boldsymbol{w}_t \in \mathbb{C}^{d_j}$. Now, the action of $G$ on these functions is the usual action on $L^2(G)$, i.e. $\forall k, g \in G$:

$$
[k.f](g) = f(k^{-1}g) = \sum_{it} w_{it} f_{jit}(k^{-1}g) = \sum_t \mathrm{ID}_t^{\psi t} \sqrt{d_j} \rho_j(g^{-1}) \left(\rho_j(k) \boldsymbol{w}_t\right)
$$

In other words, the space of functions spanned by $\{f_{jit}\}_{it}$ is isomorphic to $\bigoplus_t^{[\psi j]} V_j$, and each $\boldsymbol{w}_t$ lives in one such $V_j$ space. We call the vectors $\boldsymbol{w}_t$ the *Fourier Transform coefficients* of the function $f$. More generally, let $f$ be a generic Mackey function in $\mathrm{Ind}_H^G V_\psi$. The matrix $\hat{f}(\rho_j)$ of shape $d_j \times [\psi j]$ contains in its columns the vectors $\boldsymbol{w}_t \in \mathbb{C}^{d_j}$ for each $t$. We call the collection $\{\hat{f}(\rho_j)\}_j$ the *Fourier Transform* of the function $f$. Note that if $\psi$ is the trivial representation, this definition recovers the classical definition of Fourier transform of functions over compact groups or homogeneous spaces. Finally, it follows that $\mathrm{Ind}_H^G \psi \cong \bigoplus_j \bigoplus_t^{[\psi j]} \rho_j$.

## D.5 Reconstructing the irrep decomposition: the real case

In the real case, this decomposition is less straightforward.

Recall Corollary 5. Consider the set of all basis elements for a fixed value of $j$, i.e. $\{f_{jitr}\}_{itr}$ where:

$$
f_{jitr}(g) = c_r^\psi \cdot \mathrm{ID}_t^{\psi j} \cdot \sqrt{d_j} \rho_j(g^{-1}) e_i
$$

with $i \le m_j = d_j / E_j$. Let $f$ be a Mackey function in the span of these basis elements, i.e. there exists $\{w_{itr} \in \mathbb{R}\}_{itr}$ such that:

$$
\begin{aligned}
f(g) &= \sum_{itr} w_{itr} f_{jitr}(g) \\
&= \sum_{itr} w_{itr} c_r^\psi \, \mathrm{ID}_t^{\psi t} \sqrt{d_j} \rho_j(g^{-1}) e_i \\
&= \sum_{tr} c_r^\psi \, \mathrm{ID}_t^{\psi t} \sqrt{d_j} \rho_j(g^{-1}) \sum_{i \le m_j} w_{itr} e_i \\
&= \sum_{tr} c_r^\psi \, \mathrm{ID}_t^{\psi t} \sqrt{d_j} \rho_j(g^{-1}) w_{rt}
\end{aligned}
$$

where $[\boldsymbol{w}_{rt}]_i = w_{itr}$ if $i \le m_j$ and $[\boldsymbol{w}_{rt}]_i = 0$ otherwise. Unfortunately, this basis is not harmonic since it does not have an explicit action of $G$ through $\rho_j$ on the coefficients vectors. To see this, consider the function above transformed by an element $k \in G$:

$$
\begin{aligned}
[k.f](g) &= \sum_{itr} w_{itr} [k.f_{jitr}](g) = \sum_{itr} w_{itr} f_{jitr}(k^{-1}g) \\
&= \sum_{tr} c_r^\psi \, \mathrm{ID}_t^{\psi j} \sqrt{d_j} \rho_j(g^{-1}) \rho_j(k) w_{rt}
\end{aligned}
$$

The vector $\rho_j(k) \boldsymbol{w}_{rt}$ does not generally belong to the span of $\{e_i | i \le m_j\}$, i.e. the function $k.f$ as written above is not expressed directly in terms of the basis in Corollary 5. This means that the basis we found is not an harmonic basis yet.

Nevertheless, we know that the space of Mackey function has a compact action of $G$ and therefore must be decomposable in a direct sum of irreps of $G$. Therefore, there exists a change of basis from the current one to an harmonic one. In the rest of this section, we will derive a variation of the **Gram–Schmidt process** to compute such change of basis.

First of all, we want to identify the invariant subspaces spanned by the basis. To do so, we re-project each basis element after an action of $G$ on the basis itself, i.e. we project the function $g.f_{jitr}$, with $g \in G$ on each other basis element $f_{jksq}$. First, note that:

$$[g.f_{jitr}](k) = c_r^\psi \, \mathrm{ID}_t^{\psi j} \, \sqrt{d_j} \rho_j(k^{-1}) \rho_j(g) e_i$$
$$= \sum_{i' \le d_j} c_r^\psi \, \mathrm{ID}_t^{\psi j} \, \sqrt{d_j} \rho_j(k^{-1}) e_{i'} [\rho_j(g)]_{i',i}$$

i.e. $g.f_{jitr}$ belongs to the span of the set of functions $\{f_{jitr} \mid i \le d_j\}$ (note that $i \le d_j$ rather than $i \le m_j = d_j/E_j$). Despite the functions $\{f_{jitr} \mid m_j < i \le d_j\}$ do not belong to the basis in Corollary 5, one can easily verify that they are actually Mackey functions and, therefore, must belong to the space spanned by the basis in Corollary 5. Additionally, we will see that $\{f_{jitr} \mid i \le d_j\}$ contains functions which are orthogonal to each other; before showing this, we should define an inner product of Mackey functions:

$$\langle f_1, f_2 \rangle_G = \int_G \langle f_1(g), f_2(g) \rangle_\psi \mathrm{d}g = \int_G f_1(g)^T f_2(g) \mathrm{d}g$$

Note that we assumed real functions and real representations.

We can now compute the projection of any two functions:

$$\langle f_{jitr}, f_{jksq} \rangle = \int_{g \in G} e_i^T \rho_j(g) \sqrt{d_j} [\mathrm{ID}_t^{\psi j}]^T [c_r^\psi]^T c_q^\psi \, \mathrm{ID}_s^{\psi j} \, \sqrt{d_j} \rho_j(g)^T e_k \mathrm{d}g$$
$$= e_i^T \underbrace{\left( \int_{g \in G} \rho_j(h) \sqrt{d_j} [\mathrm{ID}_t^{\psi j}]^T [c_r^\psi]^T c_q^\psi \, \mathrm{ID}_s^{\psi j} \, \sqrt{d_j} \rho_j(g)^T \mathrm{d}g \right)}_{=: \boldsymbol{O}_{tr,sq}} e_k$$

**Lemma D.8.** *the following statements are true:*

    *1.* $\boldsymbol{O}_{tr,sq} \in \mathrm{End}_G(\rho_j)$

    *2.* $\boldsymbol{O}_{tr,sq} = \sum_l \langle \boldsymbol{O}_{tr,sq}, I_l \rangle I_l = \sum_l \mathrm{Tr}([\mathrm{ID}_t^{\psi j}]^T [c_r^\psi]^T c_q^\psi \, \mathrm{ID}_s^{\psi j} \, I_l^T) I_l$

*Proof.* 1) By definition, $\boldsymbol{O}_{tr,sq}$ is the invariant projection of a matrix with respect to a left and right action of $G$ through $\rho$. It follows that $\boldsymbol{O}_{tr,sq}$ commutes with $\rho(g)$ for any $g \in G$; hence, $\boldsymbol{O}_{tr,sq} \in \mathrm{End}_G(\rho_j)$.

2) Let $\{I_l \mid 1 \le l \le E_j\}$ be an orthonormal basis for $\mathrm{End}_G(\rho_j)$ as in Sec. C. Then, we can project $\boldsymbol{O}_{tr,sq}$ on this basis:

$$\langle \boldsymbol{O}_{tr,sq}, I_l \rangle = \frac{1}{d_j} \mathrm{Tr}(\boldsymbol{O}_{tr,sq} I_l^T)$$
$$= \frac{1}{d_j} \mathrm{Tr}(\int_{h \in G} \rho_j(h) \sqrt{d_j} [\mathrm{ID}_t^{\psi j}]^T [c_r^\psi]^T c_q^\psi \, \mathrm{ID}_s^{\psi j} \, \sqrt{d_j} \rho_j(h)^T \mathrm{d}h I_l^T)$$
$$= \frac{1}{d_j} \mathrm{Tr}(\int_{h \in G} \rho_j(h) \sqrt{d_j} [\mathrm{ID}_t^{\psi j}]^T [c_r^\psi]^T c_q^\psi \, \mathrm{ID}_s^{\psi j} \, \sqrt{d_j} I_l^T \rho_j(h)^T \mathrm{d}h)$$
$$= \mathrm{Tr}([\mathrm{ID}_t^{\psi j}]^T [c_r^\psi]^T c_q^\psi \, \mathrm{ID}_s^{\psi j} \, I_l^T)$$

Hence:

$$\boldsymbol{O}_{tr,sq} = \sum_l \langle \boldsymbol{O}_{tr,sq}, I_l \rangle I_l = \sum_l \mathrm{Tr}([\mathrm{ID}_t^{\psi j}]^T [c_r^\psi]^T c_q^\psi \, \mathrm{ID}_s^{\psi j} \, I_l^T) I_l$$

$\square$

Note that the coefficients $\langle \boldsymbol{O}_{tr,sq}, I_l \rangle$ can be easily computed numerically since all the matrices involved are known.

**Corollary 6.** *The following statements are true:*

1. $\langle \boldsymbol{O}_{tr,sq}, I_1 \rangle = \delta_{st} \delta_{rq}$

2. $\boldsymbol{O}_{tr,tr}$ *is the identity matrix*

3. $\langle f_{jitr}, f_{jktr} \rangle = \delta_{ik}$

*Proof.* 1) By considering $l = 1$, i.e. $I_l = \mathrm{id}$ the identity, in Lemma D.8, we find

$$\langle \boldsymbol{O}_{tr,sq}, I_1 \rangle = \mathrm{Tr}([\mathrm{ID}_t^{\psi j}]^T [c_r^\psi]^T c_q^\psi \, \mathrm{ID}_s^{\psi j})$$
$$= \mathrm{Tr}(\mathrm{ID}_s^{\psi j} [\mathrm{ID}_t^{\psi j}]^T [c_r^\psi]^T c_q^\psi)$$

Leveraging the orthogonality of $\mathrm{ID}^{\psi j}$ and the fact that $\mathrm{ID}_t^{\psi j}$ and $\mathrm{ID}_s^{\psi j}$ contain the rows of $\mathrm{ID}^{\psi j}$:

$$= \mathrm{Tr}(\delta_{st} \, \mathrm{id} [c_r^\psi]^T c_q^\psi)$$

There exists a unique $r'$ such that $c_{r'}^\psi = [c_r^\psi]^T c_q^\psi$ and $c_{r'}^\psi$ is not skew-symmetric iff $r' = 0$, i.e. $r = q$:

$$= \delta_{st} \delta_{rq}$$

2) Note that the matrix $\boldsymbol{O}_{tr,tr}$ is symmetric; since $I_l$ for $l \neq 1$ is antisymmetric, $\langle \boldsymbol{O}_{tr,tr}, I_l \rangle = 0$ if $l \neq 1$. Using the result in 1), it follows that $\boldsymbol{O}_{tr,tr} = \mathrm{id}$ is the identity matrix.

3) An immediate consequence of this is that $\langle f_{jitr}, f_{jktr} \rangle = e_i^T \boldsymbol{O}_{tr,tr} e_k = \delta_{ik}$. □

Note that the last proposition in Corollary 6 holds also for the functions in $\{f_{jitr} \mid i > m_j = d_j/E_j\}$ which don't belong to the basis in Corollary 5. Intuitively, for a fixed value of $(t, r)$, the basis in Corollary 5 ignores these orthogonal elements since they are linearly dependent with the basis elements associated with some other value of $(t, r)$. This explains why the form of the basis in Corollary 5 is not convenient for us. Using the projection coefficients $\boldsymbol{O}_{tr,sq}$, we can compute the necessary change of basis.

To do so, we start with the over-complete basis given by:

$$\{f_{jitr} \mid i \leq d_j, r \leq E_\psi, t \leq [\psi j]\}$$

Recall that, for a fixed value of $r$ and $t$, all basis elements $B_{tr} = \{f_{jitr}\}_i$ are orthogonal. We can therefore build a basis by iteratively picking a set $B_{tr}$ and project it to the space orthogonal to the one spanned by the basis elements already chosen.

Before making this more precise, we need to introduce a few more handful results.

**Lemma D.9.** *For any $r \leq E_\psi$, $t \leq [\psi j]$ and $l \leq E_j$:*

1. $c_r^\psi \, \mathrm{ID}_t^{\psi j} \, I_l \in \mathrm{Hom}_H(\mathrm{Res}_H^G \rho_j, \psi)$

2. *there is a set of real numbers $(w_{tr,sq,l})_{s,q}$ s.t.*

$$c_r^\psi \, \mathrm{ID}_t^{\psi j} \, I_l = \sum_{q \leq E_\psi} \sum_{s \leq [\psi j]} c_q^\psi \, \mathrm{ID}_s^{\psi j} \, w_{tr,sq,l}$$

3. $w_{tr,sq,l} = \langle \boldsymbol{O}_{tr,sq}, I_l \rangle$

*Proof.* 1) Note that $c_r^\psi \, \mathrm{ID}_t^{\psi j} \in \mathrm{Hom}_H(\mathrm{Res}_H^G \rho_j, \psi)$ and $I_l \in \mathrm{End}_G(\rho)$. It follows that, for any $h \in H < G$:

$$c_r^\psi \, \mathrm{ID}_t^{\psi j} \, I_l \rho(h) = c_r^\psi \, \mathrm{ID}_t^{\psi j} \, \rho(h) I_l = c_r^\psi \psi(h) \, \mathrm{ID}_t^{\psi j} \, I_l = \psi(h) c_r^\psi \, \mathrm{ID}_t^{\psi j} \, I_l$$

i.e. $c_r^\psi \, \mathrm{ID}_t^{\psi j} \, I_l$ commutes with the action of $H$; hence, it belongs to $\mathrm{Hom}_H(\mathrm{Res}_H^G \rho_j, \psi)$.

2) This result is an immediate consequence of 1) by using the following decomposition

$$\operatorname{Hom}_H(\operatorname{Res}_H^G \rho_j, \psi) \cong \overset{[\psi j]}{\underset{s}{\bigoplus}} \operatorname{End}_H(\psi)$$

and the basis $\{c_q^\psi\}_q$ for $\operatorname{End}_H(\psi)$.

3) Since $\{c_q^\psi \operatorname{ID}_s^{\psi j}\}_{q,s}$ is an orthonormal basis for $\operatorname{Hom}_H(\operatorname{Res}_H^G \rho_j, \psi)$, the coefficients $(w_{sq,tr,l})_{s,q}$ are found just by projecting the matrix on each basis element using a (scaled) standard inner product, i.e.:

$$
\begin{aligned}
w_{tr,sq,l} &= \langle c_q^\psi \operatorname{ID}_s^{\psi j}, c_r^\psi \operatorname{ID}_t^{\psi j} I_l \rangle \\
&= \frac{1}{d_j} \operatorname{Tr}\left( c_q^\psi \operatorname{ID}_s^{\psi j} I_l^T [\operatorname{ID}_t^{\psi j}]^T [c_r^\psi]^T \right) \\
&= \frac{1}{d_j} \operatorname{Tr}\left( [\operatorname{ID}_t^{\psi j}]^T [c_r^\psi]^T c_q^\psi \operatorname{ID}_s^{\psi j} I_l^T \right) \\
&= \langle \boldsymbol{O}_{tr,sq}, I_l \rangle
\end{aligned}
$$

$\square$

**Lemma D.10.** *Let $\{b_i\}_{i=1}^{d_j}$ a set of Mackey functions defined as:*

$$b_i(g) := \sum_{r,t} c_r^\psi \operatorname{ID}_t^{\psi j} \sqrt{d_j} \rho_j(g)^T \boldsymbol{e}_i b_{r,t}$$

*parameterized by the vector $\boldsymbol{b} = (b_{r,t})_{r,t}$. Let $b(g) = (b_1(g)|\dots|b_i(g)|\dots) \in \mathbb{R}^{d_\psi \times d_j}$ the matrix obtained by stacking horizontally the vectors $\{b_i(g)\}_i$, i.e.*

$$b(g) = \sum_{r,t} c_r^\psi \operatorname{ID}_t^{\psi j} \sqrt{d_j} \rho_j(g)^T b_{r,t}$$

*The set of functions $\{b_i\}_{i=1}^{d_j}$ is an harmonic family, i.e. if $f(g) = b(g)\boldsymbol{f}$ for a coefficients vector $\boldsymbol{f} \in \mathbb{R}^{d_j}$, then $[k.f](g) = f(k^{-1}g) = b(g)\rho_j(k)\boldsymbol{f}$. In other words, the function $k.f$ is the function parameterized by the coefficients $\rho_j(k)\boldsymbol{f}$.*

*Proof.*

$$
\begin{aligned}
f(k^{-1}g) &= b(k^{-1}g)\boldsymbol{f} \\
&= \sum_{r,t} b_{r,t} c_r^\psi \operatorname{ID}_t^{\psi j} \sqrt{d_j} \rho_j(k^{-1}g)^T \boldsymbol{f} \\
&= \sum_{r,t} b_{r,t} c_r^\psi \operatorname{ID}_t^{\psi j} \sqrt{d_j} \rho_j(g)^T \rho_j(k) \boldsymbol{f} \\
&= b(g)\rho_j(k)\boldsymbol{f}
\end{aligned}
$$

$\square$

Finally, we define a few operations which we will need to combine and project harmonic functions in our final algorithm.

**Lemma D.11.** *Let $a$ and $b$ be two set of harmonic functions as in Lemma D.10, respectively parameterized by the vectors $\boldsymbol{a}$ and $\boldsymbol{b}$. Let $\alpha \in \mathbb{R}$ and $M \in \operatorname{End}_G(\rho_j)$; then, the following sets of functions are all harmonic families as in Lemma D.10:*

1. *the set $f$ defined as $f(g) = \alpha a(g)$*

2. *the set $f$ defined as $f(g) = b(g) - a(g)$*

3. *the set $f$ defined as $f(g) = a(g)M$*

*Proof.* 1) This is a simple consequence of the linearity of the harmonic property.

2)

$$f(g) = \sum_{r,t} c_r^\psi \, \mathrm{ID}_t^{\psi j} \, \sqrt{d_j} \rho_j(g)^T b_{r,t} - \sum_{r,t} c_r^\psi \, \mathrm{ID}_t^{\psi j} \, \sqrt{d_j} \rho_j(g)^T a_{r,t}$$

$$= \sum_{r,t} c_r^\psi \, \mathrm{ID}_t^{\psi j} \, \sqrt{d_j} \rho_j(g)^T \overbrace{(b_{r,t} - a_{r,t})}^{=:f_{r,t}}$$

3) Let $M = \sum_l m_l I_l$. Then:

$$a(g)M = \sum_{r,t} c_r^\psi \, \mathrm{ID}_t^{\psi j} \, \sqrt{d_j} \rho_j(g)^T a_{r,t} M$$

$$= \sum_{r,t} c_r^\psi \, \mathrm{ID}_t^{\psi j} \, \sqrt{d_j} \rho_j(g)^T a_{r,t} \sum_l m_l I_l$$

$$= \sum_{l,r,t} a_{r,t} m_l \left( c_r^\psi \, \mathrm{ID}_t^{\psi j} \, I_l \right) \sqrt{d_j} \rho_j(g)^T$$

$$= \sum_{l,r,t} a_{r,t} m_l \left( \sum_{q,s} c_q^\psi \, \mathrm{ID}_s^{\psi j} \, w_{tr,sq,l} \right) \sqrt{d_j} \rho_j(g)^T$$

$$= \sum_{q,s} \underbrace{\left( \sum_{l,r,t} w_{tr,sq,l} a_{r,t} m_l \right)}_{\in \mathbb{R}} c_q^\psi \, \mathrm{ID}_s^{\psi j} \, \sqrt{d_j} \rho_j(g)^T$$

where we used Lemma D.9. $\qquad\square$

**Corollary 7.** *Given two harmonic families $a$ and $b$, let $\boldsymbol{O}_{a,b} = \int_G a(g)^T b(g) \mathrm{d}g$ be the matrix containing the inner product between each pair of functions in $a$ and $b$. Then, $\boldsymbol{O}_{a,b} \in \mathrm{End}_G(\rho_j)$ and $\boldsymbol{O}_{a,b} = \sum_l w_{a,b,l} I_l$, where $w_{a,b,l} = \sum_{sq,tr} a_{s,q} b_{t,r} w_{sq,tr,l}$.*

*Moreover, let $f$ be a set of functions defined as $f(g) = b - a\boldsymbol{O}_{a,b}$. Then, $\boldsymbol{O}_{a,f}$ is the* zero *matrix.*

*Proof.* The first result follows from:

$$\boldsymbol{O}_{a,b} = \int_G a(g)^T b(g) \mathrm{d}g$$

$$= \int_G \sum_{s,q} \sqrt{d_j} a_{s,q} \rho_j(g) [\mathrm{ID}_s^{\psi j}]^T [c_q^\psi]^T \sum_{r,t} c_r^\psi \, \mathrm{ID}_t^{\psi j} \, \sqrt{d_j} \rho_j(g)^T b_{r,t} \mathrm{d}g$$

$$= \sum_{s,q} \sum_{r,t} a_{s,q} b_{r,t} \int_G \sqrt{d_j} \rho_j(g) [\mathrm{ID}_s^{\psi j}]^T [c_q^\psi]^T c_r^\psi \, \mathrm{ID}_t^{\psi j} \, \sqrt{d_j} \rho_j(g)^T \mathrm{d}g$$

$$= \sum_{s,q} \sum_{r,t} a_{s,q} b_{r,t} \boldsymbol{O}_{sq,tr}$$

The second result follows from

$$\boldsymbol{O}_{a,f} = \int_G a(g)^T f(g) \mathrm{d}g = \int_G a(g)^T b(g) \mathrm{d}g - \int_G a(g)^T a(g) \mathrm{d}g \boldsymbol{O}_{a,b} = \boldsymbol{O}_{a,b} - \mathrm{id}\, \boldsymbol{O}_{a,b} = 0 \,.$$

$\qquad\square$

Additionally, note that all functions within the same harmonic family $b$ have the same norm, i.e.

$$\boldsymbol{O}_{b,b} = \sum_{t,r} b_{t,r}^2 \, \mathrm{id} = \|b\| \, \mathrm{id}$$

where we used the fact $\boldsymbol{O}_{b,b}$ is a symmetric matrix and, therefore, $w_{sq,tr,l} = \delta_{st}\delta_{qr}\delta_{l,1}$. $\|b\|$ is the (squared) norm of each function in $b$.

We now have all the ingredients to describe the final algorithm. Let $\mathcal{B} = \{\}$ the (initially empty) set of harmonic families chosen so far. In other words, an element $b_i$ of $\mathcal{B}$ is an harmonic family of $d_j$ functions as in Lemma D.10 and is represented by a vector $\boldsymbol{b}_i = (b_{i,t,r})_{t,r}$. Our goal now is to find $M_j = E_\psi[\psi j]/E_j$ vectors $\{\boldsymbol{b}_i\}_i$ such that $\boldsymbol{O}_{b_i,b_k} = \delta_{i,k}$ id.

To do so, we can repeat the following steps until the set $\mathcal{B}$ contains $M_j$ elements:

1. pick a vector $\boldsymbol{b} = (b_{t,r})_{t,r}$ , which defines the set of harmonics $b$

2. define $b' = b - \sum_{b_i \in \mathcal{B}} b_i \boldsymbol{O}_{b,b'}$, represented by the coefficients vector $\boldsymbol{b}' = (b'_{t,r})_{t,r}$, where the coefficients can be computed using Lemma D.11 and Corollary 7.

3. normalize $b'$ as $\tilde{b} = b'/\sqrt{\|b'\|}$

4. add $\tilde{b}$ to $\mathcal{B}$

Note that, at each iteration, we add $d_j$ orthogonal functions to the new basis and that the original basis in Corollary 5 is $M_j \cdot d_j$ dimensional. Therefore, the algorithm terminates in $M_j$ iterations.

# E   NUMERICAL IRREPS DECOMPOSITION OF REAL REPRESENTATIONS

Assume a compact group $G$ and a set of (representatives of the isomorphism classes of) orthogonal real irreps $\widehat{G} = \{\rho_j\}_j$. In this section, we only engage with real valued representations. In particular, recall that, for a real-valued irrep $\rho$, its endomorphism space $\mathrm{End}_G(\rho)$ is not necessarily 1-dimensional, see Section C.

As stated in Corollary 1, any orthogonal real representation $\pi$ of $G$ can be decomposed as a *direct sum* of irreps of $G$.[20] The goal of this section is that of explicitly computing this decomposition for an arbitrary finite-dimensional orthogonal real representation $\pi$ of $G$. In other words, given an input finite-dimensional (real, orthogonal) representation $\pi$, we want to find the *orthogonal* transformation/matrix $M$ and the multiplicities $\{[j\pi] \in \mathbb{N}\}_j$ such that:

$$\pi(g) = M \left( \bigoplus_{j \in \widehat{G}} \bigoplus_{i=1}^{[j\pi]} \rho_j(g) \right) M^T$$

Thereby, being orthogonal means that the scalar product is preserved, which for matrices means that the matrix is orthogonal in the classical sense – i.e., its columns build an orthonormal basis.

*Warning:* note that, with respect to Corollary 1 and the notation used in the main paper, in this section $M$ is transposed on the right-side rather than the left-side of the direct sum of irreps. To obtain the decomposition used in the other sections, one just need to transpose our final result. The reason why we consider this notation is simply that this is the convention we used in our implementation and we wanted to keep the following algorithm consistent with our code.

In this whole section, we assume that $\pi$ and all $\rho_j$ are matrix representations, and correspondingly, that $M$ is given by a matrix.

To simplify the notation, define $P = \bigoplus_j \bigoplus_i \rho_j$, such that $\pi = MPM^T$. Since $M$ is orthogonal, we can write $\pi M = MP$, i.e. $M$ needs to commute with $\pi$ and $P$. It follows that $M \in \mathrm{Hom}_{G,\mathbb{R}}(P, \pi)$.

---

[20]That corollary speaks of unitary representations. But recall that by our convention, "unitary" means "orthogonal" when the field is the real numbers.

Recall that the block-diagonal structure of $P$ induces a block-structure in the columns of $M$, visually:

$$
\pi(g) = \left[\ldots |M_{j1}| \ldots |M_{ji}| \ldots |M_{j[j\pi]}| \ldots\right] \cdot
\begin{bmatrix}
\ddots & & & & & \\
& \rho(g) & & & & \\
& & \ddots & & & \\
& & & \rho_j(g) & & \\
& & & & \ddots & \\
& & & & & \rho_j(g) \\
& & & & & & \ddots
\end{bmatrix}
\cdot
\begin{bmatrix}
\vdots \\
M_{j1}^T \\
\vdots \\
M_{ji}^T \\
\vdots \\
M_{j[j\pi]}^T \\
\vdots
\end{bmatrix}
$$

i.e., we split $M$ in a number of vertical blocks, one for each occurrence $i$ of each irrep $\rho_j$ in $P$. From the equation $\pi M = MP$ we obtain the following: each such block $M_{ji}$ satisfies $\pi M_{ji} = M_{ji}\rho_j$, i.e. $M_{ji} \in \mathrm{Hom}_G(\rho_j, \pi)$. We restrict our attention to this case. Thus, from now on, assume that $\rho$ is a real orthogonal irrep of $G$, and our goal is to find an orthogonal embedding in $\mathrm{Hom}_G(\rho, \pi)$, where $\pi$ is not necessarily irreducible.

Note that $\mathrm{Hom}_G(\rho, \pi) \cong \bigoplus_{i=1}^{[j\pi]} \mathrm{End}_G(\rho)$. We can easily find a basis for $\mathrm{Hom}_G(\rho, \pi)$ by solving a linear system. Let $d_j$ be the dimension of the representation space of $\rho_j$, and similarly, $d_\pi$ be the dimension of the representation space of $\pi$. Let $I_j$ the $d_j \times d_j$ identity matrix and $I_\pi$ the $d_\pi \times d_\pi$ identity matrix; then, the matrix $M_{ji}$ should satisfy:

$$\forall g \in G, \qquad\qquad\qquad\qquad \pi(g)M_{ji} = M_{ji}\rho_j(g)$$
$$\forall g \in G, \qquad\qquad\qquad\qquad \pi(g)M_{ji}I_j = I_\pi M_{ji}\rho_j(g)$$

vectorizing both sides of the equation and using $\mathrm{vec}\,(ABC) = \left(C^T \otimes A\right)\mathrm{vec}\,(B)$:

$$\forall g \in G, \qquad\qquad (I_j \otimes \pi(g))\,\mathrm{vec}\,(M_{ji}) = (\rho_j(g)^T \otimes I_\pi)\,\mathrm{vec}\,(M_{ji})$$
$$\forall g \in G, \qquad (I_j \otimes \pi(g) - \rho_j(g)^T \otimes I_\pi)\,\mathrm{vec}\,(M_{ji}) = 0$$

Note that $I_j \otimes \pi(g) - \rho_j(g)^T \otimes I_\pi = -\rho_j(g)^T \oplus \pi(g)$, where $\oplus$ indicates the *Kronecker sum* here (not to be confused with the *direct sum*). Note that the last line above describes a linear constraint on $M_{ji}$ for each group element $g \in G$. One can solve for the space of suitable $M_{ji}$ by stacking vertically the matrices $(I_j \otimes \pi(g) - \rho_j(g)^T \otimes I_\pi)$ for each $g \in G$ and then solving for its null space (e.g. through SVD). While this approach is feasible if $G$ is a finite group and $|G|$ is sufficiently small, this can not be used for infinite groups (or large finite groups). If $G$ is a finite group, one can easily verify that it is sufficient to consider the matrices $(I_j \otimes \pi(g) - \rho_j(g)^T \otimes I_\pi)$ associated with the set of generators of $G$. Indeed, note that if $M_{ji}$ satisfies the constraint above for two group elements $a$ and $b \in G$, then it satisfies it also for $ab$ and $ba \in G$. Finzi et al. (2021) follows precisely this approach to compute a basis for a general space $\mathrm{Hom}_G(\pi_1, \pi_2)$, when $G$ is finite. When $G$ is an infinite Lie group, Finzi et al. (2021) shows that it is sufficient to consider the Lie algebra generators of $G$ to solve this constraint. In our work, however, we do not want to rely on the Lie algebra of $G$, since it would introduce additional – non-trivial – requirements when implementing a new equivariance group $G$. Instead, we note that it is generally sufficient to consider a few random samples $\mathcal{G} \subset G$ to generate enough constraint matrices $\{I_j \otimes \pi(g) - \rho_j(g)^T \otimes I_\pi | g \in \mathcal{G}\}$ such that the null space of their vertical stack is precisely $\mathrm{Hom}_G(\rho_j, \pi)$.

A similar method was used already in Weiler et al. (2018a) to compute the Clebsh-Gordan coefficients in the decomposition of the tensor product of the irreps of $\mathrm{SO}(3)$. These coefficients appear in their kernel basis precisely as in our vectorized kernel constraint in Sec. 2.2. In Weiler & Cesa (2019) (Appendix G), the authors discuss the cost of this approach when applied on generic input and output representation pairs, and highlight the benefits of, instead, restricting the consideration to only pairs of input and output irreducible representations, as done here.

Therefore, by following the method just described, we can generate an *orthonormal* basis for the space of vectorized matrices satisfying the constraint above. By un-vectorization, any element in the space spanned by this basis generates a matrix $M_{ji}$ such that $\pi M_{ji} = M_{ji}\rho_j$. This is not sufficient yet for us, since we require the whole matrix $M$ to be *orthogonal*.

From now, we assume the previous method generated an *orthonormal* basis $B_j = \{B_{j,i}\}_i^d$ for $\mathrm{Hom}_G(\rho_j, \pi)$, i.e. a collection of $d = [j\pi] \cdot E_j$ matrices of shape $d_\pi \times d_j$ which are orthogonal with respect to a (scaled) standard inner product $\langle A, B \rangle = \frac{1}{d_j} \mathrm{Tr}(AB^T) = \frac{1}{d_j} \mathrm{vec}(A)^T \mathrm{vec}(B)$. Recall that $E_j$ is the dimensionality of $\mathrm{End}_{G,\mathbb{R}}(\rho_j)$.

In order to build a matrix $M$ which is orthogonal, for each $j$, we need to choose $[j\pi]$ orthogonal matrices $\{M_{ji}\}_i^{[j\pi]}$ in $\mathrm{Hom}_G(\rho_j, \pi)$, such that all columns of all matrices $M_{ji}$ are also orthogonal to each other.

**Lemma E.1.** *Any matrix $A_1 \in \mathrm{Hom}_G(\rho_{j_1}, \pi)$ has columns which are orthogonal to the columns of any matrix $A_2 \in \mathrm{Hom}_G(\rho_{j_2}, \pi)$, for $j_1 \neq j_2$.*

*Proof.* This is a direct consequence of the fact that any homomorphism in $A_1 \in \mathrm{Hom}_G(\rho_{j_1}, \pi)$ maps to a subspace $V_1 \subset V_\pi$ which is orthogonal to the image $V_2$ of $A_2 \in \mathrm{Hom}_G(\rho_{j_2}, \pi)$. This can be seen by realizing that the projection from $V_1$ to $V_2$ is an intertwiner, and thus needs to be zero by Schur's Lemma since $V_1$ and $V_2$ are not isomorphic. $\square$

**Lemma E.2.** *Any orthonormal basis for $\mathrm{Hom}_G(\rho_j, \pi)$ contains matrices which are orthogonal, i.e., such that the columns are orthogonal and normalized.*

*Proof.* Let $\{E_k\}_k^{E_\rho}$ be an orthonormal basis for $\mathrm{End}_G(\rho)$. We know that there exists a matrix $M$ such that 1) $M$ is orthogonal and 2) $\pi = M \left( \bigoplus_j \rho_j^{\oplus [j\pi]} \right) M^T$.[21] Therefore, there exists a collection $\{M_{ji}\}_{i=1}^{[j\pi]}$ of orthogonal matrices in $\mathrm{Hom}_G(\rho_j, \pi)$, where $M_{ji}$ are the blocks of the matrix $M$ before. It follows that there exists a basis for $\mathrm{Hom}_G(\rho, \pi)$ defined as $\mathbb{H} = \{I_k M_{ji} \mid 1 \leq k \leq E_j, 1 \leq i \leq [j\pi]\}$. This basis is made of orthogonal matrices, since both $I_k$ and $M_{ji}$ are orthogonal.

Note that we don't know $M$ and $\mathbb{H}$, but we only know they exist. Assume $\mathcal{H} = \{H_j\}_j$ is a basis for $\mathrm{Hom}_G(\rho, \pi)$ made of orthogonal matrices, i.e. for each $i$, $H_i^T H_i = I$. More generally $H_i^T H_l = \delta_{il} I$. Then any orthonormal basis for this space is made of orthogonal matrices. Indeed, assume $B = \{B_l\}_l$ is a basis such that $\langle B_l, B_l \rangle = 1$. An element $B_l$ can be written as $B_l = \sum_i b_{li} H_i$. It follows that for any element $B_l$:

$$B_l^T B_l = \left( \sum_i b_{l,i} H_i \right)^T \cdot \left( \sum_i b_{l,i} H_i \right)$$
$$= \sum_{i_1, i_2} b_{l,i_1} b_{l,i_2} H_{i_1}^T H_{i_2}$$
$$= \sum_i b_{l,i} b_{l,i} I$$
$$= \langle B, B \rangle I = I$$

i.e. any element of an orthonormal basis for $\mathrm{Hom}_{G,\mathbb{R}}(\rho_j, \pi)$ must contains matrices which are themselves orthogonal. $\square$

It follows that we only need to focus on finding a collection $\{M_{ji}\}_{i=1}^{[j\pi]}$ of matrices whose columns are orthogonal; the orthogonality of columns within the same matrix $M_{ji}$ and between different matrices $M_{j_1, i_1}$ and $M_{j_2, i_2}$ with $j_1 \neq j_2$ is already guaranteed.

Recall that $B_j = \{B_{j,i}\}_i^d$ is the orthonormal basis for $\mathrm{Hom}_G(\rho_j, \pi)$ generated by the numerical method (e.g. SVD). To find suitable $\{M_{ji}\}_i$, it is sufficient to set $M_{j0} = B_{j0}$. Let $\mathcal{M}_j$ the subset of all matrices $\{M_{ji}\}_i$ already set (initially containing only $M_{j0}$). Let $\{I_k\}_{k=1}^{E_j}$ be an orthonormal basis for $\mathrm{End}_{G,\mathbb{R}}(\rho_j)$. Let $\mathcal{B}_{i=0} = B_j$. Then we build the set $\mathcal{M}_j$ through the following iterative method:

1. create $\mathcal{B}' = \{M_{ji} I_k \mid M_i \in \mathcal{M}_j, k \in \{1, \ldots, E_j\}\}$
2. project the elements in $\mathcal{B}_i$ on the space space of matrices which is orthogonal to the span of $\mathcal{B}'$ and remove all zero elements (i.e. all elements which already belong to the span of $\mathcal{B}'$)

---

[21]This is a direct consequence of Lemma B.29 in Lang & Weiler (2020).

3. pick the first non-zero element of $\mathcal{B}_i$, set $M_{ji}$ to that and add it in $\mathcal{M}_j$

4. repeat until the set $\mathcal{B}_i$ is empty

Up to a scalar factor (of $\sqrt{d_j}$), the matrices in $\mathcal{M}_j$ are all orthogonal. Additionally, this set contains precisely $[j\pi]$ matrices.

By stacking all matrices together, one builds the matrix $M$.

Note also that this method can be used to find the multiplicities $[j\pi]$, too. Indeed, because each column of $\rho_j$ is related through an endomorphism precisely with $E_j$ other columns (see Theorem C.11), once a basis $B_j$ with $d$ elements is found, it is sufficient to compute $[j\pi] = d/E_j$, since we assume the *type* of $\rho_j$ (and therefore $E_j$) is known.

## F  REAL IRREDUCIBLE REPRESENTATIONS OF THE DIRECT PRODUCT OF TWO COMPACT GROUPS

A useful operation to generate new groups is the *direct product*. Given two groups $A$ and $B$, their direct product $G = A \times B$ is a group defined as:

- the Cartesian product of the element of $A$ and $B$, i.e. $\{(a, b) \mid a \in A, b \in B\}$
- with the group law $\cdot : G \times G \to G$ defined as $(a_1, b_1) \cdot (a_2, b_2) = (a_1 a_2, b_2 b_2)$.

Many groups can be written as a direct product of two smaller groups. In particular, $O(3) = \text{Inv} \times SO(3)$, or more generally $O(n) = \text{Inv} \times SO(n)$ when $n > 1$ is odd, where $\text{Inv} = \{\pm 1\}$ is the group generated by the reflection $x \mapsto -x$. In our work, we are interested in the direct product to easily build many subgroups of $O(n)$ (in particular, of $O(3)$) which have form $G = \text{Inv} \times H$, where $H < SO(n)$. This is the case for the cylindrical symmetries ($H = SO(2)$, $O(2)$ or their discrete subgroups) or the full isometries of the solids ($H = T$, $O$ or $I$); see Sec. G.

In order to use these groups, it is sufficient for us to build their set of irreducible representations. In this section, we describe a method to generate the set of *real* irreducible representations of a direct product of two compact groups from the irreps of the two groups.

Given two compact groups $A$ and $B$ and their respective set of (representatives of) *complex* unitary irreducible representations $\widehat{A} = \{\alpha_i\}_i$ and $\widehat{B} = \{\beta_j\}_j$, it is well known that any complex unitary irrep of $G = A \times B$ is isomorphic to a unique $\gamma_{ij} = \alpha_i \otimes \beta_j$ for a particular choice of $i$ and $j$. Thereby, $(\alpha_i \otimes \beta_j)(a, b) \coloneqq \alpha_i(a) \otimes \beta_j(b)$ is the Kronecker product of the two matrices $\alpha_i(a)$ and $\beta_j(b)$. Thus, a set of representatives of isomorphism classes of unitary irreps $\widehat{G}$ can be constructed as

$$\widehat{G} = \{\gamma_{ij} = \alpha_i \otimes \beta_j \mid \alpha_i \in \widehat{A}, \ \beta_j \in \widehat{B}\}. \tag{26}$$

However, this results does not hold in the real field. In the rest of this section, we will derive a variation of this theorem using real irreducible representations, which is more useful to describe the representations of the groups we used.

In Appendix C.3, we have already shown that any *real* irrep $\psi$ of $G$ can be classified in one of the following three categories:

- **real-type**: $\psi \cong \sigma \cong \overline{\sigma}$
- **complex-type**: $\psi = \sigma_{\mathbb{R}} \cong \sigma \oplus \overline{\sigma}$, with $\sigma \not\cong \overline{\sigma}$
- **quaternionic-type**: $\psi = \sigma_{\mathbb{R}} \cong \sigma \oplus \overline{\sigma} \cong \sigma \oplus \sigma$, as $\sigma \cong \overline{\sigma}$

where $\sigma$ is a complex irrep of $G$. Moreover, for each complex irrep $\sigma$, there exists only one real irrep $\psi$ which contains it.

We can use this classification starting from the complex irreps of $G$ which we have already defined in Eq. 26. First, we clarify some notation. We use Greek letters like $\alpha, \beta, \gamma$ to indicate complex irreps and Roman letters to indicate the corresponding real irreps $a, b, c$. For each complex irrep $\gamma_{ij}$, there can be only one real irrep $c_{ij}$ containing it. It follows that $\alpha_i \otimes \beta_j$ only appears in $c_{ij}$. Moreover, if $\alpha_i \not\cong \overline{\alpha_i}$, i.e. $a_i$ is of complex-type, we index the complex irrep $\overline{\alpha_i}$ with $\bar{i}$, i.e. $\alpha_{\bar{i}} \cong \overline{\alpha_i}$.

We can now introduce a useful result for real irreps:

**Definition F.1** (Frobenious-Schur Indicator). Let $G$ be a compact group and $\rho$ a *complex* irreducible representation of $G$. Let $\chi$ be the *character* of $\rho$, i.e. $\chi : G \to \mathbb{C}, g \mapsto \text{Tr}(\rho(g))$. The *Frobenious-Schur indicator* $\iota_\rho$ of $\rho$ is defined as:

$$\iota_\rho := \int_G \chi(g^2)\mathrm{d}g$$

This indicator can only take the following values:

$$\iota_\rho = \begin{cases} -1 & \text{if } \rho \text{ is contained in } \textit{real type} \text{ irrep} \\ 0 & \text{if } \rho \text{ is contained in } \textit{complex type} \text{ irrep} \\ 1 & \text{if } \rho \text{ is contained in } \textit{quaternionic type} \text{ irrep} \end{cases}$$

This allows us to analytically determine the type of the real irrep associated with a complex one.

We can combine this result with the definition of complex irreps of $G = A \times B$ in Eq. 26 to find the type of the real irreps of $G$. Let $\gamma_{ij} = \alpha_i \otimes \beta_j$ a complex irrep of $G$; then:

$$\iota_{\gamma_{ij}} = \int_G \text{Tr}((\alpha_i \otimes \beta_j)(g^2))\mathrm{d}g$$
$$= \int_A \int_B \text{Tr}(\alpha_i(a^2) \otimes \beta_j(b^2))\mathrm{d}a \; \mathrm{d}b$$

Using the property of the trace and the Kronecker product:

$$= \int_A \int_B \text{Tr}(\alpha_i(a^2)) \cdot \text{Tr}(\beta_j(b^2))\mathrm{d}a \; \mathrm{d}b$$
$$= \int_A \text{Tr}(\alpha_i(a^2))\mathrm{d}a \int_B \text{Tr}(\beta_j(b^2))\mathrm{d}b$$
$$= \iota_{\alpha_i} \iota_{\beta_j}$$

Hence, the types of the irreps $\alpha_i$ and $\beta_j$ of $A$ and $B$ fully determine the type of the irrep $\gamma_{ij}$ of $G$. This can be visualized in the following table:

| $\alpha_i \;\; \beta_j$ | real $\iota_{\beta_j} = 1$ | complex $\iota_{\beta_j} = 0$ | quaternionic $\iota_{\beta_j} = -1$ |
|---|---|---|---|
| real $\iota_{\alpha_i} = 1$ | $\iota_{\gamma_{ij}} = 1$ | $\iota_{\gamma_{ij}} = 0$ | $\iota_{\gamma_{ij}} = -1$ |
| complex $\iota_{\alpha_i} = 0$ | $\iota_{\gamma_{ij}} = 0$ | $\iota_{\gamma_{ij}} = 0$ | $\iota_{\gamma_{ij}} = 0$ |
| quaternionic $\iota_{\alpha_i} = -1$ | $\iota_{\gamma_{ij}} = -1$ | $\iota_{\gamma_{ij}} = 0$ | $\iota_{\gamma_{ij}} = 1$ |

In the rest of this section, we will use the following two useful facts:

$$\text{Re}\,(\gamma_{ij}) = \text{Re}\,(\alpha_i \otimes \beta_j) = \text{Re}\,(\alpha_i) \otimes \text{Re}\,(\beta_j) - \text{Im}\,(\alpha_i) \otimes \text{Im}\,(\beta_j)$$
$$\text{Im}\,(\gamma_{ij}) = \text{Im}\,(\alpha_i \otimes \beta_j) = \text{Re}\,(\alpha_i) \otimes \text{Im}\,(\beta_j) + \text{Im}\,(\alpha_i) \otimes \text{Re}\,(\beta_j)$$

We will also assume the irreps to be expressed on the bases described in Section C.3. In particular, if $\alpha_i$ (or $\beta_j$) are *real* type irreps, we assume they are expressed with respect to a basis such that they have real entries, i.e. $\alpha_i = a_i$ (and $\beta_j = b_j$). In such case, $\text{Re}\,(\alpha_i) = a_i$ and $\text{Im}\,(\alpha_i) = 0$. If $\alpha_i$ (or $\beta_j$) is a *complex* or *quaternionic* type irrep, we assume $a_i$ (and $b_j$) is expressed as the realification:

$$a_i = \begin{bmatrix} \text{Re}\,(\alpha_i) & -\text{Im}\,(\alpha_i) \\ \text{Im}\,(\alpha_i) & \text{Re}\,(\alpha_i) \end{bmatrix}$$

This enables us to extract $\text{Re}\,(\alpha_i)$ and $\text{Im}\,(\alpha_i)$ directly from the matrix coefficients of $a_i$ (or $b_j$).

Therefore, to build the real irreps of $G = A \times B$ we distinguish three cases, depending on the type of the irrep.

**Real type irrep**  Assume $\gamma_{ij}$ has *real type*. This implies that either both $\alpha_i$ and $\beta_j$ have real type or both have quaternionic type. In the first case, $g_{ij}$ is simply computed as $g_{ij} = a_i \otimes b_j$. In the second case, assume $\alpha_i$ and $\beta_j$ have the structure in Eq. 17. Then, one can verify that the change of basis

$$Q = \frac{1}{\sqrt{2}} \begin{bmatrix} 0 & iI & iI & 0 \\ -iI & 0 & 0 & iI \\ 0 & -I & I & 0 \\ I & 0 & 0 & I \end{bmatrix}$$

turns $\gamma_{ij} = \alpha_i \otimes \beta_j$ into a matrix with only real entries, i.e. $g_{ij} = Q\gamma_{ij}Q^\dagger$, where $I$ is the identity matrix of size $\dim_{\gamma_{ij}}/4$.

**Complex type irrep**  Whenever $\gamma_{ij}$ has complex type, we can easily implement $g_{ij}$ as the realification:

$$g_{ij} := \begin{bmatrix} \mathrm{Re}\,(\gamma_{ij}) & -\mathrm{Im}\,(\gamma_{ij}) \\ \mathrm{Im}\,(\gamma_{ij}) & \mathrm{Re}\,(\gamma_{ij}) \end{bmatrix}$$

By using the identities above, we can easily compute this matrix from entries of the matrices of $a_i$ and $b_j$. Note that, if both $\alpha_i$ and $\beta_j$ have complex type, the irrep $\gamma_{ij} = \alpha_i \otimes \beta_j$ and $\gamma_{i\bar{j}} = \alpha_i \otimes \overline{\beta_j}$ generate two different real irreps $g_{ij}$ and $g_{i\bar{j}}$. Since $b_j$ and $b_{\bar{j}}$ are isomorphic real irreps, only one of the two belongs to the set of representatives in $\widehat{G}$ chosen. However, $\mathrm{Re}\,(\beta_j) = \mathrm{Re}\,(\overline{\beta_j})$ and $\mathrm{Im}\,(\beta_j) = -\mathrm{Im}\,(\overline{\beta_j})$, so $g_{i\bar{j}}$ can still be computed easily from the coefficients of $a_i$ and $b_j$.

**Quaternionic type irrep**  Like for the complex types, whenever $\gamma_{ij}$ has quaternionic type, we can easily implement $g_{ij}$ as the realification:

$$g_{ij} := \begin{bmatrix} \mathrm{Re}\,(\gamma_{ij}) & -\mathrm{Im}\,(\gamma_{ij}) \\ \mathrm{Im}\,(\gamma_{ij}) & \mathrm{Re}\,(\gamma_{ij}) \end{bmatrix}$$

Additionally, note that this is the case only if either $\alpha_i$ or $\beta_j$ is of real type, while the other is of quaternionic type. Hence, the formula can be simplified by noting that either $\mathrm{Im}\,(\alpha_i)$ or $\mathrm{Im}\,(\beta_j)$ is zero.

## G  3D SYMMETRIES, AZIMUTHAL SYMMETRIES AND THEIR RELEVANCE

$E(3)$ is the group of isometries in $\mathbb{R}^3$ and includes translations $(\mathbb{R}^3, +)$, rotations $SO(3)$ and inversions[22] through the origin Inv. The group $O(3) = SO(3) \times \mathrm{Inv}$ is the compact group of origin-preserving isometries. While continuous rotational symmetry $SO(3)$ occurs commonly in molecular data, smaller symmetry groups are sometimes more relevant to describe scenes or objects in natural environments. For instance, when the vertical orientation of a the scene is obvious, the symmetry group of the data is reduced to axial symmetries. There is only a limited number of topologically closed infinite subgroups of $O(3)$ (up to conjugation). The building blocks are rotations along an axis ($SO(2)$), inversions (Inv), mirroring with respect to a plane (M) and an out-of-plane rotation by $\pi$ (F). Together, they generate 5 continuous symmetry groups:

- *axial symmetry* ($SO(2) < SO(3)$),
- *dihedral symmetry* ($O(2) = SO(2) \rtimes F < SO(3)$),
- *conical symmetry* ($O(2) \cong SO(2) \rtimes M < O(3)$),
- *cylindrical symmetry* ($\mathrm{Inv} \times SO(2) < O(3)$) and
- *full cylindrical symmetry* ($\mathrm{Inv} \times (SO(2) \rtimes F) < O(3)$).

In the planar case, Weiler & Cesa (2019) observed benefits from using discrete groups rather than continuous ones when the data is discretized on a grid and, therefore, the continuous symmetry of the data is lost. The only discrete subgroups of $SO(3)$ with more than 2 rotation axes are the symmetry groups of the Platonic solids: the *tetrahedral* (T), the *octahedral* (O) and the *icosahedral* (I) groups which have, respectively, 12, 24, and 60 elements. T and O are perfect symmetries of a voxel grid and were previously used in Worrall & Brostow (2018); Winkels & Cohen (2018). Also, note that $T < I$ but $O \not\leq I$; hence, an I-equivariant CNN is not equivariant to all symmetries of the voxel grid.

---

[22]$\mathrm{Inv} \cong C_2$ is a group containing the identity element and the inversion $x \mapsto -x$ in $\mathbb{R}^3$.

# H  FURTHER DETAILS ON NETWORK DESIGNS AND EXPERIMENTS

## H.1  ESTIMATING THE EQUIVARIANCE ERROR OF STEERABLE BASES

In this section we provide more details on Fig. 5. In this experiment, we considered equivariance to the Icosahedral group $G = \mathrm{I}$. We employ $5 \times 5 \times 5$ filters, parameterized by two spherical shells (one at radius 1 and one at radius 2) and the origin. In the two baselines, we use the orbits containing the vertices of either the icosahedron or the dodecahedron in the two spherical shells and the orbit containing only the origin. The kernel is parameterized independently along each orbit by directly using the Wigner-Eckart theorem from Lang & Weiler (2020) and, then, the kernels are embedded in $\mathbb{R}^3$ by means of a small Gaussian smoothing kernel. In our basis, we parameterize each shell with spherical harmonics band-limited to frequency 2 (the origin contains only frequency 0); each shell is diffused along the radial direction with a small Gaussian kernel.

Each histogram in Fig. 5 shows the relative equivariance error of each element of one of the bases. An histogram contains a point for each filter $k$ in the basis associated with each pair of input and output irreps $(\rho_{\mathrm{in}}, \rho_{\mathrm{out}})$ of $G = \mathrm{I}$. The error of a basis filter $k$ is computed as the average over all 60 rotations $g \in \mathrm{I}$ of the following error:

---

$$\mathtt{gx} \leftarrow \mathrm{downsample}(g.X)$$
$$\mathtt{x} \leftarrow \mathrm{downsample}(X)$$
$$\mathtt{fgx} \leftarrow \mathrm{conv3d}(\mathtt{gx}, k)$$
$$\mathtt{fx} \leftarrow \mathrm{conv3d}(\mathtt{x}, k)$$
$$\mathtt{gfx} \leftarrow \rho_{\mathrm{out}}(g)\mathtt{fx}$$
**return** $\mathrm{compare}(\mathtt{fgx}, \mathtt{gfx})$

---

where $\mathtt{x}$ (and $\mathtt{gx}$) have the same resolution of the filter $k$, such that the outputs $\mathtt{fx}$ and $\mathtt{fgx}$ do not have spatial resolutions. In this way, $g$ acts on $\mathtt{fx}$ only by transforming its channels but no spatial interpolation is necessary. To reduce the interpolation artifacts when rotating $X \mapsto g.X$, we perform the rotation at a higher resolution and then downsample the inputs by a factor of 3.

## H.2  REGULAR AND QUOTIENT NON-LINEARITIES

*Quotient* and *regular*-type features encode square-integrable scalar functions over a homogeneous space $Q = G/H$ of $G$ (with $H = \{e\}$ and $Q = G$ for the regular case). Whenever $Q = G/H$ is infinite, one can rely on a *bandlimited* representation of these functions in the Fourier domain, using the harmonic bases we derived in Section D. More precisely, *bandlimited* functions are parameterized by using a finite subset of the harmonic basis in Corollary 5, considering only those $j$ in a subset $\widetilde{G} \subset \widehat{G}$. Then, the action of $G$ on a function turns into the representation $\widetilde{\rho_Q} = \bigoplus_{j \in \widetilde{G}} \bigoplus^{M_j} \rho_j$ acting on the coefficients vector which parameterizes it, where $M_j$ is the multiplicity of the irrep $\rho_j$. Given one such vector, the function it parametrizes can be sampled at any point $q \in Q$ by evaluating the harmonic basis on $q$.

In a neural network, a feature vector $f(x) \in \mathbb{R}^{d_\rho}$ of *quotient* (or regular) type $\rho$ is interpreted as the coefficients vector parameterizing a band-limited function in $L^2(Q) = L^2(G/H)$ on a basis, which transforms according to $\rho$.

A point-wise nonlinearity $\sigma : \mathbb{R} \to \mathbb{R}$ (e.g. ELU) is applied on a quotient feature $f(x) : Q \to \mathbb{R}$ by composition, obtaining a new quotient feature $f'(x)$, defined as $[f'(x)](q) = \sigma([f(x)](q))$. When $Q$ is infinite, if $f(x)$ is band-limited and $\sigma$ is sufficiently smooth, we can recover an approximation of $f'(x)$ from a finite number of samples.

**Approximate point-wise non-linearity**  Let $\mathcal{Q} \subset Q$ be a finite number of samples from the space $Q = G/H$. We define the non-linearity as

$$f(x) \mapsto f'(x) = [\mathrm{FT} \circ \sigma \circ \mathrm{IFT}](f(x))$$

where FT is a *Fourier Transform*, which recovers the coefficients of a band-limited function from a number of samples, $\sigma$ is the pointwise non-linearity and IFT is an *Inverse Fourier Transform*, which just samples the function parametrized by the coefficients $f(x)$ on the elements in $\mathcal{Q}$. Note that IFT

can be implemented as a simple multiplication with a matrix of shape $|\mathcal{Q}| \times d_\rho$ and, similarly, FT can be implemented as a multiplication with a matrix of shape $d_\rho \times |\mathcal{Q}|$, hence

$$f'(x) = \text{FT}\,\sigma(\text{IFT}\,f(x))\,,$$

where FT and IFT are interpreted as matrices. In practice, the matrix IFT is implemented as a matrix whose rows contain the Fourier transform of Dirac delta functions centered at each sample in $\mathcal{Q}$, while FT is implemented as the *pseudo-inverse matrix* $\text{FT} = \text{IFT}^\dagger$. This operation is only approximatively equivariant and the degree of equivariance strongly depends on the number of samples $|\mathcal{Q}|$ as well as their distribution over the group (and the smoothness of $\sigma$). In particular, it is important that the samples in $\mathcal{Q}$ uniformly cover the space $Q = G/H$ and are, therefore, as far away from each other as possible.

To generate a sampling set $\mathcal{Q}$, one can use the elements of a discrete subgroup of $G$ (e.g. Icosahedral group $I < \text{SO}(3)$) or optimize the location of $|\mathcal{Q}|$ points on $Q$ by minimizing a potential energy, as done by Bekkers (2020).

**Computational benefits**  A typical feature $f$ in our models includes multiples fields of the same type, i.e. $f = \bigoplus_i f_i$ and $\rho = \bigoplus_i \rho_i$, with $\rho_i \cong \rho_j \,\forall i, j$. Assume a sequence $\texttt{ConvND} \to \sigma \to \texttt{ConvND}$ in the model, where all layers operate on feature fields of type $\rho$. Let $C = \sum_i d_{\rho_i}$ be the size of the feature vectors $f(x) \in \mathbb{R}^C$. In our neural networks design, we ensure that the number of channels in each layer is approximately constant across different models. In particular, in models using quotient (or regular) feature types, we count the number of samples $\mathcal{Q}$ as channels and, therefore, keep the quantity $\sum_i |\mathcal{Q}| \approx N$ approximately constant. Let $\alpha = |\mathcal{Q}|/d_{\rho_i}$; this implies that the first $\texttt{ConvND}$ only has $\sum_i d_{\rho_i} \approx N/\alpha$ output channels (and the second $\texttt{ConvND}$ has this many input channels). Let $k$ be the cost of a single convolution with 1 input and output channels and $m$ the number of pixels in the input grid. The cost of the non-linearity applied to a single feature field is $O(d_{\rho_i}|\mathcal{Q}|)$ and there are $N/|\mathcal{Q}|$ such input fields. Then, the cost of this layer is only $O(kmCN/\alpha + m(N/|\mathcal{Q}|)|\mathcal{Q}|d_{\rho_i}) = O(kmCN/\alpha + mNd_{\rho_i})$ which is equal to $O(kmCN/\alpha)$, since $mC \gg d_{\rho_i}$. For comparison, the same block in a conventional network has cost $O(kmCN + mN) = O(kmCN)$, hence this layer is approximatively $\alpha$ times cheaper.

In Section H.7, we compare the approximate inference time of different architecture to verify the computational benefits described above.

## H.3  Architecture Details

All models used in our experiments consist of a sequence of a basic inverted-bottleneck residual block, or small variations of it (see each experiment's section). The basic block for a conventional network consists of a sequence $\texttt{Conv3D}$ - $\texttt{BatchNorm}$ - $\texttt{ELU}$ -$\texttt{Conv3D}$, whose input and output are connected with a skip connection. The block has $C_\text{in}$ input channels and the first convolution maps to $N > C_\text{in}$ channels. To downsample, we used stride 2 in the second $\texttt{Conv3D}$ and average pooling in the skip connection. We use $\texttt{kernel\_size} = 3$ for both $\texttt{Conv3D}$. If the block has a different number of channels $C_\text{out}$ in output, the skip connection also contains a convolution with $\texttt{kernel\_size} = 1$. We refer to the linear path in the architecture, i.e. the sequence of skip connections, as the *backbone* of the model. If $C_j$ is the number of input channels of the $j$-th residual block, we refer to $[C_1, C_2, \ldots]$ as the number of channels in the backbone features of the model. If $N_j$ is the number of channels used by the non-linearity of the $j$-th residual block, we refer to $[N_1, N_2, \ldots]$ as the number of channels in the residual blocks of the model.

In an equivariant model, we associate a stack of field types $\{\rho_i\}_i$ on the input $C_\text{in}$ channels, such that $\sum_i d_{\rho_i} \approx C_\text{in}$. Similarly, we associate a stack of field types to the $N$ channels produced by the first $\texttt{Conv3D}$. The particular choice here depends on the kind of equivariant non-linearity used here instead of $\texttt{ELU}$. In general, we try to ensure that the channels used in the non-linearity are approximatively equal to $N$, for a fair comparison. We now list a number of non-linearities considered.

**Gated Non-Linearities**  The input consists of a number of feature fields $f(x) = \bigoplus_i f_i(x)$ of type $\bigoplus_i \rho_i$. Recall that $f_i(x) \in \mathbb{R}^{d_{\rho_i}}$. For each feature field $i$, the input should also contain a *gate* $g_i(x) \in \mathbb{R}$ of trivial type. Then, each input feature field $f_i(x)$ is mapped to $f'(x) = \sigma(g_i(x))f_i(x)$,

where $\sigma$ is a non-linearity like `Sigmoid`. Note that a gated non-linearity inherently loses some of its input channels, i.e. the input gates $\{g_i\}_i$. In our models, we try to ensure that $\sum_i (d_{\rho_i} + 1) \approx N$ (the $+1$ corresponds to the size of the gate $g_i(x)$ for each input feature field $f_i$). Note that a field type $\rho_i$ can be any representation. In particular, we experiment with different choices of $\rho_i$; we consider $\{\rho_i\}_i$ containing:

- each irrep of $\mathrm{SO}(3)$ of frequencies 1 to 2, in equal proportions
- each irrep of $\mathrm{SO}(3)$ of frequencies 1 to 3, in equal proportions
- each irrep of $\mathrm{SO}(3)$ of frequencies 1 to 2, with multiplicities proportional to their multiplicities in the regular representation of $\mathrm{SO}(3)$, i.e. for each trivial $\rho_0$, we include 3 copies of the frequency 1 $\rho_1$ and 5 copies of $\rho_2$
- copies of the quotient representation $\rho_{\mathcal{S}^2} = \rho_0 \oplus \rho_1 \oplus \rho_2$, band-limited up to frequency 2
- copies of the quotient representation $\rho_{\mathcal{S}^2} = \rho_0 \oplus \rho_1 \oplus \rho_2 \oplus \rho_3$, band-limited up to frequency 3
- copies of the regular representation $\rho_{\mathrm{reg}} = \rho_0 \oplus \rho_1^{\oplus 3} \oplus \rho_2^{\oplus 5}$, band-limited up to frequency 2

To match the $N$ channels, we also add trivial feature fields ($\rho_i = \rho_0$) and operate on them with a simple `ELU` non-linearity. Note that in all cases, the first `Conv3D` has $C_{\mathrm{in}}$ input channels and $N$ output channels, maintaining approximatively the same computational cost of the conventional CNN.

**Tensor-Product Non-Linearity**  Let $f(x) = \bigoplus_i f_i(x)$ be the input features field, with type $\rho = \bigoplus_i \rho_i$. In our experiments, we always chose all $\{\rho_i\}_i$ equal to a unique type, i.e. $\rho_i = \rho_j$ for all $i, j$. The tensor-product non-linearity maps $f_i(x) \in \mathbb{R}^{d_{\rho_i}}$ to $f_i'(x) = f_i(x) \otimes f_i(x) \in \mathbb{R}^{d_{\rho_i}^2}$. Note the quadratic growth in the number of channels. In the experiments, we keep the output size approximatively equal to $N$, i.e. $\sum_i d_{\rho_i}^2 \approx N$. It is common in the literature to learn a linear projection of $f_i'(x)$ to a lower-dimensional feature field $f_i''(x)$ of type $\rho_i'$, with $d_{\rho_i'} \ll d_{\rho_i}^2$. The second `Conv3D` originally maps $\sum_i d_{\rho_i}^2 \approx N$ channels to the output $C_{\mathrm{out}}$ channels, while this projection would reduce the input channels to $\sum_i d_{\rho_i'} \ll N$, thereby reducing the computational cost of the layer proportionally. This, however, results also in a potential loss in expressiveness, since a `Conv3D` layer taking $N$ channels in input can learn a larger space of operations. Since we do not expect this linear projection to improve the performance of this architecture, we do not include it in our experiments. Finally, in our experiments we consider the following choices of $\rho_i$:

- the quotient representation $\rho_{\mathcal{S}^2} = \rho_0 \oplus \rho_1 \oplus \rho_2$, band-limited up to frequency 2
- the quotient representation $\rho_{\mathcal{S}^2} = \rho_0 \oplus \rho_1 \oplus \rho_2 \oplus \rho_3$, band-limited up to frequency 3
- the regular representation $\rho_{\mathrm{reg}} = \rho_0 \oplus \rho_1^{\oplus 3} \oplus \rho_2^{\oplus 5}$, band-limited up to frequency 2

Note that a larger feature type $\rho_i$ (i.e. larger $d_{\rho_i}$) results in a smaller number of independent copies of it, since we constrain $\sum_i d_{\rho_i}^2 \approx N$. On the other hand, a larger $\rho_i$ makes the tensor-product $f_i(x) \otimes f_i(x)$ more expressive, since it combines more different input channels. Therefore, the choice of the size of $\rho_i$ should be a trade-off between expressiveness and computational and memory cost.

**(Pointwise) Regular non-Linearities**  Again, we consider all input feature fields of the same type, i.e. $\rho_i = \rho_j$ for all $i, j$. Here, $f_i(x) \in \mathbb{R}^{d_{\rho_i}}$ is interpreted as the coefficients vectors parameterizing a band-limited function in $L^2(G)$, as described in Section H.2. We consider band-limiting up to frequency 2 or 3, i.e. the three following choices of $\rho_i$:

- $\rho_i = \rho_0 \oplus \rho_1^{\oplus 3}$
- $\rho_i = \rho_0 \oplus \rho_1^{\oplus 3} \oplus \rho_2^{\oplus 5}$
- $\rho_i = \rho_0 \oplus \rho_1^{\oplus 3} \oplus \rho_2^{\oplus 5} \oplus \rho_3^{\oplus 7}$

We experiment with different sampling sets $\mathcal{G} \subset G$:

- the 24 elements of the Octahedral group $\mathrm{O} < \mathrm{SO}(3)$
- the 60 elements of the Icosahedral group $\mathrm{I} < \mathrm{SO}(3)$

- $P$ points over $G$ with maximal relative distance, obtained by optimizing a potential energy
- $P = 24 \cdot |S|$ points over $G$ with maximal relative distance, obtained by optimizing over a set $S \subset G$ a potential energy over $\mathcal{G} = \bigcup_{o \in \mathrm{O}} \{o.g | g \in S\}$, such that $\mathcal{G}$ has Octahedral symmetry: for any $o \in \mathrm{O}$, $o.\mathcal{G} = \mathcal{G}$.

In particular, the last sampling set guarantees that the resulting neural network is at least perfectly equivariant to O, i.e. the symmetries of the voxel grid. In general, we observe that at least $|\mathcal{G}| \approx 2d_{\rho_i}$ samples are necessary for achieving an equivariance error below $10\%$. For $G = \mathrm{O}(3)$, we use a similar design and just augment the sampling set with inversions, i.e. we use $\mathcal{G}' = \mathrm{Inv} \times \mathcal{G}$, where $\mathcal{G}$ is a sampling set used for $\mathrm{SO}(3)$.

We argued about the computational benefits of this design in Section H.2. For discrete groups such as O and I, we also use regular non-linearities. In this case, however, the first `Conv3D` in the residual block directly use a *regular feature types* as output types and is immediately followed by the pointwise non-linearity, with no need to perform sampling. Hence, the computational and memory cost of these models is equivalent to that of a conventional network, i.e. they have no additional computational gain.

**(Pointwise) Quotient non-Linearities** Similarly, we define quotient non-linearities, with the only difference that $f_i(x)$ parameterizes a band-limited function over a homogeneous space $Q = G/H$, for some $H < G$; see Section H.2. For $G = \mathrm{SO}(3)$, we consider the following quotient representations $\rho_i$:

- $\rho_{\mathcal{S}^2} = \rho_0 \oplus \rho_1 \oplus \rho_2$
- $\rho_{\mathcal{S}^2} = \rho_0 \oplus \rho_1 \oplus \rho_2 \oplus \rho_3$

and the following sampling sets

- the (center of the) 30 edges of the icosahedron; note that this set has Icosahedral symmetry
- $P$ points over $Q$ with maximal relative distance, obtained by optimizing a potential energy
- $P = 24 \cdot |S|$ points over $Q$ with maximal relative distance, obtained by optimizing over a set $S \subset Q$ a potential energy on $\mathcal{Q} = \bigcup_{o \in \mathrm{O}} \{o.q | q \in S\}$, such that $\mathcal{Q}$ has Octahedral symmetry: for any $o \in \mathrm{O}$, $o.\mathcal{Q} = \mathcal{Q}$.

In particular, the last sampling set guarantees that the resulting neural network is at least perfectly equivariant to O, i.e. the symmetries of the voxel grid. Additionally, note that for $Q = \mathcal{S}^2$, the second sampling set corresponds to the well known Thomson problem. When $G = \mathrm{O}(3)$, we consider $Q = \mathrm{O}(3)/\mathrm{O}(2) \cong \mathcal{S}^2$ and $Q = \mathrm{O}(3)/\mathrm{SO}(2) \cong \mathrm{Inv} \times \mathcal{S}^2$. In the first case, we can reuse the same sampling sets defined above for $Q = \mathrm{SO}(3)/\mathrm{SO}(2) \cong \mathcal{S}^2$. In the second one, we augment the sampling set with inversions. The considerations about the computational cost discussed in Section H.2 apply also here.

**Invariant Maps** Because we consider invariant tasks, an equivariant architecture should map to invariant features before the final classification layer. To do so, it is common to use some invariant map at the end of the convolutional feature extractor and then feed the invariant features in a few final fully-connected layers. Here, we consider a few different invariant maps. **Norm-pooling**: given an input field $f_i(x) \in \mathbb{R}^{d_{\rho_i}}$, we compute its norm, i.e. $f_i(x) \mapsto f'_i(x) = ||f_i(x)||_2 \in \mathbb{R}$; note that this is indeed an invariant quantity. We use this kind of map for models which should be perfectly $G$ equivariant, if $G$ is a continuous group; for instance, we use it for models which use gated non-linearities or tensor-product non-linearities. We experiment with different choices of $\rho_i$:

- the quotient representation $\rho_{\mathcal{S}^2} = \rho_0 \oplus \rho_1 \oplus \rho_2$, band-limited up to frequency 2
- the regular representation $\rho_{\mathrm{reg}} = \rho_0 \oplus \rho_1^{\oplus 3}$, band-limited up to frequency 1
- the regular representation $\rho_{\mathrm{reg}} = \rho_0 \oplus \rho_1^{\oplus 3} \oplus \rho_2^{\oplus 5}$, band-limited up to frequency 2
- $\{\rho_i\}_i$ contains copies of the irrep $\rho_0$, $\rho_1$ and $\rho_2$ in equal proportion

**Group Pooling**: if $G$ is a finite group, we can choose $\rho_i = \rho_{\mathrm{reg}}$ and, therefore, $f_i(x) \in \mathbb{R}^{|G|}$. In this case, we can compute the max $f_i(x) \mapsto \max_j [f_i(x)]_j$. Something similar can be done if $G$

is continuous and $\rho_i$ is a band-limited regular representation by using a sampling set $\mathcal{G}$. However, this operation is often more unstable. To devise a more stable alternative for continuous groups, we consider **Average Pooling**. This operation is identical to the *regular non-linearity* explained above (i.e., $f_i'(x) = [\text{FT} \circ \sigma \circ \text{IFT}](f_i(x))$), with the difference that the final Fourier Transform FT only recovers the invariant frequency 0 component. This is equivalent to averaging the output of $[\sigma \circ \text{IFT}](f_i(x))$ over the samples $\mathcal{G}$. In our architectures, we keep the number of invariant features fed into the final fully-connected layers constant. Note that, to generate $N$ invariant outputs, the invariant maps require $Nd_{\rho_i}$ inputs. This can be quite expensive if $\rho_i$ is a regular representation (e.g., a regular representation of $G = \text{SO}(3)$ band-limited to 2 has size 35). To alleviate this, we introduce a **Quotient Average Pooling**. This operation is similar to *Average Pooling* but relies on quotient representations (and samples $\mathcal{X}$ from a quotient space $X$), which have smaller size.

## H.4 MNIST EXPERIMENTS

As a proof of concept, we train a conventional and a $\text{C}_8$ equivariant model on rotated MNIST. A model includes a sequence of 6 residual blocks similar to the ones described in Section H.3. With respect to that design, here we use `Conv2D` and use stride 2 in the second convolution. We downsample (with stride 2 convolution), every two blocks. In the first layer of the first block, we use `kernel_size = 7`. The number of channels in the features of the backbone, connected through the skip connections, is $C = [1, 21, 54, 72, 108, 168]$. The number of channels in each residual block is $N = [96, 192, 288, 288, 576, 576]$. Both models map to 128 final invariant features. The conventional model uses a final $3 \times 3$ max pooling over the spatial dimension. In the equivariant model, we do not pad the convolution in the last residual block such that its output already has $1 \times 1$ resolution. However, the last block maps to 128 copies of the regular representation of $G = \text{C}_8$; we then apply *group pooling* over them to generate 128 invariant features. Finally, both models use 2 fully connected layers to perform classification.

We train both models using Adam. We performed simple tuning of batch size, learning rate and weight decay of each model independently by evaluating them on the validation set. The same number of hyper-parameter combinations was evaluated for both models. Once the best set of hyper-parameters was chosen, we re-trained a model using this configuration 5 more times using different random seeds. We report the means and the standard deviations of the test performance over these runs.

## H.5 MODELNET10 EXPERIMENTS

To generate the two ModelNet10 datasets, we use the original mesh data. Each mesh is centered and scaled to fit in the $[-1, 1]^3$ cube. Then, for each mesh we apply 3 random rotations. In the normal ModelNet10 experiments, we only sample $\text{SO}(2)$ rotations around the $Z$ axis. In the rotated ModelNet10 experiments, we sample arbitrary $\text{SO}(3)$ rotations. A mesh is then rendered by computing the *signed distance function* of each point in the voxel grid from the mesh. This distance is then used into a Gaussian kernel to generate a smooth 3D occupancy grid of resolution 33 pixels.

The architecture used consists of a first `Conv3D` with `kernel_size = 5`, followed by 4 residual blocks and final `Conv3D` layer. The second convolution layer of the 4 residual blocks uses stride 2. All models map to 128 final invariant features. The last convolution layer has `kernel_size = 3` In the conventional networks, the last convolution layer has 128 output channels and uses `padding = 1`, so its output has resolution 3px; we apply *max pooling* with a window of size 3 to generate 128 translation invariant features of resolution 1 pixel. In the equivariant networks, the last convolution uses `padding = 0` such that the output resolution is already 1 pixel; however, the last block maps to a larger number of channels $128 \cdot P$, depending on the *invariant map* chosen. We choose this design to maintain the computational and memory cost of the equivariant and the conventional models comparable; indeed, the final features of the conventional network have shape $128 \times 3^3$ while the features of the equivariant one have shape $128 \cdot P \times 1^3$, before applying, respectively, max pooling or an invariant map. We try to keep $P \approx 3^3 = 27$. In the architectures, we use $N = [240, 480, 480, 960]$ channels in the residual blocks and $C = [39, 78, 240, 480, 312]$ channels in the backbone. Finally, three fully connected layers, alternated with dropout, batchnorm and `ELU` non-linearity, use these invariant features to perform classification.

For the $\text{SO}(2)$ model, we used regular non-linearities with regular representations band-limited up to frequency 3 and $|\mathcal{G}| = 12$ samples on a regular grid. Similarly, for the $\text{O}(2)$, $\text{Inv} \times \text{SO}(2)$ and

$\text{Inv} \times \text{O}(2)$ models, we used regular representations band-limited up to frequency 3 and $\mathcal{G}$ was built by extending a grid of 12 points on their $\text{SO}(2)$ subgroup with reflections and flips. In all cases, as invariant map, we used *quotient average pooling*, with $X \cong \mathcal{S}^1 \cong \text{SO}(2)$ and maximum frequency 3.

In the $\text{SO}(3)$ models using regular non-linearities, we used features band-limited to frequency 2 when $|\mathcal{G}| = 60$ and to frequency 3 when $|\mathcal{G}| = 120$. For the $\text{O}(3)$ model, we used regular features band-limited up to frequency 2. In both $\text{SO}(3)$ and $\text{O}(3)$ models using quotient features, we used features band-limited up to frequency 2.

For each equivariant model, the choice of feature types in the backbone (the input and output types of each residual block) is independent of the non-linearities used. Moreover, since no non-linearities are applied on these features, w.l.o.g. we can consider only irreps feature types. We consider the multiplicity of the irreps a hyperparameter which we tune during hyper-parameter tuning.

We train all models using Adam. For each model, we independently performed simple tuning of batch size, learning rate and weight decay by evaluating them on the validation set. This hyper-parameter search also includes a search over the variants of each architecture design (see the variants of each non-linear operation described in Section H.3). The same number of hyper-parameter combinations was evaluated for all models. Once the best set of hyper-parameters was chosen, we re-trained a model using this configuration 5 more times using different random seeds. We report the means and the standard deviations of the test performance over these runs.

## H.6    AUTO-ENCODER EXPERIMENTS ON MODELNET10

For completeness, here we extend the comparison between the steerable bases discussed in Section 5.2 with an auto-encoding experiment with ModelNet10 data. In this experiment, we use the rotated voxel data from Section H.5 and train a model to reconstruct its own input. We consider an auto-encoder architecture whose encoder and decoder comprise 5 blocks, each up/downsampling the features by a factor of 2, such that the latent features have no spatial resolution. Each block is a residual block including a sequence `Conv3D - BatchNorm - ELU` followed by either a `Conv3D` or a `Conv3DTransposed`; all convolutions have `kernel_size = 5` and `padding = 2`, and the last `Conv3D`/`Conv3DTransposed` has `stride = 2`. The residual connection down/upsamples with a Gaussian kernel followed by a learnable $1 \times 1 \times 1$ convolution. Finally, the model includes an *invariant* map and a `Conv3D` layer to predict a single value at each voxel cell.

The outputs of the encoder's blocks have in order approximatively $[30, 90, 240, 540, 1080]$ output channels, while the decoder blocks have $[540, 240, 120, 90, 50]$ output channels. The `ELU` modules within each block have about $[180, 360, 600, 840, 1080]$ and $[1080, 840, 600, 360, 180]$ channels.

The models are trained on the data from Section H.5, which include 3 random rotations of each mesh. Testing is performed on a subset of the test meshes but each mesh is rotated by each element of the icosahedral group (i.e. $|\text{I}| = 60$ times), since the purpose is verifying the equivariance of the model to the icosahedral group I. As in Section 5.2, we consider two versions of the same model which differ by the choice of steerable basis employed. The following table reports the test mean-squared errors (MSE) of the two models.

| Description | MSE |
|---|---|
| Icosahedral Symmetry | $0.785 \pm 0.130$ |
| Icosahedral Symmetry (finite orbits basis) | $0.948 \pm 0.004$ |

Once again, we find the model using our anti-aliased basis outperforms the baseline using a finite-orbits basis.

## H.7    COMPUTATIONAL COST AND INFERENCE TIME OF THE MODELS

In Section H.2, we have argued that point-wise non-linearities based on discrete Fourier Transform and inverse Fourier Transform benefit from a lower computational cost due to the factorized linear layers. In this section, we verify this by comparing the inference time measured for different architectures. We measure inference time by feeding 100 batches of 8 random $33 \times 33 \times 33$ inputs

(we first feed 10 more batches for warm-up). We report mean and standard-deviation over these runs in Tab. 4. The architectures used are the ones described in Section H.3 and H.5. For reference, we also report the number of parameters of each model. Note also that the equivariant models include a small overhead due to the less optimized implementation of certain operations: while the I model has approximatively the same running time of a conventional CNN, the gated non-linearity model and the scaled-up regular model (marked with ∗) are slower that the first two. Despite this overhead, the normal regular $SO(3)$ model (without ∗) is much faster than all other models.

Table 4: Approximate inference times. ∗ indicates wider models to preserve the computational cost.

| $G$ | Description | time (ms) | # params ($\times 10^6$) |
|---|---|---|---|
| $\{e\}$ | Conventional CNN | $41.6 \pm 0.2$ | 36 |
| I | Icosahedral Symmetry | $43.1 \pm 0.2$ | 0.49 |
| $SO(3)$ | Chiral (Gated non-linearities) | $52.2 \pm 0.2$ | 0.37 |
| $SO(3)$ | Chiral (Regular, $\|\mathcal{G}\| = 96$) | $28.2 \pm 0.2$ | 0.25 |
| $SO(3)$ | Chiral (Regular, $\|\mathcal{G}\| = 192$)* | $51.6 \pm 0.2$ | 0.24 |

## H.8 LBA EXPERIMENTS

To perform the LBA experiments, we use the `atom3d` library from Townshend et al. (2020). In particular, we adapted the 3DCNN example from their repository. In the example code, each test molecule is randomly rotated at every run of the test. To make the testing procedure deterministic, we generate a test set by rotating each test molecule with 5 random $SO(3)$ rotations. The occupancy grid of a molecule is generated by setting to 1 the voxel cell which is closest to each atom. This results in a coarser rendering of the data, with respect to the ModelNet10 datasets. To mitigate the discretization noise, we use voxels with a higher resolution of 65 pixels. Note that Townshend et al. (2020) used voxel grids with resolution 40 pixels. However, the larger resolution of the input data can drastically increase the memory cost of the models. To prevent this, all models include a first `Conv3D` layer with `kernel_size` $= 7$ and `stride` $= 2$ which immediately downsamples the input. This first convolution layer is then followed by 4 residual blocks. In each residual block, the `stride` $= 2$ convolution is performed in the first `Conv3D` rather than the second. In the conventional network, the output resolution is 3px and we use *max pooling* to reduce it to 1 pixel. In the equivariant models, the last residual block does not use padding such that the output resolution is already 1 pixel. However, this last equivariant block has more output channels than the conventional network since this block is followed from an *invariant map* layer. See Section H.5 for more details about this design choice. In the architectures, we use $N = [120, 120, 240, 480]$ channels in the residual blocks and $C = [26, 32, 64, 128]$ channels in the backbone. Finally, all models produce 256 invariant features and terminate with two fully-connected layers, alternated with dropout, batchnorm and `ELU`, which perform regression.

In the $SO(3)$ architecture, we use regular non-linearities with regular features band-limited up to frequency 2. Because of the lower computational cost of this architecture, we scale up the number of channels in the residual block such that the computational cost is comparable with that of a conventional CNN. This results approximatively in twice the number of channels of the original model; see the paragraph *regular non-linearities* in Section H.3 for more details about the computational cost of these operations.

To make our results comparable with Townshend et al. (2020), we use the same train, validation and test split. Moreover, during training, we augment each molecule with a random $SO(3)$ rotation, as done in Townshend et al. (2020).

The training details and hyper-parameter tuning procedures are similar to those described in Section H.5.

