# OpenReview forum: "A Program to Build E(N)-Equivariant Steerable CNNs "
_ICLR.cc/2022/Conference — ICLR 2022 Poster_

### Official Review · Reviewer_3dsB · 2021-11-01

**Correctness:** 3
**Technical Novelty And Significance:** 3
**Empirical Novelty And Significance:** 3
**Recommendation:** 6
**Confidence:** 2

**Main Review:**

This paper focuses on convolutional neural networks that are equivariant with respect to certain group symmetries in arbitrary spaces. It extends upon previous paper (Lang & Weiler (2020)), and the benefits are clear: (1) it allows for parametrization with a more flexible and compact basis; (2) it is easier to implement steerable CNNs that are equivariant to a composition of group symmetries.

The paper shows extensive model designs on publicly available datasets. The flexibility of the proposed method is a bonus point. As the paper aims for generality, I would not down-score the contribution for not achieving the state-of-the-art against every other method working on a specific type of symmetry. Nonetheless, very few recent works were compared in the paper, which makes it hard to have an understanding of the practical performance of the proposed method.

I appreciate the extensive details provided in the supplementary materials. However, it makes the reviewing process difficult, as it is nearly impossible to check on every detail in the supplementary given such a tight review window. In addition, many concepts are well explained in the related works such as the irreducible representation, etc.

**Summary Of The Paper:**

The paper generalizes the Wigner-Eckart theorem (Lang & Weiler (2020)) from their own homogeneous space to arbitrary spaces.

**Summary Of The Review:**

To summarize, the contribution of this paper is solid. However, an evaluation of the practical performance compared to recent works and a better organization of the paper structure (especially the length and the complexity) would benefit the paper.

---

> ### Author Response · Authors · 2021-11-19
> **comparison with related works, appendix**
>
>
> We appreciate the reviewer found our contribution solid.
>
>
> Regarding the comparison with recent works, Tab. 1 includes our re-implementations of the most relevant related works using our method.
> We now include explicit citations to the corresponding works in the Tab. 1 to clarify this.
>
> Since representation theory has been well explained in many previous works, we only gave a self-contained minimal overview of the essential concepts in Sec 2.1 and 2.2, and pointed to some related work for a detailed explanation.
> While our Appendix is quite extensive, we believe all important concepts are described in the main paper.
> Appendix A more precisely formalizes this representation theoretic concepts, while the purpose of the rest of Appendix is only providing a precise proof of our results (Apx B) and some additional tools to simplify and automate some constructions (Apx C, D, E, F) in a practical implementation.

---

### Official Review · Reviewer_RMPC · 2021-11-03

**Correctness:** 4
**Technical Novelty And Significance:** 3
**Empirical Novelty And Significance:** 2
**Recommendation:** 6
**Confidence:** 3

**Main Review:**

I was a bit torn about this paper. While I list various strengths and weaknesses below, my high level thoughts are that the idea seems quite nice, but the current exposition is very difficult to follow for someone who is not a domain expert. Perhaps that is fine; there is definitely a community that is well-versed in the group theory underlying generalized equivariant models, but this paper probably won’t be approached by people outside of this niche. Finally, while the theory is elegant and general, I don’t know that the authors did a good enough job of motivating why we need such exotic CNNs (beyond O(3), SO(3), and SO(2)). At the end of the day, I lean towards acceptance but I’m on the fence.

On the one hand:
1. The core idea of the paper seems elegant. In particular, using restriction to generate novel kernels seems like a good idea since E(n) is itself well-studied.
2. The general problem of building equivariant models seems important and is an area with quite a bit of active interest.
3. The supplementary material seems very thorough.
4. The paper itself is quite well-written.
5. The experiments demonstrate a pretty impressive array of examples.
6. I think it will significantly improve the impact of the paper if the authors release code to generate the equivariant kernels (as they mention). However, it would have been nice if the authors had included code along with the submission so we could vet how usable it is for novel research.

At the same time, I do think the current version of the paper has some significant issues. Many of these might arise from tension between the complexity of the theory developed by the authors and the short form of ICLR conference papers. I do think the authors might consider submitting the paper to a journal that would support the longer form factor.
1. I found the exposition to be very unapproachable. For an incomplete list of examples where I thought the current structure was unnecessarily difficult see below.
2. Related to 1) it really seems like one would have to read the supplementary material to get a lot out of this paper.
3. Although I liked the examples discussed by the authors (SO(2) and C^4), I do think the paper would be better served if there was more detail in the constructions. In particular, I think the C^4 example could have explicitly constructed the kernels (perhaps in the SI).
4. While the experiments were performed on a wide range of kernels, there was not a lot of synthesis of the results (aside from weak suggestions about which model might be appropriate for which task). This seems especially true since the accuracies all seemed pretty close to one another.

As I mention above, my main complaint is that the current structure of the paper makes it unnecessarily difficult to read. Some specific instances include:
1. The authors talk a lot about the Wigner-Eckart theorem but they don’t describe what it is until section 2.2. It might be worth describing it qualitatively earlier in the paper.
2. The authors introduce (R^n, +) x G in the introduction but they don’t actually define it until section 2.1.
3. Also in section 2.1 the authors might want to state what Ind_G^{(R^n, +)xG} is (presumably the induced representation of \rho, but why does it not depend on d_\rho? Also, what is t supposed to be?
4. “Band limited” is not defined (as far as I could tell) until example 2.
5. Reference to “the constraint” in sec. 2.2. Presumably this is referring to the G-steerability constraint, why not state it by name.




**Summary Of The Paper:**

This paper introduces a general framework for constructing equivariant steerable CNNs over arbitrary subgroups of E(3). The central idea of their approach is to use the well-known basis for steerable kernels over E(3) and then restrict them to a given subgroup. In addition to a thorough supplementary material developing the theory, the authors also provide pseudo-code for generating novel kernels. This theory is then put into practice by generating steerable CNNs for an impressively wide array of groups and evaluating their performance on rotated MNIST and ModelNet10.

**Summary Of The Review:**

This paper presents an elegant and general technique for constructing equivariant kernels for subgroups of E(3). The paper is generally well-written and the authors intend on releasing code along with the paper to help researchers build their own kernels. However, the current exposition is quite difficult to approach and it seems like one would really need to read the long (40 page) appendix to get the most out of the bulk of the paper. While the authors present a wide range of experiments, they don't really provide compelling evidence that this general construction of exotic subgroups of E(3) will lead to tangible benefits. Overall, I support this paper since it seems correct and there might be researchers who will benefit from being able to construct these CNNs, but I do think the paper has issues that limit its impact as written.

---

> ### Author Response · Authors · 2021-11-19
> **complex presentation,  discussion of the experiments, code**
>
> We are happy the reviewer found our theory elegant and interesting, and that the large number of experiments was appreciated.
> We will release all the code publicly with the camera-ready version of the paper; for the moment, we will soon upload a preliminary version of our code.
> Please, note that some parts had to be removed to preserve the anonymity of the submission.
> Additionally, the current code only supports an initial steerable basis $\mathcal{B}'$ generated from the orbits of $G'$ since we always pick $G'=O(3)$ and a spherical-Gaussian profile basis in our experiments; we plan to support more general steerable bases in the final release.
>
> We try to answer the main concerns of the reviewer:
>
> 1. (and 2.) We agree our presentation is quite complex due to the large technical background knowledge required.
> However, our section 2.1 and 2.2 should be self-contained, introducing all required concepts with minimal overhead.
> Appendix A more precisely formalizes these representation theoretic concepts, allowing for a precise proof of our results in Apx B, while Apx C, D, E and F introduce some additional tools to simplify and automate some constructions in a practical implementation.
> This is not necessary for describing our final results.
> Finally, our paper builds on many previous works which have already described the theory and framework of steerable CNNs in extensive detail.
>
> 3. In the camera-ready version, we will include a section in the Appendix to more precisely describe the $SO(2)$ and $C_4$ examples from the main paper.
>
> 4. Regarding the experiments, we included an additional section discussing our results in the new version of the paper.
> We emphasise that the purpose of the experiments is only that of demonstrating the flexibility and generality of our method.
> For this reason, we do not envision specific applications for all exotic subgroups of $O(3)$ considered; while some are practically relevant (e.g. platonic and azimuthal symmetries), others are included only for demonstration purpose (see also this [comment](https://openreview.net/forum?id=WE4qe9xlnQw&noteId=nh4Ekiw1qyQ)).
> We also agree some methods achieve comparable performance (although always much better than conventional CNNs), but we show interesting computational benefits in Fourier non-linearities based methods (see also Appendix H.5 in the updated manuscript).
>
>
> Finally, we included the other comments in the updated manuscript.
> We defined "band-limited", mentioned that $\mathbb{R}^n \rtimes G$ is a semi-direct product group and added a qualitative description of the Wigner Eckart theorem in the introduction; we also referred to the steerability constraint in Sec 2.2.

---

### Official Review · Reviewer_onv5 · 2021-11-04

**Correctness:** 4
**Technical Novelty And Significance:** 3
**Empirical Novelty And Significance:** 3
**Recommendation:** 6
**Confidence:** 3

**Main Review:**

Strengths:

1. The authors provide a generalization of the Wigner-Eckart theorem for steerable kernels in Lang and Weiler (2020). They relax the requirement for the signal domain $X$ from being a homogeneous space of the group $G$ to an arbitrary space on which $G$ can act. They also provide a simpler computational procedure to build the kernels equivariant to action of $G$.

2. The theorem takes a steerable basis constructed for a larger group $G'$ and builds a basis for the group of interest $G$ by using group restriction. Group restriction takes a representation for $G'$ and converts it into a representation for $G$ using irrep decomposition projections.

3. Using this theorem, the authors show that by constructing a steerable basis for a large group $G'$, like O(n), one can easily construct steerable kernels easily for $G$, a subgroup of $G'$, without having to construct a new steerable basis for every $G < G'$.

4. The idea of a general recipe for constructing equivariant networks that is useful for a large number of subgroup transformations is interesting. My guess is it can be useful when the exact desired symmetry is unknown and only small modifications to the implementation are needed to test out equivariance to all the relevant subgroups of the original base group for which steerable equivariant kernels were designed.

5. The experimental results for 3D symmetries are also interesting and show the usefulness of such a recipe for both shape recognition and molecular datasets.


Questions and suggestions for improvement:

1. I am a little unclear on the practical significance of such an idea. The idea of equivariance is attractive as it has guarantees of robustness to transformations applied at the input. However, when equivariance is enforced for a larger or different symmetry group than required, performance can become worse. Usually, which equvariances should be enforced comes from domain knowledge, although this may be approximate (like full rotation equivariance for natural image recognition, even though not all rotations appear at the same rate in the dataset) or unclear. If we know which equivariance needs to be achieved for a dataset, is it much harder to devise steerable kernels for only the desired group?

2. Can the authors comment on whether they are seeing this paper as a way to exhaustively search all the subgroups of a larger group like O(3) in order to find the best performing equivariant network for a specific dataset? Wouldn't this be quite a slow process?

3. The paper can be made stronger with an application like omnidirectional image recognition where $X$ is a non-Euclidean manifold, rather than just \mathbb{R}^3 and SO(3) equivariance may not be necessary but only SO(2) as the cameras usually are upright.

4. The authors should also comment on how these ideas relate to papers which attempt to find the symmetries directly from data. Can we extend the ideas in the paper to find the right group restriction in an efficient manner perhaps?

**Summary Of The Paper:**

The authors present a general algorithm for constructing steerable equivariant neural networks. The main idea is that by constructing a steerable basis for a large symmetry group such as O(3), using the ideas in the paper, we can readily construct equivariant networks which are equivariants to smaller subgroups, such as SO(2). Experimental results on subgroups of isometries of $\mathbb{R}^3$ show the usefulness of the method for 3D shape recognition for voxelized data and a regression task related to molecular binding affinity.

**Summary Of The Review:**

The paper presents interesting theoretical ideas that may give a practical solution to the problem of designing equivariant steerable kernels to a large number of groups and arbitrary signal domains. The experiments are done reasonably well and provide some evidence for the utility of the method. The authors should try to improve the motivation of the paper and if possible, present more possible application areas where they see these ideas being most useful. Overall, I am inclining towards acceptance of the paper.

UPDATE AFTER AUTHOR RESPONSE:

I thank the authors for a clear response to my questions as well as the explanations. Overall, I am not convinced about the motivation of this paper and its practical significance. If, as the authors say, only some of the subgroups are of O(3) are "occasionally practically relevant", then the ideas at the moment are not very practically impactful. They may be in the future, but I still don't see why designing equivariance for the subgroups of interest for an application directly is really more difficult. I am sticking to my original rating, but the paper does have theoretically interesting ideas which may be useful to the field.

---

> ### Author Response · Authors · 2021-11-19
> **Motivation, complexity of steerable basis design, equivariance group search, omnidirectional image application**
>
>
> We are happy the reviewer found our theoretical results interesting and supported by our experiments.
>
> We agree the paper requires improvements on the motivations, in particular for the different subgroups used.
> We include more comments about this in the updated version.
> In particular, we emphasise that the goal of the paper is designing a general method to build arbitrary G-steerable kernels, which we do in Theorem 2.1.
> The choice of focusing on the 3D setting is mostly motivated by the large number of isometries available, which can be used to demonstrate the generality of our work.
> In other words, we do not necessarily envision applications of all subgroup considered, but some are occasionally practically relevant, such as platonic and azimuthal symmetries.
> The introduction of subgroups and group restriction with Theorem 2.2 is purely for practical convenience: manually designing an initial steerable basis for each group is not trivial; being able to reuse a steerable basis defined for a larger group solves this practicality issue.
>
> We now try to answer the other questions of the reviewer:
>
> 1. Regarding the complexity of designing these basis (and, therefore, the practical significance of our method), the only general strategy we can currently think of is described in Approach 1) in [this reply](https://openreview.net/forum?id=WE4qe9xlnQw&noteId=ekK0HclHlD) : it considers a finite number of orbits, parameterize filters over them and, finally, diffuse the filter in the ambient space.
> This is also what we do to build the initial $G'=G=O(3)$-steerable basis, by means of multiple spherical shells and a Gaussian radial profile.
> However, in our numerical experiments we have shown that such approach performs worse and has higher equivariance error for a smaller group such as the icosahedral one; see Sec. 5.2.
> We believe this is due to the larger freedom in choosing the orbits, which means the particular design choice has a stronger influence on the effective equivariance and final performance.
> Theorem 2.2 provides a simpler way to design steerable bases, reducing the negative influence of bad design choices by relying on an already working $G'$-steerable basis (such as the spherical-Gaussian profile one, widely used in the literature).
>
> 2. and 4.
> We did not originally think of this program directly as a way to search over different groups to discover the right / best performing symmetry, but we also find this is an interesting application.
> Indeed, our Tab. 1 is doing something quite similar in a sense.
> We agree that an exhaustive search is probably not the best strategy, but we believe our program can be an important part of the solution to this problem (e.g. in combination with better search strategies, as done in the Neural Architecture Search (NAS) literature), although this is beyond the scope of this work.
>
> 3.
> Despite our theory supports steerable kernels on an arbitrary space $X$, it solves the kernel constraint in Eq. 2, which only applies to *semi-direct product* groups of the form $X \rtimes G$.
> In other words, while the theory proposed covers any space $X$, only the case where $X$ is also a group is of relevance.
> Indeed, the constraint associated with steerable convolution on a homogeneous space $X = K/G$ is more complicated; see (Cohen et al., 2018).
> This means that the setting $X = SO(3)/SO(2) = S^2$ in the omnidirectional image recognition application is not directly supported by our method.
> We leave this more general setting for future work.
> Note that our method can be used to implement a Gauge-equivariant CNN over a sphere; enforcing only azimuthal symmetry in a Gauge-equivariant network, however, requires some additional modifications which go beyond the scope of this work.

---

> > ### Comment · Reviewer_onv5 · 2021-11-29
> > **Thank you for the response**
> >
> > I thank the authors for a clear response to my questions as well as the explanations. Overall, I am not convinced about the motivation of this paper and its practical significance. If, as the authors say, only some of the subgroups are of O(3) are "occasionally practically relevant", then the ideas at the moment are not very practically impactful. They may be in the future, but I still don't see why designing equivariance for the subgroups of interest for an application directly is really more difficult. I am sticking to my original rating, but the paper does have theoretically interesting ideas which may be useful to the field.

---

### Official Review · Reviewer_oCia · 2021-11-05

**Correctness:** 4
**Technical Novelty And Significance:** 3
**Empirical Novelty And Significance:** 2
**Recommendation:** 8
**Confidence:** 2

**Main Review:**

The suggested method seems like an important contribution to steerable kernel designs. The shortcomings of previous approaches are well explained. Overall the paper is well written. I appreciate the detailed implementation details section. However, I do have some concerns, detailed next.

*Comparison to Lang & Weiler (2020)*

Lang & Weiler (2020) parametrize kernels independently on each orbit. The authors identify that this could lead to non-smoothness. However, is it true to say that it is not necessary? That is, could it be possible the filters were learned to be somehow close between orbits? is there a theoretical justification for that?
Would it be possible to design a toy experiment and visualize learned filters with the current approach versus Lang & Weiler (2020)? In some sense figure 2 and 3 tells the same story, so I would have liked to see a figure showing learned filters from some toy example instead.

*Experiments*

The method is evaluated on three invariant tasks. I think that it is also important to test the method on an equivariant task. One example that comes to mind, which I think could also serve as a nice motivation for steerable networks, is learning a steerable latent space (for example with autoencoders). So for example on modelent10 (the dataset used in one of the experiments), would the method be able to generate rotated shapes by rotating in the latent space? Would the method be able to disentangle between invariant to equivariant features?

Why isn’t the method compared to other equivariant methods in the modelnet10 experiment?


**Summary Of The Paper:**

The paper suggests an approach to design arbitrary steerable kernels. As the kernel steerable constraint only relates to the orbits, previous approaches have not yet taken into consideration that kernels should vary smoothly among close orbits. The idea is to parametrize kernels using a steerable basis. Then, it is also possible to restrict it to subgroups, alleviating the need to redesign a new subgroup equivariant. The method is evaluated on three invariant tasks.

**Summary Of The Review:**

* Seems like a solid contribution to steerable network design.

* Paper is well written.

* I would expected to see the method tested on some equivariant task as well.

* The applicability of the method to 3D classification is not clear as it is compared to other alternatives.

---

> ### Author Response · Authors · 2021-11-19
> **Comparison with Lang et al. and Experiments**
>
>
> We appreciate the reviewer found our manuscript well-written and our contribution solid and important.
>
> **Comparison with Lang \& al.**
>
> The parameterization proposed in Lang \& Weiler can not be directly implemented since it parameterizes an infinite dimensional filter space (there is an uncountable number of orbits to parameterize independently).
> Because any implementation needs to perform some kind of discretization, it is not possible to set up an experiment where the reviewer's hypothesis could be tested.
> Instead, we see two discretization options for this:
>
> 1) parameterize the filter only on a finite set of orbits (and diffuse the filter in the rest of $\mathbb{R}^n$)
> 2) explicitly characterize the space of steerable kernels as a (infinite dimensional) vector space, build a complete basis for it and, finally, choose a finite subset of the basis spanning a band-limited subspace of filters.
>
> Our work follows approach 2).
> While approach 1) seems the closest to what the reviewer suggested, we emphasise that it already partially addresses the non-smoothness by means of the diffusion used.
> We compare the two approaches in Sec. 5.2 (Sec. 5.3 in the previous version) and find that approach 1) leads to both higher equivariance error and lower accuracy.
> We believe the reason behind this is the fact that approach 1) strongly depends on which and how many orbits to consider and only tried to mitigate smoothness issues via some diffusion; however, this generally does not give direct control over the space spanned by the kernel basis built.
> While this result does not necessarily imply that approach 1) can never learn properly band-limited bases, it suggests that using a smooth basis from the beginning as in approach 2) is beneficial.
>
> **Experiments**
>
> We agree that an experiment with an auto-encoder architecture could be a useful equivariant task to consider.
> Note that by definition of steerable CNNs, rotating the features at any layer will produce rotated features in all following layers; therefore, a similar effect is expected also in the bottleneck of an autoencoder architecture.
> However, such experiment could be a good test to compare the actual stability of different architectures to these transformation.
> We are currently working on a similar experiment, which we hope to include in the camera-ready version of the paper.
>
> Regarding the question of whether the model would be able to disentangle features, we believe this question applies generally to steerable CNNs and goes beyond the scope of our work, which instead focuses on describing a sufficiently general program to construct such architectures.
>
> Finally, regarding the (rotated) ModelNet10 experiments, we emphasise that Tab 1. includes our reimplementation of most (equivariant) related works; we now include citations in the table but we refer to Sec 5.1 (Sec 5.2 in the previous version) for more details.
> Due to the high cost of each experiment (for fairness, we perform a large search over hyper-parameters and design variants for each architecture before resampling the best one 5 more times), we limited the experiments in Tab. 2 to only a few representative groups.

---

> > ### Comment · Reviewer_oCia · 2021-11-29
> > **Reply to authors**
> >
> > Thank you for the rebuttal.
> >
> > I accept the authors' point regarding the comparison to Lang & Weller. I find the auto-encoder equivariant task and the stability experiment suggested by the authors very interesting (in my opinion it would be more valuable for evaluating equivariant networks than the invariant classification rotated mnist/ modelnet10 experiment).
> >
> > Given that the rebuttal addressed my concerns I decided to raise my score to 8.

---

### Public Comment · ~Bo_Li16 · 2022-02-22
**Thanks for paper**

Hi, I have been a user of E2CNN and the work really helps promote our research in practical steerable tasks. Thanks for the work and congratulations for this new paper being accepted. I was wondering if the code will also be released:-)

BTW, We also had a paper on constructing steerable CNN for 2D cases. Please pardon me to also refer to paper here which I thought might be related:-)

[_Li,et al. "FILTRA: Rethinking Steerable CNN by Filter Transform." ICML 2021._](https://arxiv.org/pdf/2105.11636.pdf)

---

> ### Public Comment · ~Gabriele_Cesa1 · 2022-02-23
> **The code will soon be released**
>
> Hi, I am very happy to read you appreciated our work and found it useful :)
> We will soon release our new library as promised.
>
> Thanks for the reference. We will include it in a next version of the paper.
>
> Best,
> Gabriele

---

### Public Comment · ~Alejandro_Lozano1 · 2023-02-01
**Code for experiments**

Hello, this work is wonderful!
I was wondering if the code used for the experiments has been made public (as it is stated in the reproducibility statement) thanks! :)

---

### Decision · Program_Chairs · 2022-01-20

**Decision:**

Accept (Poster)

**Comment:**

The paper proposes an approach to constructing steerable equivariant CNNs over arbitrary subgroups of E(3), by generalizing the Wigner-Eckart theorem for steerable kernels in Lang & Weiler (2020). The intuitive idea is to use a steerable basis for a large group like O(3) to build a basis for a subgroup of interest like SO(3). Reviewers were generally happy with the author response, finding the paper makes a good contribution to steerable network design, with theoretically interesting ideas. However, there were still questions after the rebuttal about the practical utility of the approach, such as the relevance of subgroups of O(3). Reviewers also felt that much of the material was not written in an accessible way, such that it could only be appreciated by experts working on group equivariant CNNs.

In a final version, the authors should try to make a much clearer case for practical relevance, introduce the key concepts assuming less prior knowledge, and present the material in a way that makes the high-level story more clear, as detailed by reviewers.